# Differences in MOPITT surface level CO retrievals and trends from Level 2 and Level 3 products in coastal grid boxes

Ian Ashpole[1] and Aldona Wiacek[1,2]

[1]Department of Environmental Science, Saint Mary's University, Halifax, Canada

[2]Department of Astronomy and Physics, Saint Mary's University, Halifax, Canada

*Correspondence to*: Ian Ashpole (ian.ashpole@smu.ca)

**Abstract**

Users of MOPITT data are advised to discard retrievals performed over water from analyses. This is because MOPITT retrievals are more sensitive to near-surface CO when performed over land than water, meaning that they have a greater measurement component and are less tied to the a priori CO concentrations (which are taken from a model climatology) that are necessarily used in their retrieval. MOPITT Level 3 (L3) products are a 1° x 1° gridded average of finer resolution (~22 x 22 km) Level 2 (L2) retrievals. In the case of coastal L3 grid boxes, L2 retrievals performed over both land and water may be averaged together to create the L3 product, with L2 retrievals over land not contributing to the average at all in certain situations. This conflicts with data usage recommendations. The aim of this paper is to highlight the consequences that this has on surface level retrievals and their temporal trends in "as-downloaded" L3 data ("L3O"), by comparing them to those obtained if only the L2 retrievals performed over land are averaged to create the L3 product ("L3L"), for all identified coastal L3 MOPITT grid boxes. First, the difference between surface level retrievals in L3L and the corresponding L2 retrievals performed over water ("L3W") is established, for days when they are averaged together to create the L3O product for coastal grid boxes (yielding a L3O surface index of "mixed" ("L3O$_m$")). Mean retrieved VMRs in L3L differ by over 10 ppbv from those in L3W, and temporal trends detected in L3L are between 0.28 ppbv y$^{-1}$ and 0.43 ppbv y$^{-1}$ stronger than in L3W, on average. These L3L – L3W differences are clearly linked to retrieval sensitivity differences, with L3W being more heavily tied to the a priori CO profiles used in the retrieval, which are a model-derived monthly mean climatology that, by definition, has no trend year-to-year. VMRs in the resulting L3O$_M$ are significantly different to L3L for 45 % of all coastal grid boxes, corresponding to 75 % of grid boxes where the L3L – L3W difference is also significant. Just under half of the grid boxes that featured a significant L3L – L3W trend difference also see trends differing significantly between L3L and L3O$_M$. Factors that determine whether L3O$_M$ and L3L differ significantly include proportion of the surface covered by land/water, and the magnitude of land-water contrast in retrieval sensitivity. Comparing the full L3O dataset to L3L, it is shown

that if L3O is filtered so that only retrievals over land (L3O$_L$) are analysed – as recommended – there is a huge loss of days with data for coastal grid boxes. This is because L2 retrievals over land are routinely discarded during the L3O creation process for these grid boxes. There is less data loss if L3O$_M$ retrievals are also retained, but the resulting L3O "land or mixed" (L3O$_{LM}$) subset still has less data days than L3L for 61 % of coastal grid boxes. As shown, these additional days with data feature some influence from retrievals made over water, demonstrably affecting mean VMRs and their trends. Coastal L3 grid boxes contain 33 of the 100 largest coastal cities in the world, by population. Focusing on the L3 grid boxes containing these cities, it is shown that mean VMRs in L3O$_L$ and L3L differ significantly for 11 of the 27 grid boxes that can be compared (there are no L3O$_L$ data for 6 of the grid boxes studied), with 9 of the 18 grid boxes where temporal trend analysis can be performed in L3O$_L$ featuring a trend that is significantly different to that in L3L. These differences are a direct result of the data loss in L3O$_L$ – data that are available in L2 data (and are incorporated into the L3L product created for this study). The L3L – L3O$_{LM}$ mean VMR difference exceeds 10 (22) ppbv for 11 (3) of these 33 grid boxes, significant in 13 cases, with significant temporal trend differences in 5 cases. It is concluded that a L3 product based only on L2 retrievals over land – the L3L product analysed in this paper, available for public download – could be of benefit to MOPITT data users.

## 1. Introduction

Carbon monoxide (CO) is directly emitted into the atmosphere from anthropogenic (e.g. fossil fuel burning) and natural (e.g. wildfire) sources, and also produced via the oxidation of hydrocarbons in the atmosphere. With an atmospheric lifetime of weeks to months (e.g. Duncan et al., 2007), it is an important tracer of pollutant transport and indicator of emission sources. While a health concern at high enough concentrations, CO also plays an important role in atmospheric chemistry, for example as a precursor to ozone formation and a primary sink for the hydroxyl radical. Atmospheric CO concentrations have decreased since the start of the 21$^{st}$ century, with a slowdown in the rate of decline observed in recent years (Buchholz et al., 2021). Trends also show substantial spatial variability (Hedelius et al., 2021). Satellite instruments have been central to our understanding of global change in CO concentrations, with the Measurement of Pollution in the Troposphere (MOPITT – Drummond et al., 2010, 2016; frequently used abbreviations are defined in Appendix A) instrument well suited to this task, providing a nearly-unbroken and consistent data record since the year 2000.

MOPITT observes upwelling radiances at thermal infrared (TIR) and near infrared (NIR) wavelengths and uses these in an optimal estimation retrieval algorithm to retrieve coarse vertical resolution CO profiles,

which are integrated to give total column amounts. Among multiple additional inputs required by the retrieval algorithm, a priori CO profiles – which describe the most probable state of the CO profile at a given location – are necessary to constrain the retrieval to physically reasonable limits (Pan et al., 1998; Rodgers, 2000; the retrieval algorithm is outlined in more detail in Sect. 2.1). For the most recent iterations of MOPITT products, these a priori CO profiles are based on a monthly climatology from a chemical transport model. The degree to which a given MOPITT retrieval reflects information obtained from the observed radiances – known as "information content" – is highly spatially and temporally variable, depending on scene-specific factors such as surface temperature, thermal contrast in the lower troposphere, and the actual ("true") CO loading itself, as well as on instrumental noise (e.g. Deeter et al., 2015). The lower the retrieval information content, the closer the retrieved CO loading will be to the a priori; a model value.

Retrievals that take place over water are known to have a lower information content than retrievals that take place over land. Primarily, this is due to weak thermal contrast near to the surface hampering the instrument's ability to sense CO absorption in the lowermost layers of the troposphere (Deeter et al., 2007; Worden et al., 2010), and this is confounded by a lack of NIR reflectance over water, which limits these retrievals to TIR wavelengths only. It is therefore recommended that MOPITT data users exclude these retrievals from any analyses they perform, to ensure that results are not biased by retrievals that have a heavy reliance on the a priori (MOPITT Algorithm Development Team, 2018; Deeter et al., 2015). Such filtering is specifically emphasised where the focus of analysis is the identification of long-term CO trends, because any real trends in the data will be weakened by the inclusion of retrievals that are tied heavily to the a priori (Deeter et al., 2015). This is because the a priori CO profiles are taken from monthly modelled CO climatologies: for a given location and day of the year, they will be the same every year and therefore feature no temporal trend (Deeter et al., 2014).

MOPITT data are available as either Level 2 ("L2") or Level 3 ("L3") products. L2 products contain each individual retrieval, at ~22 x 22 km spatial resolution. L3 products are a 1° x 1° gridded area-average of the individual L2 retrievals that fall within each grid box (see Fig. 1), with some filtering criteria applied. One criterion is the surface type over which the L2 retrievals were performed – either land, water, or "mixed". If more than 75 % of the bounded L2 retrievals were performed over the same surface type then only those retrievals are averaged to create the L3 product and the rest are discarded; otherwise, all bounded L2 retrievals are averaged, and the L3 product is given the surface type classification of "mixed" (L3 surface type classification is explained in more detail in Sect. 2.2). This creates a problem for L3 grid boxes that overlay coastlines: To a greater or lesser extent, these L3 products will have some contribution from L2 retrievals performed over water, as shown in Fig. 1. L3 product users have limited capability to discard them, at least without sacrificing temporal resolution, because each L3 grid box only has a single "retrieval" per day. By

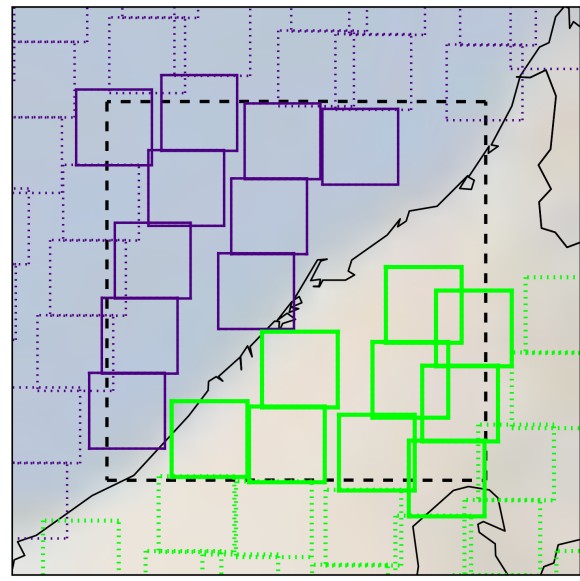

**Figure 1.** Example of a coastal L3 grid box (black dashed box) and bounded L2 retrievals from which the L3 products for that grid box are created. Purple (green) boxes correspond to L2 retrievals with a surface index of "water" ("land"). Note that only L2 retrievals with a midpoint that falls within the boundaries of the L3 grid box will be used in L3 creation for that grid box. These are indicated by solid purple/green outlines – those not included in L3 creation for this grid box are shown with dotted purple/green outlines. More information on surface indexing and L3 product creation is given in Sect. 2.2. "Coastal" L3 grid box classification is outlined in Sect. 2.3. The coastal L3 grid box visualized here contains the city of Dubai (~centre = 55.296° E, 25.277° N), which features in the case study analysis of Sect 3.4. Faint background shading is from NASA Blue Marble imagery.

contrast, with L2 products it is possible, for the same coastal grid boxes, to choose to retain only the retrievals
performed over land. In practical terms, this means that, for coastal L3 grid boxes, valuable retrieval
information over land, available in L2 products, can be lost to users of L3 products.
With a focus on the coastal L3 grid box containing the city of Halifax, Canada, Ashpole and Wiacek
(2020) demonstrate the consequences of this loss of retrieval information in L3 products. They compare the
results of analyses performed using L3 data and L2 data whereby only bounded retrievals performed over
land were retained, and find significant differences in both seasonal mean statistics and the magnitudes of
trends identified in surface level CO. These differences are a direct result of the L3 products being dominated
by L2 retrievals over water, which feature a weaker trend than the L2 retrievals over land, demonstrably due
to a greater a priori influence owing to their reduced true-profile sensitivity, especially close to the surface.
In their conclusions, Ashpole and Wiacek (2020) suggest that L2 retrievals over water should not contribute
to L3 products for coastal grid boxes, which would be consistent with previous data filtering
recommendations (MOPITT Algorithm Development Team, 2018; Deeter et al., 2015). The study presented
here expands that work to the global scale.

The aim of this paper is to compare surface level retrievals and their temporal trends in "as-downloaded" L3 data ("L3O"; a list of dataset short names is given in Table 1) with those that could be obtained if only the L2 retrievals performed over land are averaged to create the L3 product ("L3L", Ashpole and Wiacek (2022) – outlined in Sect. 2.4), for all identified coastal L3 MOPITT grid boxes around the globe. It is necessary to identify whether there are differences for two reasons: firstly, L3 data are more convenient for long time series analysis than L2 data owing to their smaller file size (~25 MB vs ~450 MB respectively, for a single daily, global file). It cannot be overlooked that working with L3 data thus requires fewer computing resources and less technical proficiency, with a range of simple-to-use tools available for working with gridded products. L3 products thus make the MOPITT data more easily accessible, especially to less-expert users, who may lack the expertise required to scrutinize the data for potential a priori bias. Secondly, many of the world's largest agglomerations are situated within a coastal L3 grid box (5 of the top 10 and 33 of the top 100 largest agglomerations by population; derivation outlined in Sect. 2.5), making these likely targets for analyses of air quality indicators, especially their changes over time. The paper focuses on the surface level of the retrieved profile specifically because this can yield information that is of use in identifying potential air quality impacts for humans (e.g. Buchholz et al., 2022), and also because this is the profile level where the greatest land-water differences in retrieved VMR statistics and trends were found in Ashpole and Wiacek (2020).

This paper is structured as follows: Section 2 describes the datasets and methods used, including outlining the creation of the new "land-only" L3 product (L3L), and its "water-only" counterpart ("L3W") created for comparison purposes, which are analysed in this paper. A method for determining which L3 grid boxes are "coastal" is also outlined (Sect. 2.3); these grid boxes are selected as the focus of analysis. Section 3.1 demonstrates the magnitude of the sensitivity difference for retrievals over land and water, zooming in to focus on coastal grid boxes. Although this paper focuses on the surface level of the retrieved vertical profile, higher levels in the profile are also briefly considered here to contextualise the land-water sensitivity contrast at the surface. Section 3.2 links the surface sensitivity contrast to differences in mean CO volume mixing ratios ("VMRs") and their temporal trends for L2 retrievals performed over land and water within coastal L3 grid boxes; and evaluates the effect that the averaging together of these retrievals has on the statistics and trends in resulting L3 "mixed" values. Section 3.3 quantifies the proportion of L2 retrievals performed over land within coastal L3 grid boxes that are lost to L3 products, before finally comparing statistics and trends in L3 and L2 products for all coastal L3 grid boxes, outlining the magnitude and significance of differences for the coastal grid boxes that contain 33 of the largest 100 cities in the world (Sect. 3.4). Results are summarised and conclusions drawn in Sect. 4.

## 2. Data and Methods

### 2.1. MOPITT Instrument and retrieval overview

Carried on board the polar-orbiting NASA Terra satellite that was launched in December 1999, MOPITT began measuring CO in March 2000 and has provided near-continuous measurements to date. With a native pixel resolution of ~22 x 22 km at nadir and a swath width of ~640 km, it offers near global coverage roughly every 3-days, crossing the equator at ~10:30 and ~22:30 local time. The instrument is a gas correlation radiometer that measures radiances in two CO-sensitive spectral bands: the TIR at 4.7 μm, which is sensitive to both absorption and emission by CO and can provide information on its vertical distribution in the troposphere; and the NIR at 2.3 μm, which constrains the CO total column amount and yields information on CO concentrations in the lower troposphere (LT), to which TIR radiances are typically less sensitive (Drummond et al., 2010; Pan et al., 1995, 1998). For the work presented here, the TIR-NIR combined MOPITT product is used, owing to its demonstrably greater sensitivity to CO loadings near to the surface than the TIR- and NIR- only products which are also available (Deeter et al., 2013). Note, however, that retrievals over water and at night are limited to the TIR band only due to the lacking NIR signal. This analysis is based on daytime-only retrievals (more information on data selection and preparation is given in Sect. 2.4).

Multiple other sources describe the retrieval algorithm in detail (e.g., Deeter et al., 2003; Francis et al., 2017). In short, it uses optimal estimation (Pan et al., 1998; Rogers, 2000) and a fast radiative transfer model (Edwards et al., 1999) to invert measured radiances and retrieve the CO volume mixing ratio (VMR) profile on 10 vertical layers. The vertical grid consists of 9 equally spaced pressure levels from 900 to 100 hPa (the uppermost level covers the atmospheric layer from 100 to 50 hPa), with a floating surface pressure level (if the surface pressure is below 900 hPa, less than 10 profile levels are retrieved). Retrieved values represent the mean CO VMR in the layer immediately above that level. These profile measurements are then integrated to provide total column CO amounts. Retrievals are only performed for scenes free of cloud (cloud clearing is based on coincident MODIS observations and MOPITT's own radiances).

In addition to the measured radiances, the retrieval requires multiple inputs including meteorological data, surface temperature and emissivity, and, of direct relevance to this study, a priori CO profiles, which are necessary to constrain the retrieval to physically reasonable limits. These a priori CO profiles come from a monthly CO climatology (years 2000-2009), simulated with the Community Atmosphere Model with Chemistry (CAM-chem) chemical transport model (Lamarque et al., 2012) at a spatial resolution of 1.9º x 2.5º, which is then spatially and temporally interpolated to the time and location of each individual MOPITT observation. A priori profiles for a given location and day of the year are therefore the same every year and

feature no temporal trend. To understand the physical significance of the MOPITT CO retrievals, it is necessary to examine the retrieval Averaging Kernels (AKs), available with all MOPITT data products, which quantify the sensitivity of the retrieved vertical profile to the "true" vertical profile. The lower the retrieval sensitivity, the greater the a priori weighting. Two different components of AKs are analysed in this paper: AK rowsums, which represent the overall sensitivity of the retrieved profile at the corresponding pressure level to the whole true profile; and AK diagonal values, which represent the sensitivity of the retrieved profile at the corresponding pressure level to the same level of the true profile (e.g. the AK diagonal value for the surface level of the retrieved profile represents its sensitivity to the surface level of the true profile).

From time-to-time, new MOPITT products become available as improvements are made to the retrieval algorithm and radiative transfer model, yielding superior validation statistics compared to earlier product versions (Worden et al., 2014). This analysis uses MOPITT Version 8 (V8) products (Deeter et al., 2019). Version 9 (V9) products became available shortly after this study was completed. V9 features cloud screening improvements that yield additional retrievals over land in comparison to V8 (the exact percent change varies significantly with geography). Validation results are comparable to V8. An overview of MOPITT V9 is given by Deeter et al (2022). A subset of the analysis presented in this paper has been duplicated using V9 data, and this confirms that the main conclusions drawn based on V8 data also hold for V9 (this analysis is outlined in the Supp. Mat. (SM1)). This is to be expected, given that the land-water sensitivity contrast remains in V9 and the L3 processing method is unchanged.

**2.2. MOPITT surface type classification**

To aid in filtering and interpreting retrievals, all MOPITT data products are distributed with a range of diagnostic fields. As retrieval information content is known to be variable depending on the type of surface over which it is performed (Deeter et al., 2007), L2 retrievals are given a surface index according to whether they were performed over land, water, or a combination of the two ("mixed"). For a given $1^{\circ}$ x $1^{\circ}$ L3 grid box, how the L2 retrievals that fall within its boundaries are processed to produce the L3 product depends on how their surface indexes vary: If more than 75 % of the bounded L2 retrievals have the same surface index, only those retrievals are averaged to produce the L3 gridded value, and the L3 surface index is set to that surface type (the other L2 retrievals are discarded). Otherwise, all L2 retrievals available in the L3 grid box are averaged together and the L3 surface index is set to "mixed", as is the case in the example shown in Fig.

1 (this information is taken from the MOPITT Version 6 L3 data quality summary[1], which at the time of writing, is the most recent data quality summary to detail exactly how L3 data are created, despite more recent data quality summaries being available). Note that the L2 VMR profiles that are averaged to produce the L3 retrieval are first converted to log(VMR) profiles, then averaged, and the mean log(VMR) profile is then converted back to a VMR profile.

Each L3 grid box only has one retrieval per day. This dictates that where the grid box overlies both land and water, its surface index could vary through time, depending on the population of L2 retrievals from which it is created. The make-up of this population can vary from day-to-day due to factors such as cloud cover, and screening for data quality issues: on day $n$ the population could be predominantly L2 retrievals over land (resulting in a surface index of "land" for the L3 retrieval), on day $n+1$ it could be predominantly L2 retrievals over water (L3 surface index = "water"), and on day $n+2$ it could be an even mix of the two (L3 surface index = "mixed"). Given that the averaging together of retrievals with significantly different sensitivity profiles – as could be the case when averaging retrievals over land and water – serves to dilute the information coming from the MOPITT observed radiances with information coming from the a priori and is therefore discouraged (MOPITT Algorithm Development Team, 2018; Deeter et al., 2015; Deeter et al., 2007); and that MOPITT data users are advised to exclude retrievals over water from analyses owing to the known reduced sensitivity, this introduces two potential problems for L3 data taken from coastal grid boxes: firstly, discarding all L3 retrievals with the surface index of water will result in a loss of temporal coverage; secondly, L3 retrievals with a surface index of mixed feature some contribution from L2 retrievals over water. The consequences of both these problems are explored in this paper.

## 2.3. Coastal grid box classification for this study

Since the focus of this paper is on "coastal" L3 grid boxes, it is first necessary to isolate these from the remaining "land-only" or "water-only" L3 grid boxes in the MOPITT data set. The initial step is to identify all grid boxes that have a surface index of "mixed" at least once during the study period. This indicates that the ground area within those grid boxes was both land and water – a characteristic that can safely be assumed true for coastal grid boxes. However, analysis of the global distribution of L3 grid boxes featuring a surface index of mixed revealed that, in addition to actual coastlines, a large proportion of inland grid boxes that are clearly not coastal are given the surface index of mixed at least once during the study period ("inland_mixed"; Fig. 2a). The reason for this is unclear, but it could be for real physical reasons, such as land grid boxes

[1] available here: https://www2.acom.ucar.edu/mopitt/mopitt-level3-ver6

sporadically flooding, or due to issues in the retrieval schemes caused by e.g. cloud screening problems or the presence of surface ice cover. One characteristic of these inland_mixed grid boxes is that, compared to the total number of days with L3, the relative frequency with which they are flagged as land is very high (expressed as the ratio "n_days($L3O_L$/$L3O$)", plotted in Fig. 2b; a list of short names and abbreviations referred to in the text can be found in Appendix A for reference). This relative frequency is much lower for "true" coastal grid boxes, to be expected given prior knowledge of 1) the fact that these grid boxes span both land and water surface types; and 2) how the surface index is determined for L3 data (as outlined in Sect. 2.2). Following iterative threshold testing, L3 coastal grid boxes are classified as grid boxes that:

1. Have at least one classification of "mixed" during the study period
2. Have an n_days($L3O_L$/$L3O$) ratio < 0.5.

The distribution of coastal grid boxes identified using these criteria is shown in Fig. 2c. Most inland_mixed grid boxes are removed from the classification, although some still pass these criteria and are therefore erroneously classified as coastal, mostly in the north of Canada and Russia. However, placing a more restrictive threshold on the n_days($L3O_L$/$L3O$) ratio to remove these areas has diminishing returns since it results in the rejection of more true coastal grid boxes. These criteria therefore strike a balance between minimising false and maximising true coastal classifications.

Applying these criteria to the MOPITT L3 data yields 4299 coastal grid boxes, from a total of 64800 L3 grid boxes (6.6 %). This mask is applied to all data, and only those L3 grid boxes that remain are classified as coastal. Only data for these coastal grid boxes are analysed in this study (with the exception of global L3 maps analysed in Sect. 3.1.1).

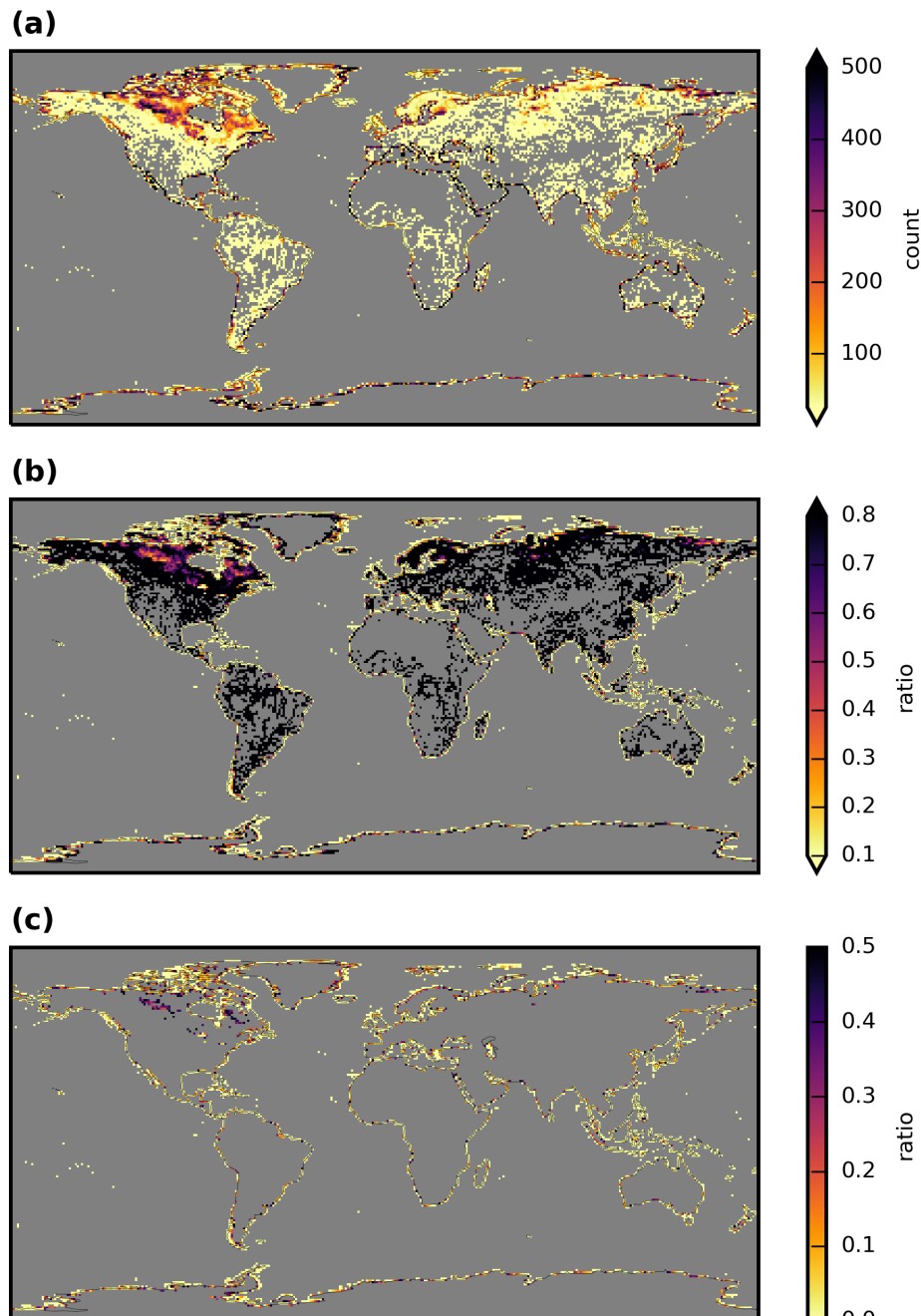

**Figure 2.** Maps showing the stages of derivation of the coastal L3 grid box mask applied in this paper to MOPITT data. **(a)** Frequency with which L3 grid boxes are given the surface index of "mixed", calculated from daily data between 2001-08-25 and 2019-02-28. **(b)** Frequency with which L3 grid boxes that have a surface index of "mixed" at least once in panel a have the surface index of "land", compared to the total number of days with which L3 data are available for that grid box (expressed as n_days(L3O$_L$/L3O)). **(c)** As b, but with a threshold of n_days(L3O$_L$/L3O) < 0.5 applied. This is the coastal L3 grid box mask used in this paper.

**2.4. MOPITT datasets analysed, and data processing method for creating land- and water-only L3 products ("L3L" and "L3W")**

All available MOPITT V8 Level 2 (L2) and Level 3 (L3) daily TIR-NIR files ("MOP02J" and "MOP03J" files, respectively) were downloaded from the NASA Earthdata portal (https://search.earthdata.nasa.gov). Although the data record begins in March 2000, analysis is restricted to the period from 2001-08-25 to 2019-02-28. Data prior to 2001-08-25 are discarded due to an instrumental reconfiguration in 2001 creating an inconsistency in the data record (Drummond et al., 2010). Data post 2019-02-28 are flagged as "beta" at the time of writing, their use in scientific analysis (especially for examining long-term records of CO) being discouraged until final processing and calibration occurs (MOPITT Algorithm Development Team, 2018). For clarity, the original, "as-downloaded" L3 time series is referred to as "L3O" for the remainder of this paper. Only retrievals that were performed during daytime hours are retained (daytime and nighttime retrievals are stored as separate fields in MOP03J files). For this analysis, separate subsets of L3O are created according to surface index: L3O land-only ("$L3O_L$"), L3O water-only ("$L3O_W$"), L3O mixed ("$L3O_M$"), L3O land-or-mixed ("$L3O_{LM}$"). When the L3O dataset is analysed with no filtering by surface index applied, it is referred to as "$L3O_{NF}$". A list of dataset short names used in this article, and their full descriptive name, is given in Table 1.

The land- and water-only L3 products are created from daily L2 data. The first step of L2 data processing required is to filter the retrievals as is done for the processing of L3O. This involves:

- Discarding all observations for Pixel 3 (this corresponds to one of MOPITT's four detectors);
- Discarding all observations where both (1) the channel 5A signal-to-noise-ratio ("SNR") < 1000 and (2) the channel 6A SNR < 400 (5A and 6A correspond to the average radiances for MOPITT's length-modulated cell TIR and NIR channels, respectively)

This filtering takes place because observations from specific elements on MOPITT's detector array were found to exhibit greater retrieval noise than the other elements, and their inclusion therefore lowered overall L3 information content (MOPITT Algorithm Development Team, 2018). Only daytime L2 retrievals are retained, using a solar zenith angle filter of < 80º.

From the remaining set of filtered L2 retrievals, separate area averages are taken for those with a surface index of land and water, for every 1º x 1º L3 grid box. This effectively creates two new L3 "land-only" and "water-only" products, which are referred to herein as "L3L" and "L3W". For clarity of analysis, remaining L2 retrievals with a surface index of mixed are discarded. These make up a very small proportion

of the overall L2 retrievals (e.g. < 5 % for the grid box containing Halifax, analysed in Ashpole and Wiacek,
2020). Both L3L and L3W are publicly available for download (Ashpole and Wiacek, 2022). Note that, as
with the creation of L3O, L2 VMR profiles for each L3 grid box are first converted to log(VMR) profiles
before averaging, and the mean log(VMR) profile is then converted back to a VMR profile to give the final
L3L and L3W retrievals. Additionally, the number of L2 retrievals that are used for calculating the area
averages when creating L3L and L3W ("n_ret$_L$" and "n_ret$_W$", respectively) is recorded. The ratio
n_ret$_L$/n_ret$_W$ (herein referred to as "ratio(land/water)" for simplicity) is used to indicate the proportion of
the L3 grid box that is covered by land vs water: a ratio of 1 indicates an even split of these surface types in
the grid box; a ratio < 1 indicates that a greater proportion of its surface is water covered; and a ratio > 1
indicates that the grid box is land-dominated.

From the L3O, L3L, and L3W datasets, only grid boxes that are classified as "coastal" using the

coastal grid box masked outlined in Sect. 2.3 are analysed (See Table 1 for a list of dataset short names used
in this article, and their full descriptive name).

Note that the analysis presented in this paper is restricted to daily products. Monthly L3 files are

available, however the absence of a monthly L2 product precludes the analysis from being conducted on
those data. Based on the results of the analysis of daily data, however, there is reason to also advise caution
if working with coastal grid boxes in the monthly L3 product. This is because the data for those grid boxes
will still be created from daily L2 retrievals over land and water, with the same implications that are discussed
in this paper.









**Table 1.** List of dataset short names used in the main article text, and their corresponding full descriptive name.

| Dataset short name | Full descriptive dataset name |
|---|---|
| L3O | Original, "as downloaded" Level 3 (L3) dataset |
| $L3O_L$ | Subset of L3O only containing L3 retrievals with a surface index of land |
| $L3O_M$ | Subset of L3O only containing L3 retrievals with a surface index of mixed |
| $L3O_{LM}$ | Subset of L3O only containing L3 retrievals with a surface index of land OR mixed |
| $L3O_W$ | Subset of L3O only containing L3 retrievals with a surface index of water |
| $L3O_{NF}$ | The L3O dataset with no filtering by surface index ($L3O_{NF}$ is identical to L3O) |
| L3L | A new L3 "land-only" dataset, created only from Level 2 retrievals performed over land (creation method outlined in Sect. 2.4) |
| L3W | A new L3 "water-only" dataset, created only from Level 2 retrievals performed over water (creation method outlined in Sect. 2.4) |



## 2.5. Time series preparation, statistical methods, and additional data sources



For every coastal L3 grid box, two separate time series from each of the L3O, L3L, and L3W datasets are
analysed:

1.  The time series analysed in Sect. 3.1 and 3.2 only contain days where L3L and L3W are both present

and the L3O surface index is mixed ("$L3O_M$"). This is to ensure that the true CO profiles are as similar

as possible when directly comparing L3L and L3W for a given coastal grid box. Furthermore, it

allows for the analysis of the resulting $L3O_M$ data on these days with knowledge of the parent L2

retrievals over land and water and their differences.


2.  In Sect. 3.3 and 3.4 the full time series from each dataset is analysed with no temporal filtering

applied.


Descriptive statistics are calculated from both time series across the whole study time period, and also
for individual years (full years only – 2002 to 2018 inclusive) in order to perform the regression analysis
outlined below.
To identify and compare temporal trends for each coastal grid box in the datasets outlined above,
weighted least squares (WLS) regression analyses is performed on yearly mean values, weighted by the
inverse of the standard deviation of the measurements used in the yearly mean (i.e. $1/\sigma$). For years that contain
just a single retrieval, the weighting is set to 1/100000 to de-weight them in the fit. If there are more than 2
years in a time series for a given grid box that have no data, the regression analysis is not performed. WLS
is preferred over OLS because it is less sensitive to outliers. For simplicity, no other trend detection methods
– e.g. the Thiel-Sen slope estimator – are applied to corroborate the trends that are detected with WLS, nor
do we analyse additional datasets to verify them. Such extra steps would be necessary if the actual trend
values were the focus of this study; however, the aim of this trend analysis is instead to identify whether the
same method can yield different results depending on which of L3O, L3L or L3W is analysed. Trend
verification is beyond the scope of this study.
To determine whether two trends identified are significantly different, their difference is evaluated
using the Z test as follows:

$$Z = \frac{Trend_1 - Trend_2}{\sqrt{SE_1^2 + SE_2^2}}$$

where $SE_1$ and $SE_2$ correspond to the standard errors of $Trend_1$ and $Trend_2$ respectively, and Z is the test
statistic. Where Z is greater (less) than 1.645 (-1.645) the trend difference is statistically significant to at least
90 % (i.e. $p < 0.1$). In addition, two trends are classified as being significantly different if $Trend_1$ is
significantly different to zero ($p < 0.1$) but $Trend_2$ is not ($p > 0.1$), and vice-versa (i.e. the conclusion would
be that $Trend_1$ is not zero, but $Trend_2$ may be).
A list of the top 100 largest agglomerations by population in the world is obtained from
http://www.citypopulation.de/ (valid at time of writing). 33 of these are situated in a coastal L3 grid box,
according to the classification in Sect. 2.3. Time series of L3L, L3W, and L3O are extracted from each of
these grid boxes for the analysis in Sect. 3.4.

## 3. Results and Discussion

### 3.1. Land-water contrast in MOPITT sensitivity

This section demonstrates the land-water sensitivity contrast in MOPITT retrievals on a global scale, and examines the magnitude of the difference within coastal L3 grid boxes. The analysis is presented for levels throughout the vertical profile in addition to the surface level, to give context as to how MOPITT retrieval sensitivity, and its land-water contrast, varies with height.

### 3.1.1. Global context

Figure 3 shows long-term mean maps for the retrieval sensitivity metrics AK diagonal value, AK rowsum, and retrieved minus a priori VMR ("VMR ret-apr") at selected profile levels, created from L3O data averaged across the entire study period (September 2001 – February 2019, inclusive). All indicators show that retrieval sensitivity is greater over land than water at the surface, with sharp differences evident at almost all land-water boundaries. The same is true at the 900 hPa and 800 hPa profile levels, although the land-water contrast clearly decreases in strength with height on average, and by 600 hPa retrieval sensitivity tends to be a little greater over water than land. Some strong land-water gradients remain present in VMR ret-apr fields at this level, most notably over North Africa, the Arabian peninsula, and south-east China, but on average these values are much more similar in magnitude across land and water than they were closer to the surface. No clear land-water contrast is evident at 300 hPa (which represents the upper troposphere), with retrieval sensitivity instead varying more with latitude, decreasing towards both poles (a companion to Fig. 3 with an altered colour bar to better show spatial patterns in AK diagonal values and rowsums at the higher profile levels considered here is provided in the Supp. Mat. (SM2)).

AK diagonal values and rowsums clearly show that retrieval sensitivity increases across both land and water with height. It is generally lowest at the surface level, with little information content in the retrieval over water (mean AK diagonal values and rowsums over water are less than half what they are over land). However, there is high spatial variability over land, with clear sensitivity hotspots (e.g. parts of central Europe, east Asia, eastern USA and tropical west Africa), but also some areas where AK values are more comparable to those over water. The rate of sensitivity increase with height is greater over water than land, with AK values more than doubling over water between the surface and 800 hPa.

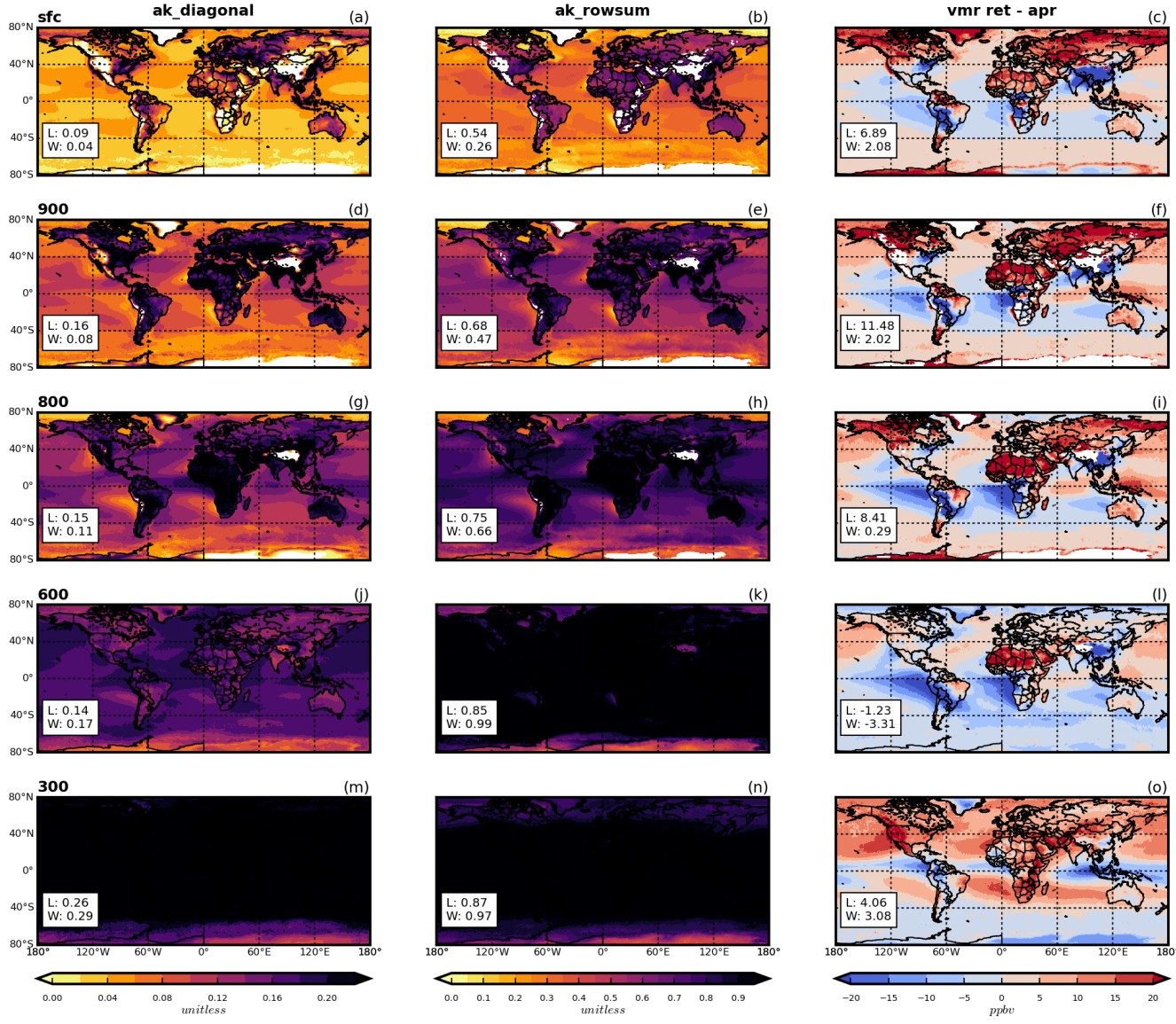

**Figure 3.** Mean sensitivity metrics from MOPITT L3 data, averaged across the entire study period (September 2001 – February 2019, inclusive). Shown are AK diagonal values (left column), AK rowsums (center column) and VMR retrieved minus a priori values (right column) for the following levels of the retrieved profile: surface (top row), 900 hPa (second row), 800 hPa (third row), 600 hPa (fourth row), and 300 hPa (bottom row). Values in white boxes correspond to mean values across all land ("L") and water ("W") L3 grid boxes.

Spatial patterns in retrieved minus a priori VMRs are slightly more complex to interpret, because they are influenced both by retrieval sensitivity and the accuracy of the a priori. For example, while VMR ret-apr values close to zero can indicate a retrieval that is heavily weighted by the a priori and therefore low retrieval sensitivity, they can also indicate that the true VMR is close to the a priori value. Despite this, retrieved minus a priori VMR values clearly reach more strongly positive or negative values over land than water at the surface, with the contrast becoming less pronounced with height. Furthermore, there are clear land-water changepoints, further demonstrating the impact of the land-water contrast in retrieval sensitivity.

An analysis of latitudinal and seasonal variability in the land-water surface level retrieval sensitivity contrast is provided in the Supp. Mat. (SM3). Briefly, this shows a tendency for greater land-water retrieval sensitivity differences in the Northern Hemisphere than Southern Hemisphere when averaged across the year. The land-water AK rowsum differences tend to vary least by season in the tropical regions (between 30º South and 30º North) and show the greatest contrast in the midlatitudes (30º – 60º) in the respective hemisphere's spring and summer months, with smallest differences in the winter months. Overall, a land-water sensitivity contrast is evident irrespective of latitude or season.

**3.1.2. Analysis of coastal L3 grid boxes**

Scatterplots of the sensitivity metrics discussed above, for coastal L3 grid boxes only, are shown in Fig. 4. Specifically, these plots show the sensitivity of the L2 retrievals over land and water that are bounded by the 1º x 1º L3 grid boxes and used to create the L3O data – represented here by L3L and L3W. As noted in Sect. 2.5, the time series analysed in this section only contain days where L3L and L3W are both present and the L3O surface index is mixed ("L3O$_M$"), for a given coastal grid box. This is to ensure that the true CO profiles are as similar as possible when directly comparing L3L and L3W for that grid box. The values that are plotted correspond to the long-term mean from these L3L and L3W timeseries.

The AK diagonal value and rowsum plots clearly demonstrate the greater sensitivity over land (L3L) than over water (L3W) at the surface level (a point below the diagonal line on these panels indicates greater values in L3L) for most grid boxes, with the difference decreasing with height, as expected from the preceding analysis. Retrieved VMRs also deviate more greatly from their a priori values in L3L than L3W closer to the surface, with smaller land-water differences higher up in the retrieved profile. All mean values are significantly different ($p < 0.005$) apart from AK diagonal values at 300 hPa and retrieved minus a priori VMR at 300 hPa ($p = 0.13$ and $0.07$ respectively). Sensitivity metrics are generally better correlated over land and water higher in the retrieved profile than at the surface.

This analysis clearly shows how L2 retrievals that are averaged together to create the L3O data over
coastal grid boxes have differing degrees of sensitivity, depending on the surface type that they were retrieved
over, especially at the surface and lower profile levels. This is explicitly cautioned against in the MOPITT
data user's guide (MOPITT Algorithm Development Team, 2018). The remainder of this paper focuses on
the surface level of the retrieved profile, since this is where land-water discrepancies are greatest, and the
cause of this sensitivity disparity is well established: differing thermal contrast conditions near to the surface
over land and water; and a lack of NIR radiances being used in the retrieval over water. Furthermore, surface
level retrievals are of most interest for identifying potential air quality impacts for humans (e.g. Buchholz et
al., 2022).


**Figure 4. (next page)** Mean sensitivity metrics and VMRs (retrieved and a priori) from coastal L3 grid boxes. Values compared in the scatterplots are mean values from matched L3L and L3W retrievals within these grid boxes. "Matched" means that only days when both L3L and L3W are present, and the L3O surface index is mixed, are used to create the mean values analysed. Shown are AK diagonal values (left column), AK rowsums (second column), absolute VMR retrieved minus a priori values[1] (third column), retrieved (fourth column) and a priori (fifth column) VMRs, for the following levels of the retrieved profile: surface (top row), 900 hPa (second row), 800 hPa (third row), 600 hPa (fourth row), and 300 hPa (bottom row). Values in boxes in the top-left corner of each panel correspond to mean values across all L3L and L3W grid boxes. These means are significantly different using a 2-tailed t-test (unequal variance) with $p < 0.005$ in all cases except ak_diagonal at 300 hPa where $p = 0.13$, vmr_ret_minus_apr at 300 hPa where $p = 0.07$, vmr_ret at 600hPa where $p = 0.30$, vmr_ret at 300hPa where $p = 0.11$. No vmr_apr mean differences are significant. Values in the bottom-right corner of each panel correspond to the Spearman's rank correlation coefficient ($p < 0.005$ in all cases).

[1] Note that for ease of interpretation, the absolute retrieved minus a priori VMR values are plotted, i.e. ignoring whether the result is positive or negative. However, the results hold if using signed values, and a duplicate of Fig. 4 with signed retrieved minus a priori VMR values is included in the Supp. Mat. for reference (SM4).


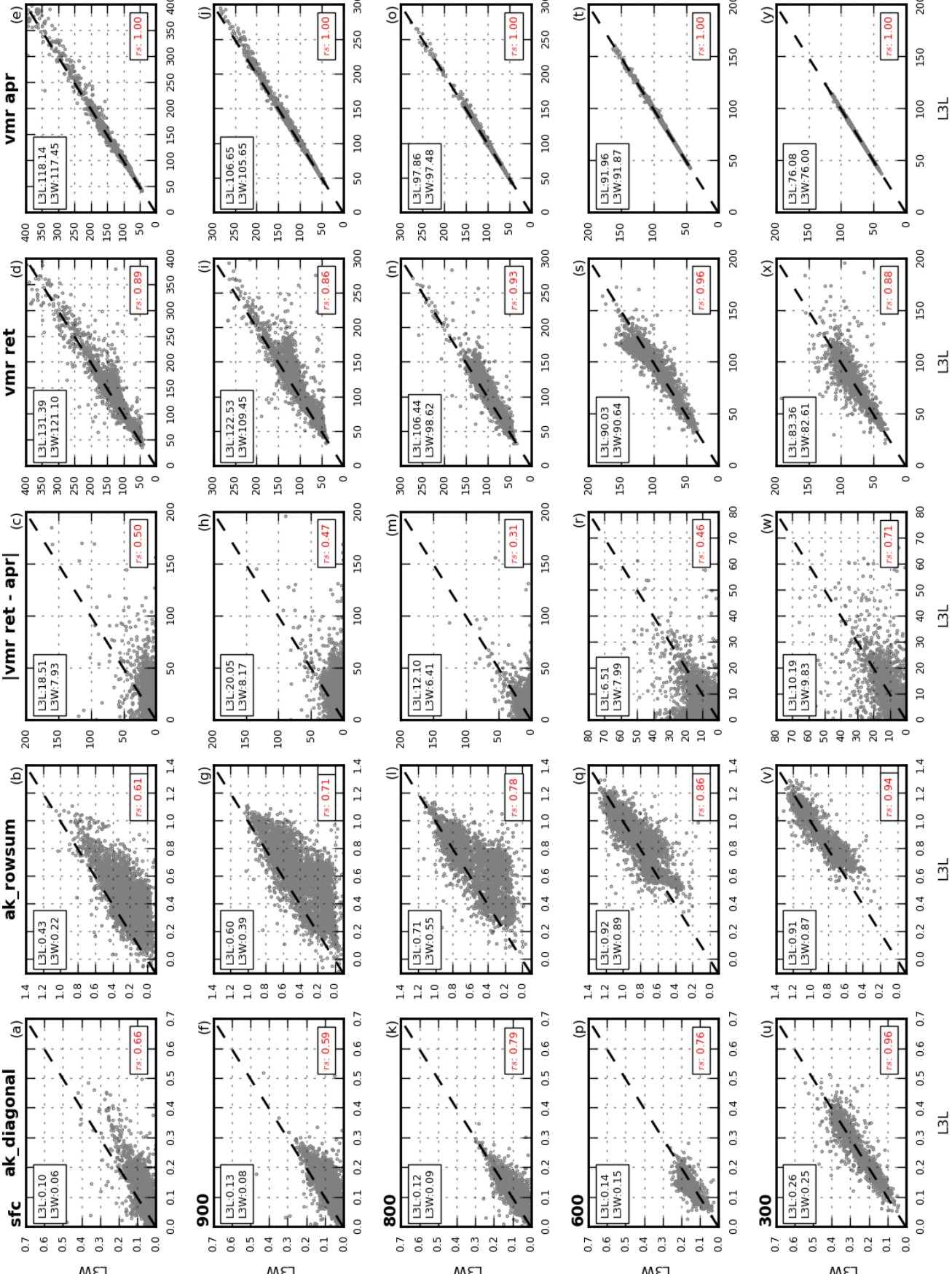

**3.2. Differences in retrieved surface level VMRs and temporal trends, and their relation to the land-water sensitivity contrast**

In this section, retrieved surface level VMRs and their temporal trends in L3L and L3W are compared, and their differences related to the established land-water sensitivity contrast. The effect that averaging together these retrievals has on the statistics and trends in resulting L3O "mixed" (L3O$_M$) data is then evaluated. As with Sect. 3.1.2, the time series analysed in this section only contain days where L3L and L3W are both present and the L3O surface index is mixed.

**3.2.1. L3L vs L3W**

*Retrieved VMR comparison between L3L and L3W*

In addition to the clear land-water sensitivity contrast in coastal grid boxes at the surface, there are clear differences in the retrieved VMRs here (Fig. 4; Fig. 5a (black boxplots)). The retrievals performed over land yield surface level VMRs that are over 10 ppbv greater than over water, on average. As with sensitivity, land-water differences in retrieved VMRs decrease higher up in the profile.

Greater land-water sensitivity differences also tend to be associated with greater retrieved VMR differences. Figure 5b shows the distribution of retrieved surface level VMR differences (L3L – L3W) stratified by the corresponding surface level AK rowsum difference. Larger retrieved VMR differences are clearly associated with greater AK rowsum differences (some degree of spread in the results is expected, since the relationship also depends on the accuracy of the a priori, as outlined previously).

60 % of the coastal grid boxes compared show a significant difference ($p < 0.1$, determined using a 2-tailed student's t-test) in mean VMRs in L3L and L3W (Fig. 5a). Compared to grid boxes where the mean VMR difference is not significant, there are several notable differences (detailed in Table 2). As expected from the previous analysis, the land-water sensitivity contrast is greater when mean VMRs are significantly different ("SIGDIFF$_{L3L-L3W}$") than when not ("NOT_SIGDIFF$_{L3L-L3W}$"). This is evident in AK rowsum and VMR retrieved minus a priori differences (the magnitude of difference between subsets is around 50 % and 100 %, respectively). Interestingly, the AK difference is due to sensitivity being lower over water in SIGDIFF$_{L3L-L3W}$ than in NOT_SIGDIFF$_{L3L-L3W}$; sensitivity over land is similar in both subsets. This may be explained as follows: when sensitivity over water is especially low, as is the case in SIGDIFF$_{L3L-L3W}$, the retrieved VMR will be heavily weighted by the a priori and unable to match the variation present in the more

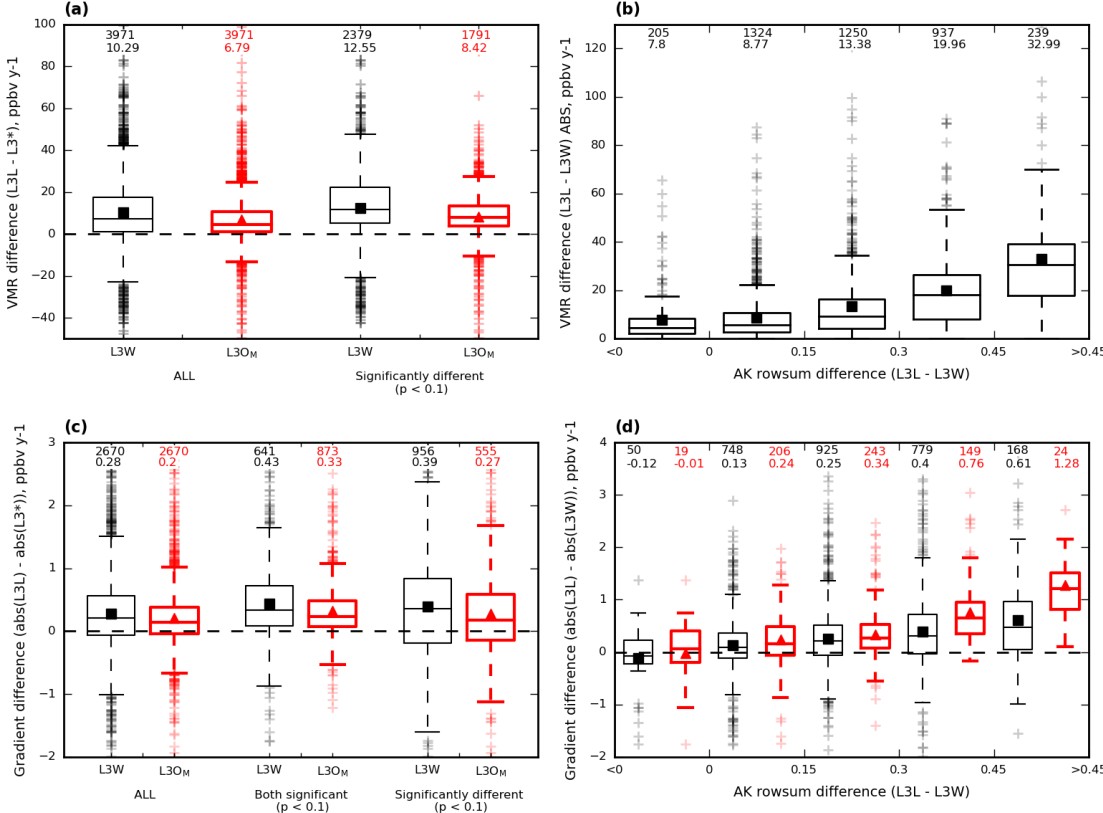

**Figure 5.** Boxplots showing how mean VMRs and trends from WLS analysis compare for coastal L3 grid boxes, calculated from matched retrievals within these grid boxes. "Matched" means that only days when both L3L and L3W are present and the L3O surface index is mixed are used to create the mean values analysed. Mean values are represented by filled squares/triangles, and values above the boxplots correspond to number of grid boxes with data for that boxplot, and the mean value, respectively. **(a)** Mean VMR differences for L3W (black, mean values represented by filled squares) and L3O$_M$ (red, thicker lines, mean values represented by filled triangles) compared to L3L (L3L – L3* in both cases). Shown are the differences for all coastal grid boxes, and only for those grid boxes where the difference is significant (p < 0.1), determined using a 2-tailed t-test. **(b)** Absolute mean VMR differences[1] between L3L and L3W, stratified according to corresponding AK rowsum difference (L3L – L3W in both cases). **(c)** Absolute differences in gradients[2] detected using WLS regression analysis for L3W (black, mean values represented by filled squares) and L3O$_M$ (red, thicker lines, mean values represented by filled triangles), compared to L3L (L3L – L3* in both cases). Shown are differences for all coastal grid boxes where WLS analysis could be performed, for grid boxes where both trends compared are significantly different to zero (p < 0.1), and for grid boxes where the trend difference is significant (p < 0.1). **(d)** Absolute differences in gradients[2] detected using WLS regression analysis between L3L and L3W, stratified according to corresponding AK rowsum difference (L3L – L3W in both cases). Shown are the differences for all coastal grid boxes where WLS could be performed (black, mean values represented by filled squares), and only for those grid boxes where the detected trend is significant (p < 0.1) in both L3L and L3W (red, thicker lines, mean values represented by filled triangles).

[1]Absolute retrieved VMR difference values are shown in Fig. 5b for clarity, since L3L – L3W can be either positive or negative depending on whether a priori VMRs used in the retrieval are greater or less than the "true" VMR being retrieved, which complicates the analysis. The corresponding plot with raw values (i.e. not discarding the +/- sign) is included in the Supp. Mat. however, and the same conclusions can be drawn based on this figure (SM5).

[2] For clarity, differences between the absolute trend values (i.e. ignoring the +/- sign of the trend) are presented, since this shows the degree of difference in the trend magnitude, irrespective of trend direction. A positive trend difference in this case signifies a stronger (faster) trend in L3L than L3* (panel c) or L3W (panel d).

**Table 2.** Mean values for selected variables from L3L and L3W for coastal L3 grid boxes, matched retrievals only. "Matched" means that only days when both L3L and L3W are present and the L3O surface index are mixed are used to create the mean values analysed. Mean values are calculated and presented separately according to the results of a 2-tailed student's t-test (unequal variance) performed on mean retrieved VMR values in L3L and L3W (n = 3971). Mean L3L – L3W differences are also shown for each subset ('L-W').

| | P < 0.1 ("SIGDIFF$_{L3L-L3W}$") (n=2379, 60 %) | | | P > 0.1 ("NOT_SIGDIFF$_{L3L-L3W}$") (n=1592, 40 %) | | |
|---|---|---|---|---|---|---|
| | L3L | L3W | L-W | L3L | L3W | L-W |
| Mean vmr_ret | 129.97 | 117.41 | 12.55 | 133.52 | 126.60 | 6.90 |
| Mean vmr apr | 113.78 | 113.18 | 0.61 | 124.65 | 123.83 | 0.83 |
| Mean ret-apr | 16.18 | 4.24 | 11.94 | 8.87 | 2.77 | 6.09 |
| Mean ak rowsum | 0.43 | 0.18 | 0.24 | 0.44 | 0.27 | 0.16 |

sensitive retrieval over land. As sensitivity over water increases, this a priori weighting weakens and the
retrieved VMR will more closely track the retrieval over land, resulting in a less significant difference. Also
of note, a priori VMRs are much lower in SIGDIFF$_{L3L-L3W}$ than in NOT_SIGDIFF$_{L3L-L3W}$, on average.
Considered alongside the greater retrieved minus a priori differences, this suggests that the a priori VMR
could be a less accurate estimate of the "true" VMR for the SIGDIFF$_{L3L-L3W}$ subset, whereas it is closer to
reality for the NOT_SIGDIFF$_{L3L-L3W}$ subset. Intuitively, this makes sense: for a hypothetical situation where
the a priori VMR is a perfect match for the "true" VMR, and both are uniform across a coastal L3 grid box,
retrievals over the land and water portions of the grid box would be expected to be identical irrespective of
any differences in retrieval sensitivity over those surfaces. To summarise: assuming "true" VMRs are similar
over land and water within coastal L3 grid boxes, differences in retrieved VMRs depend not only on the
sensitivity of the retrieval, but also on the accuracy of a priori VMRs used in the retrievals.
It should be noted that there are additional physical factors that could plausibly play a role in
generating the L3L – L3W retrieved VMR difference that is observed, in addition to retrieval sensitivity.
Given that most CO sources are land-based, a decrease in VMRs from land to water might be expected,

especially near to the surface. However, this assumption only seems reasonable where large CO sources are proximal to the coastline, as it is unrealistic to expect gradients as large as are observed in background CO (which coastal grid boxes far from large CO sources are more likely to represent) across the relatively small distance covered by a L3 grid box. Given the relatively long-lived, well-mixed nature of atmospheric CO, VMRs retrieved at a given location are a function of both local emissions *and* transport, and the portion of coastal L3 grid boxes situated over water therefore do not represent pristine conditions in comparison to the adjacent land-based portion of the grid boxes. This is verified by comparing a priori VMRs (also shown in Fig. 4), which suggest the land-water difference in CO concentrations should be negligible (mean L3L – L3W a priori VMR difference = 0.69 ppbv, compared to a mean retrieved VMR difference of 10.29 ppbv). Indeed, in some specific cases – e.g. uninhabited coastal areas downwind of large trans-oceanic pollution sources – VMRs may be higher over the water portion of coastal gridboxes than the adjacent land portion (note that Fig. 4 does show that this is the case in some grid boxes). The above reasoning can also be applied to the question of whether wind direction is responsible for creating the observed L3L – L3W difference in retrieved VMRs: It could be hypothesised that a prevailing onshore wind may lead to CO concentrations being higher over land than water, yet the negligible L3L – L3W a priori VMR difference, the fact that atmospheric CO is well-mixed, and the clear land-water sensitivity gradient that has been demonstrated suggest that wind direction does not play a big role in creating the land-water difference observed in retrieved VMRs. To further rule out the role of wind direction, the L3L – L3W retrieved VMR comparison has been analysed alongside wind direction for several case study grid boxes, and there appears to be no notable shift in wind direction whether L3L or L3W is greater for a given grid box. Results for this analysis are given in the Supp. Mat. (SM6). The weight of evidence therefore points towards L3L – L3W retrieved VMR differences being a function of reduced retrieval sensitivity over water compared to land.

*Trend comparison between L3L and L3W*

We now compare temporal trends detected in surface level retrievals in L3L and L3W for coastal grid boxes, and relate differences to the land-water sensitivity contrast outlined previously.

On average, across all grid boxes where WLS can be performed in both datasets following the criteria outlined in Sect. 2.5 (n = 2670), trends are stronger in L3L than L3W (Fig. 5c (black boxplots)), with the range of differences around 2.5 ppbv y$^{-1}$ (~-1 ppbv y$^{-1}$ to 1.5 ppbv y$^{-1}$). When the comparison is restricted to grid boxes where both trends are significantly different to zero (p < 0.1; 641 of the 2670 grid boxes, 24 %), a greater proportion of those grid boxes have a stronger trend in L3L than L3W (> 75%), but the overall range of differences doesn't shift by much. The L3L – L3W trend difference is significant in 956 of the 2670

coastal grid boxes for which the analysis can be performed (36 %), with the range in differences spanning around 4 ppbv y$^{-1}$. The trends are negative at 75 % of coastal grid boxes in both datasets, this value increasing to 95% when the trend in both L3L and L3W is significant. Descriptive stats corresponding to the trends values compared are detailed in Table 3).

To determine whether differences in trend can be linked to differences in retrieval sensitivity, L3L – L3W trend are stratified by L3L – L3W surface level AK rowsum differences (Fig. 5d). As with mean VMR differences, the size of the trend difference tends to increase as the difference in AK rowsums increases. In addition, as the magnitude of AK rowsum difference increases in the positive direction (i.e. increasingly greater sensitivity over land), a greater proportion of trend differences are positive (i.e. a stronger trend over land). This pattern is even more pronounced when restricted to grid boxes where both trends are significant (also shown in Fig. 5d).

In summary, these results show a general tendency for trend underestimation in surface level retrievals over water compared to surface level retrievals over land in the same coastal grid boxes obtained at the same times, which appears to be linked to differences in retrieval sensitivity. The relationships found in these analyses are not perfect because trend differences are sensitive to several other factors, in addition to differences in retrieval sensitivity. For example, a greater trend difference would be evident if the rate of change in "true" CO concentrations is faster than if it is slow/negligible, for a given sensitivity difference. Similarly, there should be zero trend difference if "true" CO concentration levels are stable over time, irrespective of the magnitude of difference in retrieval sensitivity. The accuracy of the a priori is a further complicating factor. An underlying assumption is also that the temporal trend in "true" VMRs should not vary much across a 1° x 1° L3 grid box. Hedelius et al. (2021) lends credence to this assumption with the finding that CO trends are similar within regions spanning a few thousand kilometres (L3 grid boxes are ~ 100 km$^2$), and that trends within urban areas are generally indistinguishable from the trend of the broader region encompassing the urban area.

**Table 3.** Descriptive stats corresponding to the WLS trends detected in L3L, L3W, and L3O$_M$ that are compared in the boxplots of Fig. 5c.

| | | | Mean | Std | Median | IQR |
|---|---|---|---|---|---|---|
| All | L3L – L3W (n = 2670) | L3L | -0.55 | 1.27 | -0.47 | 1.00 |
| | | L3W | -0.49 | 1.08 | -0.34 | 0.65 |
| | L3L – L3O$_M$ (n = 2670) | L3L | -0.55 | 1.27 | -0.47 | 1.00 |
| | | L3O$_M$ | -0.51 | 1.03 | -0.38 | 0.73 |
| Both significant (p < 0.1) | L3L – L3W (n = 641) | L3L | -1.39 | 1.66 | -1.15 | 1.08 |
| | | L3W | -1.06 | 1.56 | -0.78 | 0.92 |
| | L3L – L3O$_M$ (n = 873) | L3L | -1.24 | 1.64 | -1.06 | 1.07 |
| | | L3O$_M$ | -1.02 | 1.38 | -0.83 | 0.88 |
| Significantly different (p < 0.1) | L3L – L3W (n = 956) | L3L | -0.64 | 1.39 | -0.65 | 0.92 |
| | | L3W | -0.52 | 1.06 | -0.43 | 0.67 |
| | L3L – L3O$_M$ (n = 555) | L3L | -0.69 | 1.36 | -0.67 | 0.85 |
| | | L3O$_M$ | -0.60 | 1.00 | -0.51 | 0.68 |


**3.2.2. Consequences for L3O data with a surface index of mixed ("L3O$_M$")**

To recap, L3O data are given the surface index "mixed" ("L3O$_M$") when neither land nor water is the
dominant surface type of the bounded L2 retrievals, for a given retrieval time. When this is the case, the
retrievals over land and water are averaged together. Users of L3O data do not have the option of choosing
to only analyse the subset of retrievals made over land (L3L) or water (L3W), as was done in the preceding
analysis. To do so requires the original L2 retrievals. In this section, the L3O$_M$ retrievals are compared to the
L3L retrievals that were analysed in the previous section. The aim here is to demonstrate how, for some L3
grid boxes, information on "true" VMRs and temporal trends that is available in the L2 retrievals over land
(L3L) is effectively lost to users of L3O data by their averaging together with the less sensitive L2 retrievals
over water (L3W).


*Retrieved VMRs in L3O$_M$*

For long-term mean VMRs, L3O$_M$ unsurprisingly represents a mid-point between L3L and L3W, with lower
VMRs than L3L, but a smaller difference range overall than L3W (Fig. 5a, red boxplots). The L3L – L3O$_M$
differences in long-term mean VMR are significant at 45 % (1791) of coastal grid boxes. All but 3 of these
grid boxes also see a significant difference between long-term mean VMRs in L3L and L3W. This makes
sense: retrievals in L3L would not be expected to differ significantly from those in L3O$_M$ if they do not also
differ significantly from L3W. In total, 75 % of grid boxes that feature a significant difference between L3L
and L3W also see a corresponding significant difference between L3L and L3O$_M$. There are several notable
differences between this subset of coastal grid boxes ("BOTH$_{VMRs}$"), compared to those that see a significant
difference between L3L – L3W but not between L3L and L3O$_M$ ("L3L_L3W_ONLY$_{VMRs}$"), detailed in Table
4a:

•  The grid boxes of BOTH$_{VMRs}$ see greater retrieved VMR differences between L3L and L3W than the

grid box subset of L3L_L3W_ONLY$_{VMRs}$ (mean L3L – L3W difference of 13.84 vs 8.67 ppbv). This

is logical: L3O$_M$ only differs significantly from L3L if the underlying L3L – L3W difference is

sufficiently large to persist through averaging.

•  The grid boxes of BOTH$_{VMRs}$ also feature a greater land-water sensitivity contrast than those of

L3L_L3W_ONLY$_{VMRs}$. This is indicated both by L3L – L3W AK rowsum differences, driven

predominantly by decreased sensitivity over water in BOTH$_{VMRs}$; and by L3L – L3W retrieved minus

a priori VMR differences.

•  The grid boxes of BOTH$_{VMRs}$ tend to have a greater proportion of their surface covered by water than

land when compared to L3L_L3W_ONLY$_{VMRs}$. This is determined by analysis of ratio(land/water)

values for each grid box (derivation of this metric is outlined in Sect. 2.4).  A mean ratio(land/water)

of 0.87 for BOTH$_{VMRs}$ indicates a greater water influence on L3O$_M$ than for the grid boxes of

L3L_L3W_ONLY$_{VMRs}$, for which a mean ratio(land/water) of 1.00 indicates a more even land/water

split. Thus, L3O$_M$ more closely resembles L3W – which is significantly different to L3L – in

BOTH$_{VMRs}$ than in L3L_L3W_ONLY$_{VMRs}$.


It is easy to understand how each of these can lead to a L3O$_M$ retrieval that differs significantly from the
corresponding L3L retrieval. Interestingly, it is also notable that retrieved and a priori VMRs are lower in
BOTH$_{VMRs}$ than in L3L_L3W_ONLY$_{VMRs}$, and that retrieved minus a priori VMR values are greater in
BOTH$_{VMRs}$ than in L3L_L3W_ONLY$_{VMRs}$. This could imply that the a priori VMRs are closer to reality (i.e.
the a priori CO amount is closer in value to the actual ("true") CO amount that is being measured) for the
grid boxes of L3L_L3W_ONLY$_{VMRs}$ than those of BOTH$_{VMRs}$, however to properly assess this it would be
necessary to know what the actual "true" VMR values are that are being measured.


**Table 4.(a)** Descriptive stats corresponding to matched retrievals over land and water (L3L and L3W) where the long-term mean retrieved surface level VMR in L3L and L3W is significantly different ($p < 0.1$, n = 2379). Grid boxes are divided into two subsets depending on whether long-term mean VMRs in L3L and L3O$_M$ are significantly different ($p < 0.1$; "BOTH$_{VMRs}$") or not ($p > 0.1$; "L3L_L3W_ONLY$_{VMRs}$"). The metric "ratio(land/water)" indicates the relative land vs water surface coverage of a L3 grid box. A ratio(land/water) value $> 1$ ($< 1$) implies that more of the grid box surface is covered by land (water).

**(b)** Descriptive stats corresponding to matched retrievals over land and water (L3L and L3W) where the temporal trend detected using WLS regression analysis on yearly-mean retrieved surface level VMR in L3L and L3W is significantly different ($p < 0.1$, n = 956). Grid boxes are divided into two subsets depending on whether the trend in L3L is significantly different to the corresponding trend detected in L3O$_M$ ($p < 0.1$; "BOTH$_{TRENDS}$") or not ($p > 0.1$; "L3L_L3W_ONLY$_{TRENDS}$"). The metric "ratio(land/water)" indicates the relative land vs water surface coverage of a L3 grid box. A ratio(land/water) value $> 1$ ($< 1$) implies that more of the grid box surface is covered by land (water).

| (a) | BOTH$_{VMRS}$ (n = 1788, 75 %) | | | L3L_L3W_ONLY$_{VMRS}$ (n = 591, 25 %) | | |
|---|---|---|---|---|---|---|
| **Mean ratio(land/water)** | 0.87 | | | 1.00 | | |
| | **Land** | **Water** | **L-W** | **Land** | **Water** | **L-W** |
| **Mean vmr_ret** | 127.21 | 113.37 | 13.84 | 138.30 | 129.64 | 8.67 |
| **Mean vmr_apr** | 109.11 | 108.62 | 0.49 | 127.94 | 126.96 | 0.98 |
| **Mean ret-apr** | 18.11 | 4.75 | 13.36 | 10.36 | 2.68 | 7.68 |
| **Mean AK rowsum** | 0.42 | 0.16 | 0.26 | 0.46 | 0.26 | 0.20 |
| (b) | BOTH$_{TRENDS}$ (n = 447, 47 %) | | | L3L_L3W_ONLY$_{TRENDS}$ (n = 509, 53 %) | | |
| **Mean ratio(land/water)** | 0.77 | | | 0.99 | | |
| | **Land** | **Water** | **L-W** | **Land** | **Water** | **L-W** |
| **Mean WLS trend** | -0.72 | -0.58 | -0.14 | -0.58 | -0.47 | -0.11 |
| **Mean ABS WLS trend** | 1.18 | 0.76 | 0.42 | 1.04 | 0.68 | 0.35 |
| **Mean trend standard error** | 0.55 | 0.39 | 0.16 | 0.58 | 0.36 | 0.22 |
| **Mean vmr_ret** | 128.25 | 121.36 | 6.90 | 129.22 | 120.20 | 9.02 |
| **Mean vmr_apr** | 117.21 | 117.13 | 0.08 | 116.01 | 115.73 | 0.29 |
| **Mean ret-apr** | 11.05 | 4.22 | 6.82 | 13.21 | 4.47 | 8.74 |
| **Mean AK rowsum** | 0.46 | 0.22 | 0.25 | 0.44 | 0.20 | 0.24 |

 *Trends in L3O$_M$*


Temporal trends detected in L3O$_M$ are now compared to those in L3L (Fig. 5c, red boxplots). Overall, a
greater number of grid boxes feature a significant trend in both L3L and L3O$_M$ than in L3L and L3W (873
vs 641; 33 % vs 24 %), and fewer see a significant difference between trends (555 vs 956; 21 % vs 36 %).
This is to be expected, given that the L2 retrievals contributing to L3L also contribute to L3O$_M$. The trends
in L3L and L3O$_M$ are significantly different in just under half (47 %) of the grid boxes where the trend is also
significantly different between L3L and L3W ("BOTH$_{TRENDS}$"; Table 4b). These grid boxes are clearly more
water-dominated than the remaining 53 % of grid boxes where the trend difference between L3L and L3W
is significant but the L3L – L3O$_M$ difference is not ("L3L_L3W_ONLY$_{TRENDS}$"). This is indicated by a mean
ratio(land/water) of 0.77 for BOTH$_{TRENDS}$ vs 0.99 for L3L_L3W_ONLY$_{TRENDS}$. Additionally, detected trends
in the grid boxes of BOTH$_{TRENDS}$ are slightly stronger, with a greater difference between L3L and L3W, than
for the L3L_L3W_ONLY$_{TRENDS}$ subset. Those L3 grid boxes featuring the strongest land-water trend
difference are therefore most likely to also see a significant trend difference between L3L and L3O$_M$. Again,
this is logical. Unlike with the retrieved VMR comparison above however, there are no clear differences in
mean retrieved or a priori VMRs, nor sensitivity metrics, between these two grid box subsets (also detailed
in Table 4b). However, it is not necessarily expected that there would be clear differences in these parameters
for this analysis, since trend magnitudes themselves are also a variable (i.e. the trend in "true" CO varies
across space, independent of retrieval sensitivity or CO concentration, complicating the relationships outlined
above).

Most of the grid boxes where the L3L and L3O$_M$ trends are significantly different also feature a

significant difference between L3L and L3W (453 of 555; 82 %). There are no clear differences between
these and the remaining 18 % of grid boxes that, counter-intuitively, feature a significant difference between
trends in L3L and L3O$_M$ but not between trends in L3L and L3W. However, small discrepancies are to be
expected for results based on statistical thresholds, especially where the variables being compared are subject
to multiple different factors (e.g. land-water surface cover ratio in L3O$_M$; land-water sensitivity contrast;
retrieved VMR differences; differences in the "true" CO concentration being retrieved and its change over
time).



### 3.3. Implications for users of L3O data

So far, this paper has shown a clear difference in retrieval sensitivity over land and water for coastal grid boxes, demonstrated how long-term VMR statistics and temporal trends calculated using these retrievals (L3L and L3W) differ, and outlined consequences of averaging these retrievals together to create L3O$_M$. The full time series of available data in L3O is now compared with L3L and L3W, without the constraint that a retrieval needs to be present in both L3L and L3W for it to be included in the analysis. This replicates what a user of the L3O data would do, i.e., work with all available data.

Users of MOPITT data are advised to restrict their analysis to retrievals performed over land. This poses a quandary for users of L3O: what to do about days with a surface index of mixed? Therefore, the implications of choosing to include or discard these days are also considered. In the subsequent sections, the following subsets of the full L3O time series for each coastal grid box are analysed: the full L3O time series with no filtering by surface index ("L3O$_{NF}$"); only days with a surface index of land ("L3O$_L$"); and days where the surface index is land or mixed ("L3O$_{LM}$" – i.e., only days with a L3O surface index of water are discarded).

### 3.3.1. Loss of available data

The guideline to only analyse retrievals performed over land results in a huge loss of data for coastal grid boxes when using the L3O dataset. This is quantified by comparing the total number of days with data for analysis at each coastal grid box in L3O$_L$ ("n_days(L3O$_L$)") and L3O$_{NF}$ ("n_days(L3O$_{NF}$)") (Fig. 6a). Strikingly, 35 % of coastal grid boxes (total coastal grid boxes = 4299) have zero days in L3O$_L$, and 67 % have a surface classification of land less than 5 % of the time in L3O (yielding a n_days(L3O$_L$/L3O$_{NF}$) ratio of 0.05 or less in Fig. 6a). Importantly, retrievals over land are made on a large proportion of these filtered days; but they are either discarded altogether or averaged together with retrievals made over water to create L3O$_M$. This point is demonstrated by comparison to the total number of days with data for analysis at coastal grid boxes in L3L ("n_days(L3L)"). In contrast to a mean (median) n_days(L3O$_L$/L3O$_{NF}$) ratio of 0.08 (0.01), a mean (median) n_days(L3L/L3O$_{NF}$) ratio of 0.44 (0.40) demonstrates the stark loss of available data. This is further highlighted by the fact that over half (56%) of coastal grid boxes have at least 25 times more days with retrievals made over land than are available for analysis in the L3O dataset if filtering guidelines are followed (as shown by the ratio n_days(L3L/L3O$_L$) in Fig. 6b (black line)).

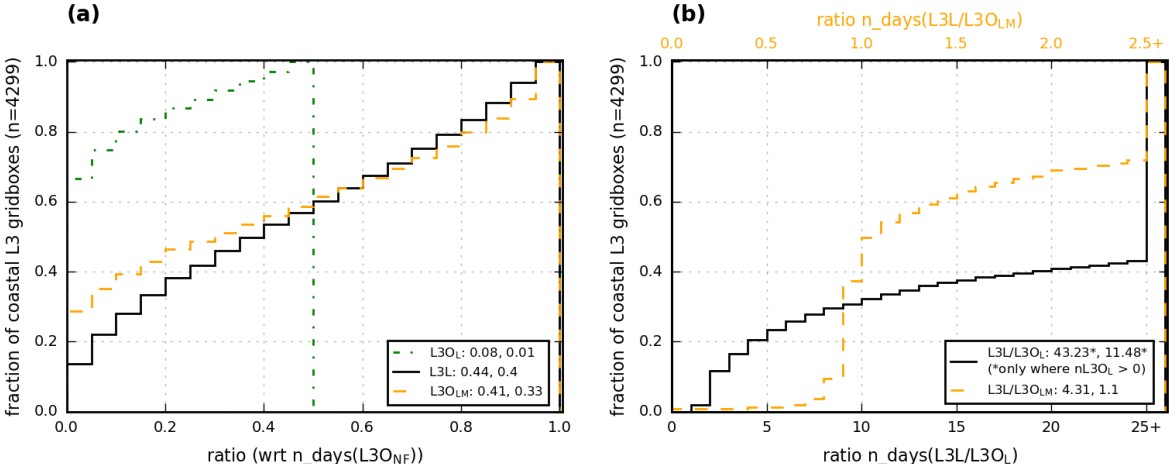

**Figure 6.** Cumulative frequency histograms comparing the number of days with data for different L3Osubsets and L3L at coastal L3 grid boxes. A ratio < 1 (> 1) indicates the plotted dataset has less (more) days with data than the comparison dataset that is indicated on the x-axis. **(a)** L3O$_L$ (dash-dot green line), L3L (solid black line), and L3O$_{LM}$ (dashed orange line) are compared to the "as-downloaded" L3O dataset, without any filtering by surface index ("L3O$_{NF}$"). Values in legend correspond to mean and median ratio for indicated dataset, respectively. Note, as a result of how coastal grid boxes are classified (outlined in Sect. 2.3), all n_days(L3O$_L$/L3O$_{NF}$) ratios are below 0.5 (i.e. at best, L3O has a surface classification of land on 50 % of days). **(b)** L3L is compared with L3O$_L$ (solid black line, bottom x-axis) and L3O$_{LM}$ (dashed orange line, top x-axis). Values in legend correspond to mean and median ratios, respectively.

The situation can be improved for L3O users by keeping days when the L3O surface index is classified
as mixed, in addition to land ("L3O$_{LM}$"). Even in this best-case scenario however, L3O$_{LM}$ sees less days with
data than L3L for over 60% of coastal grid boxes (ratio n_days(L3L/L3O$_{LM}$) in Fig. 6b (orange line)).
Moreover, the large proportion of these L3O$_{LM}$ days have the surface index of mixed and therefore suffer
from the averaging together of retrievals over land with retrievals over water which, as has been shown, can
significantly impact the results of analyses using these data. This point is returned to in following sections.
Intuitively, it is to be expected that the ratio n_days(L3L/L3O$_{LM}$) should *never* be < 1. L2 retrievals
over land obviously contribute to days when L3O is classified as land, and should, by definition, also
contribute to days when L3O is classified as mixed. In these cases, L3L will therefore also be present.
However, there are two instances where L2 retrievals over land in fact do not contribute to a L3O retrieval
classified as mixed. Firstly, L2 retrievals themselves also have a surface classification of mixed, when the
L2 retrieval does not predominantly overlie water or land. L3O can thus have a surface classification of mixed
when created from bounded L2 retrievals that are either only retrieved over a mixed surface, or a combination

of mixed and water: in both cases, there are no L2 retrievals over land, and therefore no L3L. Secondly, analyses performed for this paper identified numerous instances where L3O is classified as mixed, but the only contributing L2 retrievals are retrievals over water. In these instances, L3O therefore seems to be misclassified. On days when this is the case, there will be no corresponding L3L retrieval. This is documented further in the Supp. Mat. (SM7). Attempting to quantify the extent of this misclassification influence is beyond the scope of this paper. In the vast majority of cases where a given grid box has a n_days(L3L/L3O$_{LM}$) ratio < 1, the difference is negligible (i.e. 75 % of these grid boxes have a ratio between 0.9 and 1). Irrespective, in terms of the number of days with retrievals available for analysis, L3L is an improvement over L3O$_{LM}$ for more grid boxes than it is not.

### 3.3.2. Scientific implications

Long-term mean (ltm) retrieved VMR values from the different L3O subsets are compared to L3L for all coastal grid boxes. As expected from the analyses in Sect. 3.2, all L3O subsets that have some influence from L2 retrievals over water have a ltm retrieved VMR that is below that in L3L, on average (Fig. 7a). Unsurprisingly, the closest match to L3L is L3O$_L$ (mean difference -3.1 ppbv), with the mean difference increasing for each L3O subset as the influence of retrievals over water increases (e.g. L3O$_{LM}$ differs less on average from L3L (mean difference = 5.2 ppbv) than L3O$_{NF}$ (mean difference = 9.1 ppbv), which additionally features days when L3O is solely created from L2 retrievals performed over water).

Note that ltm retrieved VMRs in L3O$_L$ and L3L are not a perfect match because L3O$_L$ is only a subset of L3L for each grid box considered in the analysis: L3L may be present on a day when L3O$_L$ is not owing to the way that the L3O data are created (i.e., classified based on the ratio of L2 retrievals over land and water, with retrievals over land potentially being discarded if these are not the majority). Apart from L3O$_L$, less than 25 % of the coastal grid boxes have a retrieved ltm VMR that is greater in an L3O subset than in L3L. The range of ltm differences for each of these L3O subset comparisons to L3L exceeds 35 ppbv (excluding outliers), with over 25 % of coastal grid boxes compared having ltm differences exceeding 9 ppbv (as indicated by boxplot upper quartile values).

The percentage of coastal grid boxes that feature a significant difference between ltm retrieved VMRs in L3L and each L3O subset (indicated in blue above each boxplot) is high: strikingly, it is found that, for the two subsets that L3O users could realistically choose to analyse if following data filtering guidelines (L3O$_L$ or L3O$_{LM}$), almost a quarter (L3O$_L$) or almost half (L3O$_{LM}$) of coastal grid boxes see a significant difference to L3L.

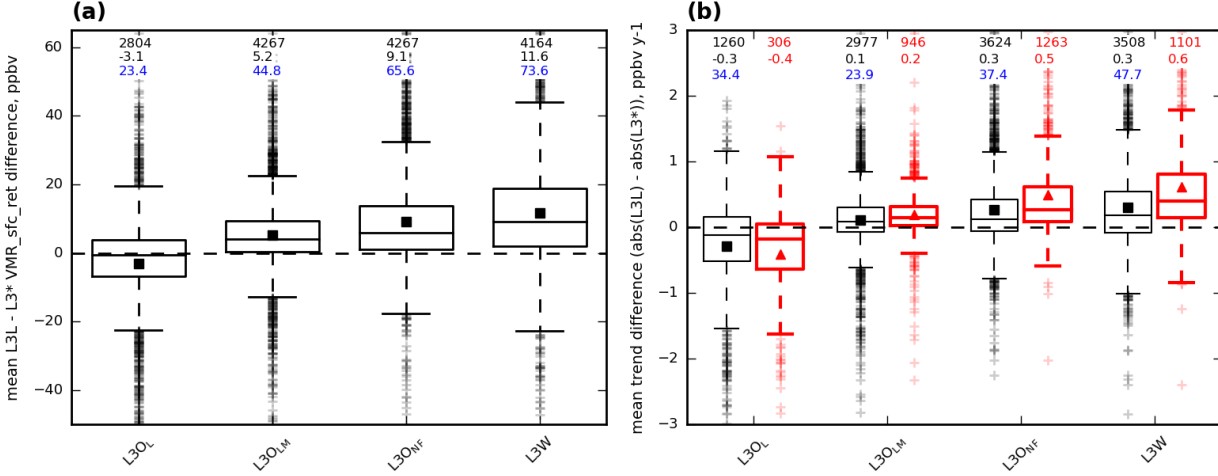

**Figure 7.** Boxplots showing how mean VMRs and trends compare from selected L3O subsets and L3W to L3L. Values compared are calculated from all available data across the study period. Mean values are represented by filled squares, and values above the boxplots correspond to number of grid boxes with data for that boxplot (black, top row), the mean value (black, second row), and the percentage of grid boxes represented in that boxplot that feature a significant difference with L3L (blue, third row), respectively. The comparison is calculated as L3L – L3* in both cases; therefore a point above (below) the black y=0 line indicates that the value being compared is greater (lower) in L3L. **(a)** Mean VMR differences between L3L and the indicated L3O subset or L3W. Note that the n value is different for each boxplot because not all L3 subsets are present at every coastal grid box, as shown in Sect. 3.3.1. **(b)** Differences in gradients (absolute values) detected using WLS regression analysis between L3L and the indicated L3O subset or L3W. Shown are the differences for all coastal grid boxes where WLS could be performed for both datasets compared (black, mean values represented by filled black squares), and only for the sample of those grid boxes where the detected trend is significant (p < 0.1) in both (red, thicker lines, mean values represented by filled triangles).

The results of WLS regression analysis on yearly mean values from each dataset are now compared.
As expected from the earlier analysis, trends are strongest, on average, in L3L and L3O_L – this is especially
so when the comparison is restricted only to trends that are significantly different from zero (p < 0.1) (Table
5). These datasets also have the largest measures of spread, indicating their tendency to yield stronger trends
than the other L3O subsets (and L3W), and these measures lessen for each L3O subset as the influence of
retrievals over water increases. Concomitant with trends decreasing in strength as the influence of retrievals
over water increases in each L3O subset, overall retrieval sensitivity also decreases, as indicated by the mean
averaging kernel metrics shown in Table 5. Comparing the magnitude of trends at each coastal grid box,
significant trends are stronger in L3L for at least 75% of grid boxes for all comparison datasets apart from
L3O_L (Fig. 7b). L3O_L sees stronger trends than L3L on average, but the comparison of these two datasets
needs to be interpreted with caution due to L3O_L being a subset of L3L that features far fewer days with data,
as discussed previously. Like with ltm retrieved VMRs discussed above, the percentage of coastal grid boxes
that feature a significant difference between trends detected in L3L and each L3O subset is high, with over a
third (almost a quarter) of the trends in L3O$_L$ (L3O$_{LM}$) being significantly different to L3L.


Table 5. Descriptive stats corresponding to the WLS trends detected in L3L, L3W, and selected L3O subsets. Also shown are mean averaging kernel rowsums and diagonal values corresponding to the retrievals from which trends are calculated. std = standard deviation, IQR = interquartile range.

| | | L3L | L3O$_L$ | L3O$_{LM}$ | L3O$_{NF}$ | L3W |
|---|---|---|---|---|---|---|
| Calculated from all gridboxes where WLS could be performed | Number of grid boxes | 3624 | 1260 | 2999 | 4288 | 4169 |
| | Mean (std) trend | -0.59 (1.22) | -0.52 (1.38) | -0.50 (0.95) | -0.54 (0.67) | -0.54 (0.66) |
| | Median (IQR) trend | -0.45 (0.89) | -0.46 (1.08) | -0.37 (0.67) | -0.42 (0.53) | -0.40 (0.54) |
| | Mean AK rowsum | 0.45 | 0.45 | 0.33 | 0.28 | 0.22 |
| | Mean AK diagonal value | 0.10 | 0.10 | 0.08 | 0.07 | 0.06 |
| Calculated only from gridboxes where WLS trend is significant (p < 0.1) | Number of grid boxes | 1447 | 453 | 1265 | 2588 | 2499 |
| | Mean (std) trend | -1.23 (1.55) | -1.17 (1.90) | -0.95 (1.18) | -0.79 (0.73) | -0.78 (0.72) |
| | Median (IQR) trend | -0.98 (0.94) | -1.09 (1.28) | -0.74 (0.75) | -0.62 (0.56) | -0.62 (0.57) |
| | Mean AK rowsum | 0.51 | 0.48 | 0.39 | 0.33 | 0.29 |
| | Mean AK diagonal value | 0.11 | 0.10 | 0.08 | 0.07 | 0.06 |



**3.4. Illustrative examples comparing L3O and L3L: analysis of the most populous coastal cities**

In this section, time series from the 33 L3 coastal grid boxes that contain cities classified amongst the 100
most populous in the world (derivation outlined in Sect. 2.5) are analysed to illustrate the differences between
mean values and trends obtained from the L3O and L3L datasets. The comparison is focussed on L3O$_L$ and
L3O$_{LM}$, as these are the L3O subsets that data users would realistically choose to analyse if following the
data filtering guidelines. For clarity, from here these grid boxes are referred to by the name of the city that
they contain. A detailed case study for the L3 grid box containing the city of Dubai is first presented, before
considering results for all cities analysed.


### 3.4.1. Detailed case study: L3 grid box containing Dubai

Summary stats derived from the L3O subsets, L3L, and L3W (included for comparison), for the L3 grid box containing the city of Dubai, are given in Table 6. Figure 8 visualises the daily retrieved VMR time series from L3L, with L3O$_L$ overlaid for comparison purposes.

Of a possible 1620 days with data in the unfiltered L3O dataset for this grid box, a mere 70 days (4 %) remain for analysis when following data filtering guidelines to restrict analysis to retrievals performed over land only (the L3O$_L$ subset). By contrast, there are 1523 days available for analysis using the L3L dataset for this grid box (94 % of total days with retrievals in the L3O dataset). However, in L3O, on most days these retrievals over land are averaged together with retrievals over water to create L3O$_M$, as evidenced by the L3O$_{LM}$ subset containing 1486 days with data for this grid box (92 % of total days in the L3O dataset). That L3L has a greater number of days with data than the L3O$_{LM}$ subset indicates that there are days in L3O with a surface index of water where L2 retrievals were present over land but were discarded because of the L3 creation process.

Long-term mean retrieved VMR is greatest in the land-only datasets L3O$_L$ and L3L. The value in L3O$_L$ is 10 ppbv greater than in L3L. Given that L3O$_L$ is a very small subset of L3L, this appears to be a large overestimate, when compared to L3L. Long-term mean retrieved VMR in L3O$_{LM}$ is 11 ppbv lower than in L3L. This is clearly a result of the inclusion of retrievals over water in this dataset, via L3O$_M$, with long-term mean retrieved VMR in L3W being 17 ppbv lower than L3L. Both the L3L vs L3O$_{LM}$ and L3L vs L3W mean differences are significant ($p < 0.1$). Consistent with the results shown in Sect. 3.2.2 when identifying factors that determine whether the averaging of L2 retrievals over land and water to create L3O$_M$ can yield statistically significantly different retrievals to L3L, this L3 grid box is water-dominated, with a mean ratio(land/water) of 0.60. It is also notable that the standard deviation of long-term mean retrieved VMR in L3L (and L3O$_L$) is roughly twice as large as that in L3O$_{LM}$ and L3W, which is to be expected given that retrievals over water are more greatly tied to their a priori than retrievals over land due to their comparatively lower sensitivity (as discussed in Sect. 3.2.1).

The trends detected using WLS analysis following the method outlined in Section 2.5 are visualised in Figure 9 (note that trend values are also given in Table 6 in both ppbv y$^{-1}$ and % y$^{-1}$), along with the yearly mean VMR values that were used in the regression. Detected trends are clearly strongest in the land-only datasets L3O$_L$ and L3L, with the L3O$_L$ trend being significantly stronger ($p < 0.1$) than the L3L trend – a difference of equating almost 1 % y$^{-1}$ (2.01 ppbv y$^{-1}$). Again, given the far superior temporal coverage of L3L, this is the more reliable result. The trend in L3L is 0.65 % y$^{-1}$ (1.28 ppbv y$^{-1}$) stronger than in L3O$_{LM}$, which corresponds to a difference of almost 12 % over the 18-year period of analysis. The trend in L3O$_{LM}$ is

clearly weakened by inclusion of retrievals over water, with the trend in L3W being over 1 % y$^{-1}$ weaker than
in L3L. Note that this trend analysis has been repeated using an alternative regression method which is less
sensitive to outlying values (Theil-Sen slope estimator), and the results are unchanged. This is detailed further
in the Supp. Mat. (SM8).

**Table 6**. Summary stats from L3O subsets compared, L3L, and L3W (for comparison), for the L3 grid box containing the city of Dubai. Note that across the whole study period (2001-09-01 to 2018-12-31), there are 5988 MOPITT files available. There are 1620 days with data in the L3O dataset (unfiltered by surface index), 27 % of the whole study period. The WLS trend in units of % y$^{-1}$ is calculated by dividing the trend in units of ppbv y$^{-1}$ by the respective long-term mean VMR value.

| Dataset | n days with data (% of days in L3O (n = 1620)) | Long-term mean VMR (± standard deviation) (*ppbv*) | WLS trend (± standard error) (*ppbv y$^{-1}$*) | WLS trend (± standard error) (*% y$^{-1}$*) |
|---|---|---|---|---|
| L3O$_L$ | 70 (4 %) | 190 (± 56) | -4.91 (± 1.21) | -2.59 (± 0.64) |
| L3O$_{LM}$ | 1486 (92 %) | 169 (± 25) | -1.62 (± 0.18) | -0.96 (± 0.10) |
| L3L | 1523 (94 %) | 180 (± 44) | -2.90 (± 0.26) | -1.61 (± 0.14) |
| L3W | 1565 (97 %) | 163 (± 18) | -0.90 (± 0.13) | -0.55 (± 0.08) |

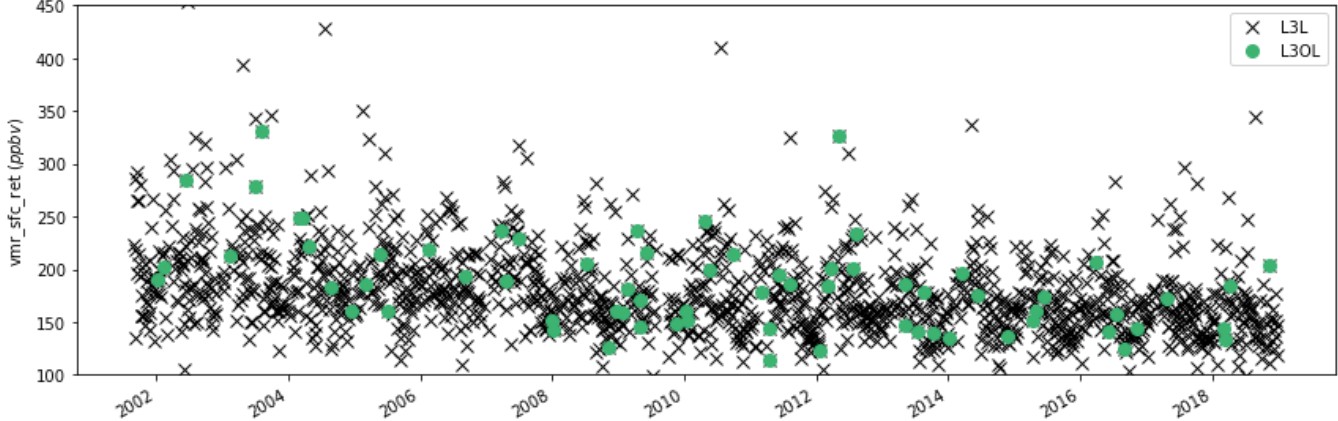

**Figure 8**. L3L (black crosses) and L3O$_L$ (green circles) time series for the entire study period. Note that the size of plotted symbol required to visualise the whole time series artificially exaggerates the sense of temporal coverage; in reality, L3L is only present on 25 % of the days across the study period, and L3O$_L$ just 1 %.

To summarise: If L3O users follow data filtering guidelines and restrict analysis to retrievals only
performed over land, there is a huge loss of data coverage in the L3O dataset for the coastal L3 grid box
containing the city of Dubai. Choosing to work with L3O$_L$ despite this would lead to results that are clearly
erroneous, when compared to L3L, which has far greater temporal coverage (almost 22 times more days with
data than L3O$_L$). L3O users could make the decision to include days with a L3 surface classification of
"mixed" in their analysis to increase temporal coverage (the L3O$_{LM}$ dataset analysed here). However, doing
so would yield both lower retrieved VMRs, on average, and significantly weaker decreasing trends, than
L3L. This is demonstrably due to the incorporation of retrievals over water into L3O$_{LM}$ (via L3O$_M$), as shown
by the comparison with L3W.

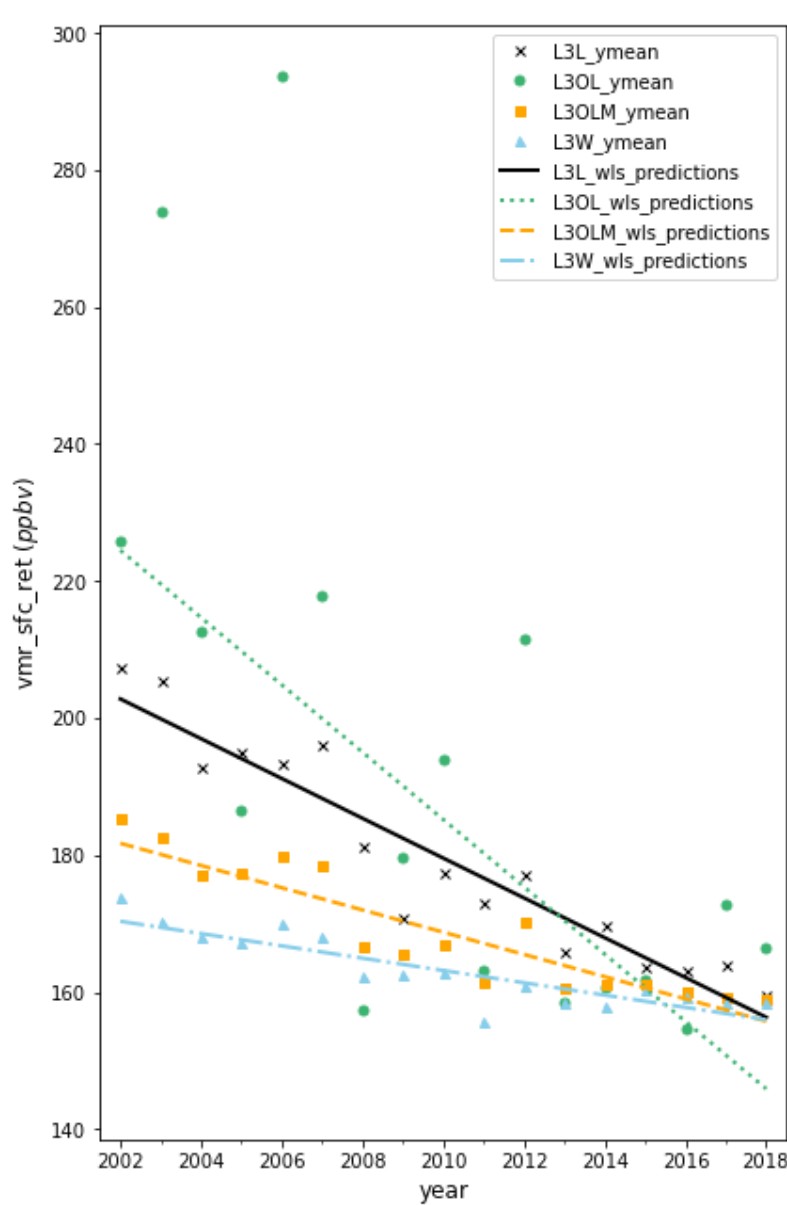

**Figure 9.** Yearly mean ("ymean" in legend) retrieved VMR in the different datasets being investigated, and the trendlines obtained from WLS regression analyses on each of these datasets ("wls_predictions" in legend). Black crosses and solid black lines correspond to L3L; green filled circles and dotted green lines correspond to L3O$_L$; orange filled squares and dashed orange lines correspond to L3O$_{LM}$; blue filled triangles and dash-dot blue lines correspond to L3W. Trend values for each dataset are also given in Table 6.

## 3.4.2. Discussion of results for all cities analysed

The above analysis is repeated for all 33 cities. Number of days with data, long-term mean retrieved VMRs, and temporal trends are given in Table 7 for the L3 grid boxes containing these cities for each of the L3O subsets considered, L3L, and L3W (for comparison). These metrics are evaluated in turn below.

*Temporal coverage*

The loss of data in L3O if filtering for retrievals over land only (L3$O_L$) is clear: 6 of the cities cannot be studied at all using L3$O_L$ (number of days with data = 0), and of the remaining 27 cities with data in this L3O subset, only a single city (Osaka) has more than 50 % of the days with data in L3L. The mean n_days(L3$O_L$/L3L) ratio for these 27 cities is 0.18 – i.e., on average, there are over 5 times more days with data in L3L than are available in L3O when filtering for retrievals over land only.

L3$O_{LM}$ compares more favourably to L3L in terms of number of days with data, due to the inclusion of days when the L3O surface index is "mixed", with a mean n_days(L3$O_{LM}$/L3L) ratio of 0.85. n_days(L3$O_{LM}$) > n_days(L3L) for 11 of the 33 cities, although the difference is generally small. L3$O_M$ is the dominant component of L3$O_{LM}$ in all cases here, being the classification on 84 % of days, on average, across all 33 cities (max = 100 %, min = 45 %)).

*VMR comparison*

The consequence of the loss of data in L3$O_L$ is clear: compared to L3L, mean VMR in L3$O_L$ is higher, and the magnitude of this difference generally depends upon how much data is lost in L3$O_L$. Mean VMR across all cities (excluding the 6 cities where n_days(L3$O_L$) = 0) is 17 ppbv higher in L3$O_L$ than in L3L. This falls to 10 ppbv if restricted to cities where the n_days(L3$O_L$/L3L) ratio is greater than 0.05 (n=17), and 7 ppbv if restricted to cities where the n_days(L3$O_L$/L3L) ratio is above 0.2 (n=11). The mean VMR difference (L3L – L3$O_L$) is significant ($p < 0.1$) for 11 of the 27 cities that can be compared; in these cases, L3$O_L$ is a smaller subset of L3L than for the cities where mean VMR difference is not significant (n_days(L3$O_L$/L3L) = 0.15 vs 0.22, respectively), and the mean VMR difference is unsurprisingly much greater (-36 vs -4 ppbv).

**Table 7.** Summary stats for the L3 grid boxes containing the 33 cities of interest from each of the L3O subsets considered, L3L, and L3W (for comparison). For each grid box and dataset, the following stats are shown: 1. ratio(land/water), which is an indicator of the relative land vs water surface coverage of a L3 grid box; 2. the number of days with data across the whole study period; 3. the mean retrieved VMR (± the standard deviation), in ppbv; and 4. the trend from WLS regression analysis (± the standard error), in ppbv y$^{-1}$. Dash symbols ('-') indicate that the stat cannot be calculated for a given grid box and dataset owing to lack of data. Bold text indicates that a dataset mean or trend value is significantly different to the value in L3L for that city (p < 0.1). Italicised text indicates that the trend value is not significantly different to zero (p < 0.1).  Bold italics indicate that the trend value is not significantly different to zero AND that it is significantly different to the trend in L3L for that city.

[1] The modified mean, shown in the bottom row of the table, corresponds to the mean value that is calculated only for cities where is a corresponding stat in the $L3O_L$ dataset. For 1-3, this corresponds only to cities where number of days with data $L3O_L > 0$ (n = 27). For 4, this corresponds only to cities where there are enough days with data for the regression analysis to be performed in $L3O_L$ (n = 18). By contrast, the mean value, shown in the penultimate table row, simply represents the mean of all values in that column

| city | 1. ratio (land/water) | 2. number of days with data | | | | 3. mean (±std) [ppbv] | | | | 4. trend (± standard error) [ppbv y-1] | | | |
|---|---|---|---|---|---|---|---|---|---|---|---|---|---|
| | | L3L | L3O$_L$ | L3O$_{LM}$ | L3W | L3L | L3O$_L$ | L3O$_{LM}$ | L3W | L3L | L3O$_L$ | L3O$_{LM}$ | L3W |
| Tokyo | 1.57 | 620 | 98 | 627 | 575 | 185 (43) | 188 (38) | 184 (36) | **178 (34)** | -1.7 (0.3) | -2.3 (0.5) | -1.7 (0.3) | -1.7 (0.3) |
| Shanghai | 1.35 | 378 | 54 | 374 | 416 | 373 (130) | 374 (112) | 363 (111) | **338 (108)** | -5.9 (1.4) | -7.0 (1.6) | -5.7 (1.4) | -3.4 (1.2) |
| Manila | 0.05 | 127 | 0 | 86 | 811 | 150 (28) | - | 151 (19) | **145 (22)** | -1.3 (0.5) | - | -1.2 (0.4) | -1.3 (0.2) |
| Mumbai | 0.12 | 790 | 1 | 388 | 1356 | 227 (166) | **291 (0)** | 218 (56) | **184 (66)** | *-1.2 (0.9)* | - | *-0.6 (0.6)* | *-0.1 (0.3)* |
| New York | 0.07 | 216 | 0 | 178 | 919 | 296 (69) | - | 315 (59) | 332 (64) | -1.4 (1.1) | - | **-2.3 (0.8)** | -1.9 (0.5) |
| Lagos | 0.13 | 116 | 4 | 92 | 660 | 337 (109) | 312 (75) | 305 (67) | 232 (69) | *1.2 (2.0)* | - | *0.5 (1.7)* | *0.2 (0.4)* |
| Bangkok | 0.52 | 445 | 33 | 415 | 755 | 314 (77) | **346 (78)** | 308 (62) | **261 (79)** | -3.0 (0.6) | **-8.6 (2.1)** | -3.1 (0.7) | -2.0 (0.4) |
| Osaka | 2.08 | 297 | 171 | 309 | 270 | 187 (48) | 189 (39) | 183 (39) | 172 (34) | -2.5 (0.5) | -2.3 (0.5) | -2.3 (0.4) | **-1.3 (0.4)** |
| Karachi | 1.83 | 1108 | 423 | 1117 | 884 | 139 (33) | **130 (32)** | 136 (30) | **131 (30)** | -0.8 (0.2) | -0.6 (0.2) | -0.7 (0.2) | -0.5 (0.3) |
| Buenos Aires | 3.05 | 864 | 241 | 863 | 719 | 94 (18) | 95 (17) | 94 (16) | 95 (16) | *-0.1 (0.1)* | **-0.5 (0.2)** | *-0.2 (0.1)* | *-0.1 (0.1)* |
| Istanbul | 0.11 | 322 | 2 | 436 | 998 | 152 (30) | **185 (25)** | 154 (19) | **157 (21)** | -1.2 (0.4) | - | ***-0.4 (0.2)*** | -0.8 (0.2) |
| Chennai | 0.08 | 331 | 0 | 95 | 1133 | 223 (56) | - | **205 (25)** | 203 (28) | *0.0 (0.8)* | - | *0.5 (0.5)* | **-0.9 (0.3)** |
| Xiamen | 0.08 | 215 | 1 | 97 | 854 | 263 (74) | **402 (0)** | 258 (69) | 232 (67) | -2.6 (0.9) | - | -4.1 (1.7) | -1.9 (0.4) |
| Taipei | 0.01 | 36 | 0 | 5 | 758 | 192 (50) | - | 210 (26) | 183 (43) | -3.7 (1.0) | - | - | **-1.5 (0.4)** |
| Kuala Lumpur | 0.95 | 142 | 60 | 143 | 200 | 233 (81) | 239 (109) | 234 (84) | 238 (97) | -2.7 (1.3) | -3.4 (1.2) | -3.9 (1.0) | -5.1 (1.1) |
| Saigon | 1.50 | 249 | 122 | 255 | 325 | 254 (65) | **267 (62)** | **244 (60)** | **189 (51)** | *-1.4 (0.9)* | **-3.6 (1.3)** | **-2.3 (0.8)** | **-2.3 (0.8)** |
| Luanda | 0.67 | 173 | 54 | 175 | 341 | 260 (101) | **312 (100)** | 268 (101) | 213 (109) | *-0.5 (2.1)* | *-2.6 (3.7)* | *0.5 (2.2)* | *-0.2 (1.0)* |
| San Francisco | 0.23 | 522 | 15 | 598 | 889 | 236 (92) | 237 (67) | 243 (53) | 250 (60) | *-1.1 (0.7)* | - | *-0.7 (0.5)* | **-1.0 (0.6)** |
| Singapore | 0.05 | 32 | 0 | 18 | 425 | 387 (248) | - | 387 (117) | 341 (133) | - | - | - | -4.3 (2.4) |
| Shantou | 1.79 | 396 | 175 | 398 | 457 | 312 (96) | 326 (104) | 304 (91) | **264 (80)** | -5.4 (0.5) | -5.9 (1.4) | -5.7 (0.4) | **-3.8 (0.7)** |
| Hong Kong | 0.14 | 228 | 3 | 164 | 704 | 336 (83) | **432 (70)** | **312 (71)** | **260 (93)** | -8.1 (0.9) | - | **-5.1 (1.3)** | **-3.5 (0.5)** |
| Toronto | 2.85 | 401 | 186 | 416 | 274 | 238 (58) | 232 (50) | 239 (47) | **254 (44)** | *-1.1 (0.8)* | *-0.3 (1.1)* | *-1.2 (0.7)* | **-2.0 (0.6)** |
| Miami | 0.35 | 411 | 32 | 357 | 1038 | 161 (32) | 157 (26) | 160 (25) | **143 (25)** | -1.5 (0.4) | ***-1.2 (1.2)*** | -1.3 (0.3) | **-0.8 (0.2)** |
| Surat | 1.68 | 943 | 289 | 940 | 760 | 181 (44) | **175 (43)** | 182 (43) | 179 (54) | *-0.4 (0.3)* | **-1.6 (0.7)** | *-0.4 (0.3)* | *-0.1 (0.3)* |
| Dar Es Salaam | 0.01 | 44 | 0 | 17 | 1040 | 103 (46) | - | **86 (12)** | 86 (17) | *-0.3 (0.7)* | - | - | **-0.2 (0.1)** |
| Qingdao | 2.35 | 587 | 186 | 589 | 566 | 372 (102) | 365 (96) | 370 (94) | **383 (111)** | -3.8 (1.5) | ***-2.0 (1.7)*** | -3.7 (1.4) | -4.2 (0.9) |
| Yangon | 0.41 | 590 | 6 | 498 | 930 | 271 (70) | **236 (37)** | **281 (66)** | 266 (79) | -1.5 (0.8) | - | -1.7 (0.6) | -2.1 (0.5) |
| Abidjan | 0.48 | 86 | 38 | 83 | 349 | 218 (58) | 232 (59) | 215 (58) | **156 (44)** | *-2.1 (1.4)* | **-3.6 (1.8)** | *-0.3 (1.4)* | **-0.7 (0.3)** |
| Wenzhou | 0.56 | 386 | 25 | 347 | 705 | 268 (75) | 308 (122) | **256 (65)** | 231 (64) | -4.2 (0.7) | **-10.1 (2.4)** | -3.5 (0.6) | -2.8 (0.6) |
| Sydney | 0.38 | 709 | 6 | 676 | 1000 | 94 (36) | 92 (17) | **90 (16)** | **87 (15)** | -0.7 (0.2) | - | -0.5 (0.1) | **-0.2 (0.1)** |
| Accra | 0.17 | 155 | 7 | 116 | 740 | 245 (84) | 262 (68) | **224 (63)** | **161 (48)** | 2.9 (1.5) | - | 2.9 (0.8) | **-0.5 (0.3)** |
| Dubai | 0.60 | 1523 | 70 | 1486 | 1565 | 180 (44) | 190 (56) | **169 (25)** | **163 (18)** | -2.9 (0.3) | **-5.0 (1.2)** | **-1.6 (0.2)** | **-0.9 (0.1)** |
| Chittagong | 0.81 | 653 | 49 | 628 | 888 | 296 (66) | **316 (79)** | **304 (66)** | 296 (91) | *-0.7 (0.5)* | *-2.1 (2.2)* | *-0.7 (0.5)* | *-0.9 (0.7)* |
| Mean | 0.82 | 427 | 71 | 394 | 736 | 236 | 255 | 232 | 212 | -1.9 | -3.5 | -1.7 | -1.6 |
| Modified mean[1] | 0.99 | 493 | 87 | 466 | 712 | 238 | 255 | 233 | 212 | -2.3 | -3.5 | -2.1 | -1.8 |

842

The L3L – L3O$_{LM}$ mean VMR difference is relatively small, by comparison (4 ppbv, all 33 cities). However, this does hide some much larger discrepancies between L3L and L3O$_{LM}$ for certain cities, with the difference exceeding 10 ppbv in 11 cases and 20 ppbv for 3 of them. The difference is significant ($p < 0.1$; "SIGDIFF$_{L3L-L3OLM}$") for 13 of 33 cities (39 %). Compared to the subset where the L3L – L3O$_{LM}$ mean difference is not significant (n = 20, 61 %; "NOT_SIGDIFF$_{L3L-L3OLM}$"), the following characteristic differences are found (also detailed in Table 8):

- The grid boxes in SIGDIFF$_{L3L-L3OLM}$ have a greater proportion of their surface covered by water than NOT_SIGDIFF$_{L3L-L3OLM}$: this is evidenced by a mean ratio(land/water) of 0.51 in SIGDIFF$_{L3L-L3OLM}$ vs 1.02 in NOT_SIGDIFF$_{L3L-L3OLM}$, indicating there are relatively more retrievals over water than land in the former; and also by the fact that on average, L3O$_L$ only contributes to L3O$_{LM}$ in SIGDIFF$_{L3L-L3OLM}$ on 9 % of days, vs 20 % of days for NOT_SIGDIFF$_{L3L-L3OLM}$ (which means that retrievals over water contribute via L3O$_M$ more frequently to L3O$_{LM}$ in SIGDIFF$_{L3L-L3OLM}$ than NOT_SIGDIFF$_{L3L-L3OLM}$).
- The L3L – L3W VMR ret differences are larger in SIGDIFF$_{L3L-L3OLM}$ than NOT_SIGDIFF$_{L3L-L3OLM}$ (mean = 31.15 vs 18.44 ppbv), meaning they are less likely to be hidden by averaging to create L3O$_M$.
- Land-water mean averaging kernel differences suggest there is not a large land-water sensitivity contrast between the SIGDIFF$_{L3L-L3OLM}$ and NOT_SIGDIFF$_{L3L-L3OLM}$ subsets. However, the L3L – L3W ret-apr difference, which is another indicator of sensitivity difference, is much greater for SIGDIFF$_{L3L-L3OLM}$ than NOT_SIGDIFF$_{L3L-L3OLM}$ (21.66 vs 3.22 ppbv, respectively (21.98 vs 11.88 ppbv if using absolute values)). There is some evidence that this may be a function of the a priori VMRs being closer to "true" VMRs in NOT_SIGDIFF$_{L3L-L3OLM}$, with mean retrieved minus a priori VMR values being closer to zero than in SIGDIFF$_{L3L-L3OLM}$.

These findings are all consistent with what was shown in Sect. 3.2.2 when identifying factors that determine whether the averaging of L2 retrievals over land and water to create L3O$_M$ can yield a statistically significantly different retrieval to L3L. As outlined above, L3O$_M$ is the dominant component of L3O$_{LM}$ in all cases considered here (being the classification on 84 % of days, on average (max = 100 %, min = 45 %)).

**Table 8.** Selected parameters from L3 grid boxes containing cities, stratified according to whether mean VMR in L3L and L3O$_{LM}$ is significantly different ("SIGDIFF$_{L3L-L3OLM}$"; p < 0.1) or not ("NOT_SIGDIFF$_{L3L-L3OLM}$").

| | **P < 0.1**<br>**("SIGDIFF$_{L3L-L3OLM}$")**<br>**(n = 13)** | **P > 0.1**<br>**("NOT_SIGDIFF$_{L3L-L3OLM}$")**<br>**(n = 20)** |
|---|---|---|
| Mean ratio(land/water) | 0.51 | 1.02 |
| % days from L3O$_L$ | 9 | 20 |
| Δ VMR ret (L3L – L3W) (*ppbv*) | 31.15 | 18.44 |
| Δ AK rowsum (L3L – L3W) | 0.25 | 0.21 |
| Δ AK diagonal (L3L – L3W) | 0.10 | 0.08 |
| Δ VMR (ret - apr) (L3L – L3W) (*ppbv*) | 21.66 | 3.22 |
| \|Δ VMR (ret - apr)\| (L3L – L3W) (*ppbv*) | 21.98 | 11.88 |
| L3L VMR (ret - apr) | -19.82 | -7.07 |
| \|L3L VMR (ret - apr)\| | 39.86 | 18.79 |
| L3W VMR (ret - apr) | -14.75 | -6.73 |
| \|L3W VMR (ret - apr)\| | 18.21 | 15.57 |

875

876

*Trend comparison*

878

On average, the strongest trends are seen in L3O$_L$. However, as with the Dubai case study, this often appears as an outlier compared to the other datasets – a consequence of its comparatively very sparse temporal coverage. As expected from previous sections, the weakest trends are detected in L3W, with L3O$_{LM}$ representing a mid-point between this and L3L.

Of the 18 cities where WLS analysis can be performed in L3O$_L$, there are 9 where the resulting trend – and thus conclusion drawn from the analysis – is significantly different to that in L3L. In 3 of these cases (Dubai, Wenzhou, Bangkok), the trend in L3O$_L$ can be judged to be a strong over-estimate given the large difference to the corresponding trends in L3L (trend standard errors do not overlap), and the very small number of days with data that these trends are based on when compared to L3L (n_days(L3O$_L$/L3L) ratio < 0.08 in each case). There are 4 additional cities where a significant trend in L3O$_L$ appears to be an over-estimate, when compared the L3L: Abidjan, Surat, Saigon, and Buenos Aires. This is because the trend for these cities in L3L is not significantly different to 0 which, given the higher number of days with data in L3L (n_days(L3O$_L$/L3L) ratio = 0.44, 0.31, 0.49, 0.28, respectively), appears to be the more reliable result. The

L3O$_L$ trend for Miami is insignificant and derived from very low n. L3O$_L$ is also the only dataset to yield an
insignificant trend for Qingdao.
As with mean VMRs, trends in L3O$_{LM}$ compare better than L3O$_L$ to L3L. However, there are still 5
cases where L3O$_{LM}$ and L3L yield significantly different results. For 3 of these (Hong Kong, Istanbul, and
Dubai, as covered in detail in Sect. 3.4.1), interpretation of the difference is simple: L3O$_{LM}$ is a significant
under-estimate of the CO change over time. This is very likely due to the inclusion of retrievals over water
in this dataset, as evidenced by L3W yielding a significantly weaker trend than L3L in all 3 cases. In the
remaining 2 cases – New York and Saigon – interpretation is more complicated. For both these cities, the
trend detected in L3L is not significantly different from zero, whereas the trend in L3O$_{LM}$ is. Does this mean
that the trend in L3O$_{LM}$ is an over-estimate? Possibly. However, in both cases, the trends are within one
standard error of eachother and therefore within the range of sampling uncertainty. There are an additional 2
cities where WLS could be performed in L3L but not L3O$_{LM}$ (Dar Es Salaam and Taipei), but n_days(L3L)
is so low (44 and 36, respectively) that these results are not deemed to be trustworthy.
As outlined in Sect. 2.5, it is important to note that the trends presented in this section are for
illustrative purposes only, with the intention of demonstrating that different results can be obtained depending
on whether L3O or L3L (and, by extension, L2) data are analysed. More focused analysis is needed to verify
these trends, which is beyond the scope of this paper. The trend analysis has been repeated using an alternative
regression method which is less sensitive to outlying values (Theil-Sen slope estimator), and the main results
reported above stand. This is detailed further in the Supp. Mat. (SM8).


**4. Summary and Conclusions**

The aim of this paper was to compare surface level retrievals and their temporal trends in "as-downloaded"
L3 data ("L3O") with those that could be obtained if only the L2 retrievals performed over land are averaged
to create the L3 product ("L3L"), for all coastal L3 MOPITT grid boxes around the globe (n = 4299). This
work is motivated by a conflict between the recommendation that MOPITT data users restrict analyses to
retrievals performed over land owing to known sensitivity issues over water (MOPITT Algorithm
Development Team, 2018; Deeter et al., 2015), and the reality that L3O data are created from L2 retrievals
performed over both land and water for coastal L3 grid boxes, limiting the ability of L3 data users to follow
the recommendation in these cases. In short, this study has sought to answer the question: "does it matter"?
Analysis has focussed on comparing the original, "as-downloaded" L3 dataset ("L3O") with new land-only

and water-only L3 products ("L3L" and "L3W" respectively) that have been created from the L2 retrievals. The main results are summarised below.

First, a direct comparison of the L2 retrievals performed over land (L3L) and water (L3W) that are averaged together to create L3 products on days when the L3 surface index is "mixed" (L3O$_M$) identified that:

- Retrieval information content is clearly greater in L3L than L3W. The corresponding mean L3L – L3W VMR difference is over 10 ppbv, significant ($p < 0.1$) at 60 % of the coastal grid boxes compared.

- Temporal trends are also stronger, on average, in L3L (mean diff = 0.28 ppbv y$^{-1}$, 0.43 ppbv y$^{-1}$ if only considering trends significantly different to zero), with the L3L – L3W trend difference significant ($p < 0.1$) at 36 % of grid boxes where a trend comparison was possible.

- Larger L3L – L3W differences in mean VMRs and trends are clearly associated with greater differences in retrieval sensitivity.

- The resulting VMRs in L3O$_M$ are significantly different to L3L for 75 % of grid boxes where the L3L – L3W difference is also significant; this corresponds to 45 % of all coastal grid boxes compared. Whether or not L3O$_M$ and L3L differ significantly depends on multiple factors including the ratio of land/water surface cover in the grid box, the strength of the land-water sensitivity contrast and VMR difference, and, potentially, the accuracy of the a priori.

- Just under half of the grid boxes that featured a significant L3L – L3W trend difference also see trends differing significantly between L3L and L3O$_M$. As with the mean VMR comparison, these grid boxes are more water-dominated than the subset whereby the L3L – L3W trend difference is significant but the L3L – L3O$_M$ trend difference is not. They also feature stronger L3L – L3W trend differences overall, but no other variables (such as ltm VMRs and sensitivity metrics) show clear differences.

Having established the degree of difference in L3O$_M$ and L3L retrievals that is caused directly by averaging L3L with the less-sensitive L3W, the full L3O dataset with differing surface filtering options was compared to L3L:

- If L3O is filtered so that only retrievals over land (L3O$_L$) are analysed, as has been recommended (MOPITT Algorithm Development Team, 2018; Deeter et al., 2015), there is a huge loss of data, in terms of days with data to analyse. This is a direct result of L2 retrievals over land routinely being discarded during the L3O creation process, or averaged with L2 retrievals over water, creating L3O$_M$

(at least for coastal grid boxes). The problem can be alleviated by also retaining L3O$_M$ retrievals, but these additional days with data feature some influence from retrievals made over water that can affect results, as outlined. The resulting L3O$_{LM}$ subset still has less days with data than in L3L for 61 % of coastal grid boxes.

- Almost a quarter (half) of coastal grid boxes see a significant difference in ltm VMR between L3L and L3O$_L$ (L3O$_{LM}$). Over a third (almost a quarter) of the trends in L3O$_L$ (L3O$_{LM}$) are significantly different to L3L.

- Focusing on the L3 grid boxes containing the 33 largest coastal cities in the world, mean VMRs in L3O$_L$ and L3L differ significantly for 11 of the 27 grid boxes that can be compared (40 %; there are no L3O$_L$ data for the remaining 6 cities). The L3L – L3O$_{LM}$ mean VMR difference across all 33 grid boxes is relatively small (3.7 ppbv), but this does hide some much larger discrepancies, with the difference exceeding 10 ppbv for 11 of the 33 grid boxes and 20 ppbv for 3 of them. The difference is significant for 13 of 33 grid boxes (39 %). Of the 18 grid boxes where WLS analysis can be performed in L3O$_L$, there are 9 cases where the trend is significantly different to that in L3L. The trends in L3O$_{LM}$ and L3L differ significantly for 5 of the 33 grid boxes.

From these results, it can be concluded that, yes, for at least a quarter of all MOPITT coastal L3 grid boxes, it does matter that there is limited capacity to filter out the influence of retrievals over water in L3 data – at least without a huge loss of temporal coverage. Demonstrably, there are significant differences in the mean VMRs and temporal trends that can be obtained using L3O and L3L, sometimes very large. These differences could have tangible consequences, depending on the purpose for which the MOPITT data are being used. While acknowledging that this analysis has also shown that there is a sizeable proportion of coastal grid boxes where statistically, mean VMRs and trends do not differ significantly between L3L and L3O, there is enough evidence to suggest that an additional L3 "land-only" product, created only from averaging bounded L2 retrievals performed over land – the L3L dataset that has been analysed in this paper – could be beneficial to the research community. This L3L dataset enables L3 users to maximize retrieval information content for coastal L3 grid boxes, as is currently only possible with L2 data, while also preserving the benefits of L3 products, such as smaller file size and greater accessibility of gridded products. The L3L dataset analysed in this paper is publicly available for download (Ashpole and Wiacek, 2022; L3W is also available). Although this paper has focused only on analysis of MOPITT data, it is reasonable to question whether the findings are applicable to data products from other satellite instruments that make CO retrievals based on observed thermal-infrared radiances, such as AIRS (Atmospheric InfraRed Sounder), TES (Tropospheric Emission Spectrometer), and IASI (Infrared Atmospheric Sounding Interferometer).

**Appendix A:** List of short names and abbreviations used in the main article text, their full descriptive name, and the purpose of use (along with the section it is first introduced)

| Short name / abbrev. | Full descriptive name | Purpose (Section introduced) |
|---|---|---|
| AK | Averaging Kernel | General abbreviation (2.1) |
| LTM | Long-term mean | General abbreviation (3.3.2) |
| MOPITT | Measurements of Pollution in the Troposphere | Instrument abbreviation (1) |
| VMR | Volume Mixing Ratio | General abbreviation (1) |
| VMR ret | Retrieved VMR | General abbreviation (3.1.1) |
| VMR apr | a priori VMR | General abbreviation (3.1.1) |
| L2 | Level 2 dataset | Dataset identifier (1) |
| L3 | Level 3 dataset | Dataset identifier (1) |
| L3L | A new L3 "land-only" dataset, created only from Level 2 retrievals performed over land (creation method outlined in Sect. 2.4) | Dataset identifier (1) |
| L3O | Original, "as downloaded" Level 3 (L3) dataset | Dataset identifier (1) |
| $L3O_L$ | Subset of L3O only containing L3 retrievals with a surface index of land | Dataset identifier (2.4) |
| $L3O_{LM}$ | Subset of L3O only containing L3 retrievals with a surface index of land OR mixed | Dataset identifier (2.4) |
| $L3O_M$ | Subset of L3O only containing L3 retrievals with a surface index of mixed | Dataset identifier (2.4) |
| $L3O_{NF}$ | The L3O dataset with no filtering by surface index ($L3O_{NF}$ is identical to L3O) | Dataset identifier (2.4) |
| $L3O_W$ | Subset of L3O only containing L3 retrievals with a surface index of water | Dataset identifier (2.4) |
| L3W | A new L3 "water-only" dataset, created only from Level 2 retrievals performed over water (creation method outlined in Sect. 2.4) | Dataset identifier (1) |
| n_days(L3[A]) | Number of days in L3 dataset A, e.g. n_days(L3L) | Dataset metric (2.3) |
| n_days(L3[A]/L3[B]) | A ratio quantifying the relative number of observations in L3 dataset A compared to L3 dataset B, e.g. n_days($L3O_L$/L3O) | Dataset metric (2.3) |
| $n\_ret_L$ | Number of L2 retrievals that are used for calculating the area averages when creating L3L | Dataset metric (2.4) |
| $n\_ret_W$ | Number of L2 retrievals that are used for calculating the area averages when creating L3W | Dataset metric (2.4) |
| ratio(land/water) | $n\_ret_L$/$n\_ret_W$: A ratio used to indicate the proportion of an L3 grid box that is covered by land vs water | Dataset metric (2.4) |
| $SIGDIFF_{L3L-L3W}$ | L3 gridboxes where the mean VMR in L3L and L3W is significantly different ($p < 0.1$) | Grid box subset identifier (3.2.1) |
| $NOTSIGDIFF_{L3L-L3W}$ | L3 gridboxes where the mean VMR in L3L and L3W is *not* significantly different ($p > 0.1$) | Grid box subset identifier (3.2.1) |
| $BOTH_{VMRs}$ | L3 grid boxes where the mean VMR in L3L is significantly different to that in both L3W and $L3O_M$ | Grid box subset identifier (3.2.2) |
| $L3L\_L3W\_ONLY_{VMRs}$ | L3 grid boxes where the mean VMR in L3L is significantly different to that in L3W but *not* in $L3O_M$ | Grid box subset identifier (3.2.2) |
| $BOTH_{TRENDS}$ | L3 grid boxes where the detected trend in L3L is significantly different to that in both L3W and $L3O_M$ | Grid box subset identifier (3.2.2) |
| $L3L\_L3W\_ONLY_{TRENDS}$ | L3 grid boxes where the detected trend in L3L is significantly different to that in L3W but *not* in $L3O_M$ | Grid box subset identifier (3.2.2) |
| $SIGDIFF_{L3L-L3OLM}$ | L3 gridboxes where the mean VMR in L3L and $L3O_{LM}$ is significantly different ($p < 0.1$) | Grid box subset identifier (3.4.2) |
| $NOTSIGDIFF_{L3L-L3OLM}$ | L3 gridboxes where the mean VMR in L3L and $L3O_{LM}$ is *not* significantly different ($p > 0.1$) | Grid box subset identifier (3.4.2) |

990

**Data availability**

The "L3L" and "L3W" datasets analysed in this study are available from the following link: https://doi.org/10.5683/SP3/ERCG2H (see also Ashpole and Wiacek, 2022). Code for creating these datasets is available here: https://github.com/ianashpole/MOPITT_L3L_L3W. The MOPITT V8 joint TIR-NIR files Level 2 ("MOP02J") and Level 3 ("MOP03J") datasets can be accessed from the following URLs, respectively: https://doi.org/10.5067/TERRA/MOPITT/MOP02J_L2.008 (NASA/LARC/SD/ASDC, 2000a) and https://doi.org/10.5067/TERRA/MOPITT/MOP03J_L3.008 (NASA/LARC/SD/ASDC, 2000b)

**Author contributions**

IA and AW jointly conceived of and designed the study. IA performed data analysis; both authors examined and interpreted the results, and prepared the manuscript.

**Competing interests**

The authors declare that they have no conflict of interest.

**Acknowledgements**

The authors received funding from the Canadian Space Agency through the Earth System Science Data Analyses program (grant no. 16SUASMPTN), the Canadian National Science and Engineering Research Council through the Discovery Grants Program, and Saint Mary's University. We thank the MOPITT team for providing the data used in this study. The authors would also like to thank the anonymous reviewers and the associate editor whose thoughtful comments helped to improve this manuscript.

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
