# Peer review of "Differences in MOPITT surface level CO retrievals and trends from Level 2 and Level"

_Atmospheric Measurement Techniques, 2022_

## Referee Comment (RC2)

**Table 1**

| city | surface (miles^2) | surface (km^2) | L3 pixels | L2 pixels |
|---|---|---|---|---|
| Chittagong | 65 | 168.35 | 0.01 | 0.35 |
| Dubai | 13.5 | 34.97 | 0.00 | 0.07 |
| Accra | 87.13 | 225.67 | 0.02 | 0.47 |
| Sydney | 4775 | 12367.25 | 1.00 | 25.55 |
| Wenzhou | 4658 | 12064.22 | 0.98 | 24.93 |
| Abidjan | 818 | 2118.62 | 0.17 | 4.38 |
| Yangon | 231.2 | 598.81 | 0.05 | 1.24 |
| Qingdao | 4335 | 11227.65 | 0.91 | 23.20 |
| Dar Es Salaam | 614 | 1590.26 | 0.13 | 3.29 |
| Surat | 183 | 473.97 | 0.04 | 0.98 |
| Miami | 55.25 | 143.10 | 0.01 | 0.30 |
| Toronto | 243.3 | 630.15 | 0.05 | 1.30 |
| Hong Kong | 427 | 1105.93 | 0.09 | 2.28 |
| Shantou | 868 | 2248.12 | 0.18 | 4.64 |
| Singapore | 281.3 | 728.57 | 0.06 | 1.51 |
| San Francisco | 46.87 | 121.39 | 0.01 | 0.25 |
| Luanda | 43.63 | 113.00 | 0.01 | 0.23 |
| Saigon | 809 | 2095.31 | 0.17 | 4.33 |
| Kuala Lumpur | 93.82 | 242.99 | 0.02 | 0.50 |
| Taipei | 104.9 | 271.69 | 0.02 | 0.56 |
| Xiamen | 657 | 1701.63 | 0.14 | 3.52 |
| Chennai | 164.5 | 426.06 | 0.03 | 0.88 |
| Istambul | 2063 | 5343.17 | 0.43 | 11.04 |
| Buenos Aires | 78.38 | 203.00 | 0.02 | 0.42 |
| Karachi | 1459 | 3778.81 | 0.31 | 7.81 |
| Osaka | 86.1 | 223.00 | 0.02 | 0.46 |
| Bangkok | 606 | 1569.54 | 0.13 | 3.24 |
| Lagos | 452 | 1170.68 | 0.10 | 2.42 |
| New York | 302.6 | 783.73 | 0.06 | 1.62 |
| Mumbai | 233 | 603.47 | 0.05 | 1.25 |
| Manila | 14.88 | 38.54 | 0.00 | 0.08 |
| Shanghai | 2448 | 6340.32 | 0.51 | 13.10 |
| Tokyo | 847 | 2193.73 | 0.18 | 4.53 |

---

## Author Comment (AC1)

We sincerely thank the reviewers for their very helpful feedback on this paper. We address all their comments and suggestions below.

*Reviewer comments are in black italicised text.*

Our responses are in blue, regular text.

Where practical/necessary, we provide a screenshot of the track-changes document to show the changes that we have made (in outlined boxes). In these, text that is removed is  and coloured red, while new text is underlined and coloured blue.

Ian Ashpole (on behalf of both authors).

**REVIEWER # 1**

*23 "These L3L L3W differences are clearly linked to retrieval sensitivity differences"*

*While reading the abstract it was unclear if the authors were taking into consideration the fact that CO emissions over water are negligible and whether that affects the difference between L3L and L3W.*

This is a fair point, but it is also important to consider that due to the relatively long-lived, well-mixed nature of atmospheric CO, the portion of coastal L3 grid boxes situated over water are unlikely to represent "pristine" conditions. Moreover, there are a large portion of coastal L3 grid boxes where emissions from the land-based component are also negligible, due to the lack of large anthropogenic CO sources or natural emission hotspots (see distribution of coastal L3 grid boxes in Figure 2). Finally, we do demonstrate cases where retrieved surface-level CO concentrations in L3L are less than in L3W for a given L3 grid box, and we would not expect to see this if the differences are simply related to differences in emissions. It therefore seems unlikely that the L3L-L3W CO concentration and trend differences presented are strongly impacted by land-water emission differences within these grid boxes – especially against the weight of evidence that links these to well-understood retrieval sensitivity contrasts. However, we do understand that it is important to mention that there are plausible physical factors that could contribute to L3L-L3W differences in some circumstances (i.e. that not all differences are solely a retrieval artefact), and we have included a paragraph in Section 3.2.1 (where L3L-L3W retrieved VMR differences are shown and discussed) to address this:

| | |
|---|---|
| 493 |   It should be noted that there are additional physical factors that could |
| 494 | plausibly play a role in generating the L3L – L3W retrieved VMR difference that is observed, in addition to |
| 495 | retrieval sensitivity. Given that most CO sources are land-based, a decrease in VMRs from land to water |
| 496 | might be expected, especially in the LT. However, this assumption only seems reasonable where large CO |
| 497 | sources are proximal to the coastline, as it is unrealistic to expect gradients as large as we observe in |
| 498 | background CO (which coastal grid boxes far from large CO sources are more likely to represent) across the |
| 499 | relatively small distance covered by a L3 grid box. Given the relatively long-lived, well-mixed nature of |
| 500 | atmospheric CO, VMRs retrieved at a given location are a function of both local emissions *and* transport, |
| 501 | and the portion of coastal L3 grid boxes situated over water therefore do not represent pristine conditions in |

**screenshot continues on next page**

502    comparison to the adjacent land-based portion of the grid boxes. This is verified by comparing a priori VMRs
503    (also shown in Figure 4), which suggest the land-water difference in CO concentrations should be negligible
504    (mean L3L – L3W a priori VMR difference = 0.69 ppbv, compared to a mean retrieved VMR difference of
505    10.29 ppbv). The above reasoning can also be applied to the question of whether wind direction is responsible
506    for creating the observed L3L – L3W difference in retrieved VMRs: It could be hypothesised that a prevailing

507    onshore wind may lead to CO concentrations being higher over land than water, yet the negligible L3L –
508    L3W a priori VMR difference, the fact that atmospheric CO is well-mixed, and the clear land-water
509    sensitivity gradient that has been demonstrated suggest that wind direction does not play a big role in creating
510    the land-water difference observed in retrieved VMRs. To further rule out the role of wind direction, the L3L
511    – L3W retrieved VMR comparison has been analysed alongside wind direction for several case study grid
512    boxes, and there appears to be no notable shift in wind direction whether L3L or L3W is greater for a given
513    grid box. Results for this analysis are given in the Supp. Mat. (SM5). The weight of evidence therefore points
514    towards L3L – L3W retrieved VMR differences being a function of reduced retrieval sensitivity over water
515    compared to land.

*421 "As expected from the previous analysis, the land-water sensitivity contrast is greater when mean VMRs are significantly different than when not."*

*It's as if they assume that all land-water contrast is a processing artifact. See line 446*

This comment is addressed by our response to the reviewer's comments on L23 (above) and L446 (below).

*446 "An underlying assumption is that the temporal trend in "true" VMRs should not vary much across a 1° x 1° L3 grid box."*

*This was revisited in line 399. It seems as if the authors neglect to account for wind direction. If the prevailing wind is blowing from the ocean inland (e.g. the coast of California) then the CO concentration could be much higher over the land than the water in the same gridbox. Whereas if the wind was blowing*

*out to sea, one would expect far less difference between L3L and L3W. Yet it appears that the authors are not accounting for this.*

The reviewer is correct that we do not account for the effect of wind direction on L3L-L3W differences. While we agree in principle with their statements about the expected effect that wind direction could have on L-W differences in VMRs in some cases, we did not address this for the following reasons:

1. Ashpole and Wiacek (2020 AMT) explicitly considered this for the coastal city of Halifax, Canada (population > 400,000) and found that shifts in wind direction (based on ERA Interim reanalysis data) could not account for differences in retrieved CO over land and water for the L3 grid box containing Halifax.
2. Coastal marine CO is unlikely to be "background" levels due to the relatively long-lived, well-mixed nature of atmospheric CO. Additionally, not all coastal grid boxes are separated from downstream sources by thousands of km's of ocean (as in the California example – see distribution of coastal grid boxes in Figure 2).
3. The wind direction is unlikely to be onshore (synoptic or from a well-developed sea breeze) for all coastal grid boxes where L3L > L3W.
4. Hedelius et al. (2021, JGR), demonstrate that most large cities do not stand out against the surrounding region in trend analyses based on MOPITT data (a maximum of 21 of 500 cities compared saw faster trends than the regional average).

However, to evaluate any potential impact of wind direction on our results, we have compared wind direction (taken from ERA Interim data, u and v wind vectors at 10-m level) for 6 of the case-study L3 grid boxes containing large cities considered in Sect 3.4 for days when L3L > L3W and L3L < L3W. We found no marked differences in wind direction on these days for any of the grid boxes considered, giving confidence that wind direction does not have a large impact on our results. Results are presented for the case study of the grid box containing Dubai below (Figure R1), and we include results for all 6 cities in the supplementary material (SM5). As with the comment above, we do acknowledge that it is important to mention that there are plausible physical factors that could contribute to L3L-L3W differences in some circumstances (i.e. that not all differences are solely a retrieval artefact), and we have included a paragraph in Section 3.2.1 (where L3L-L3W retrieved VMR differences are shown and discussed) to address this (see paragraph inserted above in response to comment for L23).

[Figure]

**Figure R1:** Analysis of surface wind direction for days when there is a surface level VMR retrieval over both land and water in the L3 grid box containing Dubai ("L3L" and "L3W", resp.) Wind direction data are calculated from u and v vector components at 10-m above the surface, taken from daily mean ERA Interim data. Wind rose barbs in panels b-d depict the direction in which wind is blowing *from* – e.g. predominantly *from the west* in all cases shown. Note that only the sub-period 01-01-2002 to 31-08-2017 is considered in this analysis (as opposed to the full study period of 2001-08-25 to 2019-02-28 in the submitted manuscript) owing to local availability of ERA Interim data. **(a)** NASA Blue Marble image of the region surrounding Dubai. The boundaries of the L3 grid box containing the city are shown by red-dashed lines, with the city location indicated by the pink marker. **(b)** Wind rose showing wind direction taken from the grid box containing Dubai in ERA Interim data. Wind direction data are taken for all days with L3L and L3W data for this grid box for the study period, and the wind rose displays the mean wind direction for all these days. Number of days represented is given by n value in panel label. **(c)** As b, but only for days when retrieved surface-level VMR is greater in L3L than L3W ("L3L > L3W"). **(d)** As b, but for days when L3L < L3W.

*110 "It cannot be overlooked that working with L3 data thus requires fewer computing resources and less technical proficiency"*

*Agreed. Furthermore gridded products are accessible by many more tools that users are familiar with such as Panoply.*

Agreed. We have added mention of the availability of simple-to-use tools for working with gridded products such as MOPITT L3:

| 114 | size (~25 MB vs ~450 MB respectively, for a single daily, global file). It cannot be overlooked that working |
| --- | --- |

| 115 | with L3 data thus requires fewer computing resources and less technical proficiency, with a range of simple- |
| --- | --- |
| 116 | to-use tools available for working with gridded products. L3 products thus make the MOPITT data more |
| 117 | easily accessible, especially to less-expert users, who may lack the expertise required to scrutinize the data |

*177 "Validation results are comparable to V8. It is expected that the main conclusions of this paper to hold for V9, since the land-water sensitivity contrast remains and L3 processing method appears to be unchanged."*

*Actually, V9 discards far fewer L2 pixels due to cloudiness than V8 which may affect the results. I suggest the authors repeat some experiments with V9 to confirm their assumption. Yes, they are correct that the L2 → L3 processing method is unchanged.*

Thank you for confirming that the L2 → L3 processing method is unchanged.

Regarding the potential for results to be different in V9 than V8: we have re-done certain sections of the analysis using V9 data to verify that the main conclusions presented in our submitted manuscript based on V8 data hold for V9 too. Results of this analysis are outlined below. We hope it is appreciated that re-doing the entirety of the analysis is not practical (or reasonable) given that the original work was undertaken

before V9 was released. We have included these results in the supplementary material (SM1), and reference these where V9 is outlined:

| | |
|---|---|
| 184 | overview of MOPITT V9 is given by Deeter et al (2021).  |
| 185 |  |
| 186 |  A subset of the analysis presented in this paper has been duplicated using V9 data, and this |
| 187 | confirms that the main conclusions drawn based on V8 data also hold for V9 (this analysis is outlined in the |
| 188 | Supp. Mat. (SM1)). This is to be expected, given that the land-water sensitivity contrast remains in V9 and |
| 189 | the L3 processing method is unchanged.  |

Results of analysis with V9:

1. The global land-water sensitivity contrast shown in Figure 3 (V8) is also present if Figure 3 is re-plotted using V9. This confirms that the land-water sensitivity contrast remains. Both Figures are shown in Figure R2, below.

2. The land-water sensitivity differences within coastal L3 grid boxes (demonstrated by comparing L3L and L3W for these grid boxes), shown by the scatterplots in Figure 4 (V8), is also present if the analysis is reproduces using V9 and Figure 4 replotted with these data. Both Figures are shown in Figure R3, below.

3. We repeated the comparison of a) mean surface level retrieved VMR, and b) temporal trends therein, for selected grid boxes containing large cities analysed in Sect. 3.4 of the submitted manuscript, using V9 data. Results for both V8 and V9 are shown in Table R1, below, and similar differences to V8 exist in V9. Although this analysis is restricted to L3L and L3W only, given that the L2 → L3 processing method is unchanged there is every reason to expect that similar differences would emerge for L3O V9 subsets too.

(Note that due to time and data storage limitations, the V9 analysis was restricted to the data years 2010-2015 inclusive for results 1 and 2 above. Given the clarity of the results however, we are confident that this would remain if the whole study period considered for V8 had been reanalysed using V9. L3L and L3W time series for the full period studied using V8 were able to be obtained for V9 for the analysis leading to result 3 above).

**Table R1.** Mean retrieved surface level VMR, and its temporal trend, in L3L and L3W, for L3 grid boxes containing selected cities from Sect. 3.4 of the submitted manuscript, using V8 and V9.

| | | V8 | | | V9 | | | V9 conclusion |
|---|---|---|---|---|---|---|---|---|
| | | L3L | L3W | Δ (L–W) | L3L | L3W | Δ (L–W) | same as V8? |
| Bangkok | Mean | 314.4 | 261.3 | 53.1 | 286.4 | 268.3 | 18.1 | Y |
| | Trend | -3.03 | -2.00 | -1.03 | -2.93 | -2.21 | -0.72 | Y |
| Dubai | Mean | 180.0 | 163.3 | 16.7 | 186.6 | 168.8 | 17.8 | Y |
| | Trend | -2.90 | -0.90 | -2.00 | -2.97 | -1.06 | -1.91 | Y |
| Hong Kong | Mean | 336.1 | 260.1 | 76.0 | 307.2 | 270.5 | 36.7 | Y |
| | Trend | -8.06 | -3.55 | -4.51 | -7.50 | -4.37 | -3.13 | Y |
| Miami | Mean | 160.7 | 143.5 | 17.2 | 158.7 | 149.8 | 8.9 | Y |
| | Trend | -1.52 | -0.75 | -0.77 | -1.44 | -1.08 | -0.36 | Y |
| Sydney | Mean | 94.0 | 86.8 | 7.2 | 89.2 | 88.6 | 0.6 | Y |
| | Trend | -0.74 | -0.24 | -0.50 | -0.58 | -0.42 | -0.16 | Y |
| Toronto | Mean | 238.4 | 254.5 | -16.1 | 240.7 | 255.9 | -15.2 | Y |
| | Trend | -1.09 | -1.99 | 0.9 | -1.71 | -1.87 | 0.16 | Y |

**(a – V8)**

[Figure]

**Figure R2. (a)** Mean sensitivity metrics and VMRs (retrieved and a priori) from coastal L3 grid boxes, from MOPITT V8 data. Values compared in the scatterplots are mean values from matched L3L and L3W retrievals within these grid boxes. "Matched" means that only days when both L3L and L3W are present, and the L3O surface index is mixed, are used to create the mean values analysed. Shown are AK diagonal values (left column), AK rowsums (second column), absolute VMR retrieved minus a priori values (third column), retrieved (fourth column) and a priori (fifth column) VMRs, for the following levels of the retrieved profile: surface (top row), 900 hPa (second row), 800 hPa (third row), 600 hPa (fourth row), and 300 hPa (bottom row). Values in boxes in the top-left corner of each panel correspond to mean values across all L3L and L3W grid boxes. NOTE: This is a reproduction of Figure 4 from the submitted manuscript.

***Figure R2 (b) shown on next page***

**(b – V9)**

[Figure]

**Figure R2. (b)** As Figure R2 (a) except using MOPITT Version 9 data and for the sub-period 2010-2015 (inclusive), as explained in response text above.

**(a – V8)**

[Figure]

**Figure R3. (a)** Mean sensitivity metrics from MOPITT L3 data (Version 8), averaged across the entire study period (September 2001 – February 2019, inclusive). Shown are AK diagonal values (left column), AK rowsums (center column) and VMR retrieved minus a priori values (right column) for the following levels of the retrieved profile: surface (top row), 900 hPa (second row), 800 hPa (third row), 600 hPa (fourth row), and 300 hPa (bottom row). Values in white boxes correspond to mean values across all land ("L") and water ("W") L3 grid boxes. NOTE: This is a reproduction of Figure 3 from the submitted manuscript.

***Figure R3 (b) shown on next page***

**(b – V9)**

[Figure]

**Figure R3. (b)** As Figure R3 (a) except using MOPITT Version 9 data and for the sub-period 2010-2015 (inclusive), as explained in response text above.

*193 "which at the time of writing, is the most recent data quality summary"*

*More recent data quality statements are available now. See*
*https://asdc.larc.nasa.gov/documents/mopitt/mopitt_quality_statements.html*

We thank the reviewer for pointing this out. However, our reference needs to remain to the data quality statement for Version 6 data, because none of the more recent data quality summaries explain how the Level 3 surface classification is derived (which is what our reference to the data quality statement is for). We have clarified this in the paper:

| 203 | 1 (this information is taken from the MOPITT Version 6 L3 data quality summary[1], which at the time of |
| 204 | writing, is the most recent data quality summary to detail exactly how L3 data are created, despite more |
| 205 | recent data quality summaries being available). Note that the L2 VMR profiles that are averaged to produce |

*483-485 "However, the results presented do imply a general tendency for trend underestimation in retrievals over water within coastal grid boxes compared to retrievals over land in the same grid boxes obtained at the same times, which appears to be linked to differences in retrieval sensitivity."*

*This feels like the most important point of the paper. Perhaps more effort to demonstrate and quantify would be helpful.*

This point is explicitly discussed in Section 3.2.1, with trend differences between L3L and L3W demonstrated and quantified in Figure 5c-d and Table 2.

To help with the presentation and communication of our results, we have undertaken a thorough edit of the manuscript, where the focus has been on clarifying the points being made and, where possible, moving methodological details (which are flagged as a source of confusion by the reviewer elsewhere in this review) either to the end of paragraphs as caveats to the points, or to figure captions. The most comprehensive edits are in the Results and Discussion Section (Section 3).

We include a screenshot below of the subsection containing the above quote, to demonstrate how the point being made is now more clearly framed in the revised manuscript following the edit.

559    In summary, these results show a general tendency for trend underestimation in surface

560  level retrievals over water compared to retrievals over land in the same coastal grid boxes obtained at the

561  same times, which appears to be linked to differences in retrieval sensitivity.

562

563   The relationships found in these analyses are not

564  perfect because trend differences are sensitive to several other factors, in addition to differences in retrieval

565  sensitivity. For example, a greater trend difference would be evident if the rate of change in "true" CO

566  concentrations is faster than if it is slow/negligible, for a given sensitivity difference. Similarly, there should

567  be zero trend difference if "true" CO concentration levels are stable over time, irrespective of the magnitude

568  of difference in retrieval sensitivity. The accuracy of the a priori is a further complicating factor.

569

570

571   An underlying assumption is also that the temporal

572  trend in "true" VMRs should not vary much across a $1°$ x $1°$ L3 grid box. Hedelius et al. (2021) lends credence

573  to this assumption with the finding that CO trends are similar within regions spanning a few thousand

574  kilometres (L3 grid boxes are ~ 100 $km^2$), and that trends within urban areas are generally indistinguishable

575  from the trend of the broader region encompassing the urban area, despite an expectation that urban trends

576  should exceed the regional background due to a concentration of CO emission sources here.

*The discussion in the paragraph starting on line 417 is very important however it would have even more impact if it it included the consideration of one more bit of information. The skill of MOPITT retrievals of CO (VMR) is not random. It is dependent on conditions such as thermal contrast between the surface and the air, which is what the authors are describing when they see discontinuities between L3L and L3W. However another factor is that MOPITT sees CO better when there's a lot of it. The uncertainty (as measured by DFS) decreases when the CO signal is large. So if there's less CO over the ocean due to fewer sources, that will also affect the results of this analysis.*

This comment has mostly been addressed above, in response to the reviewer's comments on L23, L421 and L426. To re-iterate our response: it is true that CO emissions are negligible over ocean compared to land, however the potential impact that this will have on the results of our analysis appears weak in the face of:

1. The portion of coastal L3 grid boxes situated over water being unlikely to represent "pristine" background conditions owing to the well-mixed and relatively long-lived nature of atmospheric CO. i.e. the CO loading for a given grid box is not simply a result of emission from within that grid box;

2. There being a large portion of coastal L3 grid boxes where emissions from the land-based component are also negligible, due to the lack of large anthropogenic CO sources or natural emission hotspots;

3. There being cases where retrieved surface-level CO concentrations in L3L are less than in L3W for a given L3 grid box, and we would not expect to see this if the differences are simply related to differences in emissions.

4. CO Trends detected over cities in MOPITT data not tending to stand out from the regional average despite the presumption of greater emissions there (Hedelius et al., 2021).

It therefore seems unlikely that the L3L-L3W CO concentration and trend differences presented for coastal grid boxes are strongly impacted by land-water emission differences within these grid boxes – especially against the weight of evidence that links these to well-understood retrieval sensitivity contrasts, as the review themselves outline. However, we have included a more thorough discussion of this point in the text (Section 3.2.1 – see response to comments on L23, L421 and L426 above).

Additionally, we do mention in the text (introduction section) that retrieval sensitivity is also linked to the amount of CO present:

| 70 | these a priori CO profiles are based on a monthly climatology from a chemical transport model. The degree |
|----|------|
| 71 | to which a given MOPITT retrieval reflects information obtained from the observed radiances – known as |
| 72 | "information content" – is highly spatially and temporally variable, depending on scene-specific factors such |
| 73 | as surface temperature, thermal contrast in the lower troposphere, and the actual ("true") CO loading itself, |
| 74 | as well as on instrumental noise (e.g. Deeter et al., 2015). The lower the retrieval information content, the |
| 75 | closer the retrieved CO loading will be to the a priori; a model value. |

*451 – The word "gradient" appears but I'm having trouble understanding its definition here. Is it just the difference between temporal trends of L3L and L3W? Or is it spatial?*

Apologies for the confusion – we were referring to "trend", which is the word that should have been used in the first place for clarity and consistency with the rest of our wording. Note that this sentence has been removed in the revised version of the manuscript, following a thorough edit of the text for clarity.

*Table 1 – It took me several attempts to understand what the "d" column was.*

Apologies for the confusion – we have changed the column heading and explained its meaning in the caption, for greater clarity.

**Table 1.** Mean values for selected variables from L3L and L3W for coastal L3 grid boxes, matched retrievals only. "Matched" means that only days when both L3L and L3W are present and the L3O surface index are mixed are used to create the mean values analysed. Mean values are calculated and presented separately according to the results of a 2-tailed student's t-test (unequal variance) performed on mean retrieved VMR values in L3L and L3W (n = 3971). Mean L3L – L3W differences are also shown for each subset ('L-W')

| | P < 0.1 (n=2379, 60 %) | | | P > 0.1 (n=1592, 40 %) | | |
|---|---|---|---|---|---|---|
| | land | water | d | land | water | d |
| Mean vmr_ret | 129.97 | 117.41 | 12.55 | 133.52 | 126.60 | 6.90 |
| Mean vmr apr | 113.78 | 113.18 | 0.61 | 124.65 | 123.83 | 0.83 |
| Mean ret-apr | 16.18 | 4.24 | 11.94 | 8.87 | 2.77 | 6.09 |
| Mean ak rowsum | 0.43 | 0.18 | 0.24 | 0.44 | 0.27 | 0.16 |

| | P < 0.1 ("SIGDIFF") (n=2379, 60 %) | | | P > 0.1 ("NOT_SIGDIFF") (n=1592, 40 %) | | |
|---|---|---|---|---|---|---|
| | L3L | L3W | L-W | L3L | L3W | L-W |
| Mean vmr_ret | 129.97 | 117.41 | 12.55 | 133.52 | 126.60 | 6.90 |
| Mean vmr apr | 113.78 | 113.18 | 0.61 | 124.65 | 123.83 | 0.83 |
| Mean ret-apr | 16.18 | 4.24 | 11.94 | 8.87 | 2.77 | 6.09 |
| Mean ak rowsum | 0.43 | 0.18 | 0.24 | 0.44 | 0.27 | 0.16 |

*483-485 "However, the results presented do imply a general tendency for trend underestimation in retrievals over water within coastal grid boxes compared to retrievals over land in the same grid boxes obtained at the same times, which appears to be linked to differences in retrieval sensitivity."*

*This seems like a valid conclusion based on the analysis performed. There are a lot of details about the methodology to arrive at this conclusion that confused me more than served as support for clear statements such as this.*

We apologise that some of our writing caused confusion to the reviewer. As outlined in response to an earlier comment, we have undertaken a thorough edit of the text, where the focus has been on clarifying the points being made and, where possible, moving extra methodological details either to the end of paragraphs as caveats to the points, or to figure captions.

*613 "In these instances, L3O would therefore seem to be misclassified."*

*This is a valuable insight.*

*820-823 "there is enough evidence to support the suggestion from Ashpole and Wiacek (2020) that an additional L3 "land-only" product, created only from averaging bounded L2 retrievals performed over land – the L3L dataset that has been analysed in this paper – would be beneficial to the research community."*

*This recommendation will be brought to the attention of the MOPITT science team. It will hopefully be incorporated in the archival processing version.*

*General Comments:*

*These researchers took a very close look at how the MOPITT L3 product is created and have identified a flaw in the way the MOPITT team processes pixels into coastal grid cells by mixing retrievals of uneven quality. This distorts the values reported for a non-insignificant number of gridcells. Their conclusions appear valid and robust. However, I had a difficult time following the arguments and methodolgy of the paper (1). I didn't understand why they were focused on surface level retrievals instead of higher in the atmosphere where MOPITT is considerably more sensitive (2). I was curious if they would have come to the same conclusion if they looked at MOP03M (monthly mean) products which have far less random noise and greater coverage than the daily L3O products (3). In several places, the authors were making a clear distinction between two situations and I had trouble understanding the meaning of this distinctions. For example: "For other datasets, whether the marker is filled or not, and whether the lines are solid or dash/dot, depends on the outcome of an independent, 2-tailed t-test assuming unequal variance (aka "Welch's test") against L3L: filled markers and solid lines indicate the mean is significantly different to L3L (p < 0.1); open markers and dash/dot lines indicate there is no significant difference to L3L." This distinction was too difficult for me to understand its significance (4).*

*I believe the researchers can transform this paper into a valuable analysis by having a more clarified statement of their conclusions and focusing the readers' attention on the evidence that supports that point.*

We thank the reviewer for their time and thoughts on this paper, helpful suggestions for improvement, and their positive comments. We have addressed their specific comments above, but note four more points from this summarising paragraph which we address in turn below: (the numbers 1-4 below correspond to the numbers we have added in parentheses to the general comment above, to highlight the separate points being made and make our response clearer).

(1) To make the arguments and methodology clearer and easier to follow, we have undertaken a thorough edit of the text, where the focus has been on clarifying the points being made and, where possible, moving methodological details either to the end of paragraphs as caveats to the points, or to figure captions. We have also clarified our methods further in Section 2 (e.g. outlining the different time series being analysed in different parts of the results section).

(2) As stated at the end of Section 3.1.2 (L383-385), this paper focuses on the surface-level of the retrieved profile, since the LT is where discrepancies are greatest, and the cause of this sensitivity contrast is well established (as outlined in the introduction). The surface-level is also of most

interest for identifying potential air quality impacts for humans (e.g. Buchholz et al., 2022). We have added this second justification to the text:

414       This analysis clearly shows how L2 retrievals that are averaged together to create the L3O data over
415 coastal grid boxes have differing degrees of sensitivity, especially in the LT. This is explicitly cautioned
416 against in the MOPITT data user's guide (MOPITT Algorithm Development Team, 2018). The remainder of
417 this paper focuses on the surface-level of the retrieved profile, since the LT is where discrepancies are

418 greatest, and the cause of this sensitivity disparity is well established: differing thermal contrast conditions
419 near to the surface over land and water; and a lack of NIR radiances being used in the retrieval over water.
420 Furthermore, the surface-level is of most interest for identifying potential air quality impacts for humans (e.g.
421 Buchholz et al., 2022).
422

(3) MOP03M products are unsuitable for this analysis as there is not a corresponding L2-monthly product to co-analyse. However, there is every reason to expect that the conclusions would be the same, since the MOP03M products are created from the L2 retrievals that we analyse in this paper. We have added a note clarifying this in the text, at the end of Section 2.4 where the products we analyse are outlined:

295       Note that the analysis presented in this paper is restricted to daily products. Monthly L3 files are
296 available, however the absence of a monthly L2 product precludes the analysis from being conducted on
297 those data. Based on the results of the analysis of daily data, however, there is reason to also advise caution
298 if working with coastal grid boxes in the monthly L3 product. This is because the data for those grid boxes
299 will still be created from daily L2 retrievals over land and water, with the same implications that are discussed
300 in this paper.
301
302
303 **2.5. Timeseries preparation, sStatistical methods used for this study, and additional data sources**

(4) Again, we apologise that some of our writing caused confusion to the reviewer. As outlined previously, we have undertaken a thorough edit of the text to address this, and hope that the meaning of distinctions such as that outlined is now clearer. Regarding the situation that is outlined, the text now makes clear that L3L is the dataset that others are being compared to.

*The manuscript describes a study of MOPITT V8 TIR-NIR surface CO retrievals over 33 coastal cities. Daily L3 data (data gridded to 1°x1°, 111x111 km$^2$ per pixel) and daily L2 data (22x22km$^2$ per pixel at nadir) are analyzed. This study's main findings are that statistics of coastal cities obtained from L3 and L2 products differ, that "mixed" L3 pixels (L3 pixels averaging both water and land L2 pixels) are not suitable to study coastal cities, and that a L3 land only product for coastal pixels is needed. In order to demonstrate these points, several comparisons and statistical analyses between land and water L3 TIR-NIR pixels (original and re-created from L2 data) are performed. The manuscript is well written.*

*Two major issues are described below.*

*1. Use of TIR-NIR data in land/water comparisons*

*As described in Deeter et al., 2013, among others, TIR-NIR retrievals over land and over water are fundamentally different, since NIR radiances cannot be used in the latter. The authors acknowledge the fact that retrievals over water are limited to the TIR band due to the lack of NIR signal, but don't acknowledge the implications, which are key. Using the TIR-NIR product for this study is not appropriate, since there are two effects causing land/water differences in the averaging kernels: thermal contrast effects and the lack of NIR radiances in retrievals over water. The two effects cannot be separated.*

We agree with the reviewer that NIR lacking over water is a potential problem for comparing retrievals made over land and water in TIR-NIR joint products. However, far from making the TIR-NIR joint product inappropriate for this study, this actually strengthens the case for why L/W should not be averaged together in coastal L3 grid boxes in these products, e.g. if the lack of NIR radiances in the retrievals over water makes them so fundamentally different to retrievals over land, then they should not be averaged together. We feel that this emphasises the need for a study like this, which emphasises the consequences of mixing these retrievals, to be published.

For greater clarity, we have added a statement emphasising that a lack of NIR radiances hampers the retrieval over water in the Introduction section; and we reiterate at the end of Section 3.1 that NIR

radiances are not used in the retrievals over water and therefore also contribute to the L-W sensitivity disparity, in addition to near-surface thermal contrast differences:

- Screenshot showing addition to Introduction section:

| 76 | Retrievals that take place over water are known to have a lower information content than retrievals |
| 77 | that take place over land.  Primarily, this is due to weak thermal contrast near to the surface hampering |
| 78 | the instrument's ability to sense CO absorption in the lowermost layers of the troposphere (Deeter et al., |
| 79 | 2007; Worden et al., 2010), and this is confounded by a lack of NIR reflectance over water, which limits |
| 80 | these retrievals to TIR wavelengths only. It is therefore recommended that MOPITT data users exclude these |

- Screenshot showing addition to Section 3.1:

| 414 | This analysis clearly shows how L2 retrievals that are averaged together to create the L3O data over |
| 415 | coastal grid boxes have differing degrees of sensitivity, especially in the LT. This is explicitly cautioned |
| 416 | against in the MOPITT data user's guide (MOPITT Algorithm Development Team, 2018). The remainder of |
| 417 | this paper focuses on the surface-level of the retrieved profile, since the LT is where discrepancies are |

| 418 | greatest, and the cause of this sensitivity disparity is well established: differing thermal contrast conditions |
| 419 | near to the surface over land and water; and a lack of NIR radiances being used in the retrieval over water. |
| 420 | Furthermore, the surface-level is of most interest for identifying potential air quality impacts for humans (e.g. |
| 421 | Buchholz et al., 2022). |
| 422 | |

*2. Use of L3 data to study coastal cities*

*(1) L3 products (either TIR-NIR or TIR) are not suited for the analysis of the coastal cities listed, given the horizontal extent of the targets. A cursory search (please see Table 1 attached) shows that 30 of the 33 cities in Fig. 9 correspond to a very small fraction of a single L3 pixel footprint. Only 3 of the 33 cities are close to covering or barely cover one L3 pixel footprint. Basing such analysis on L2 data could be an*

*adequate choice, at least for some of these cities. About half of the 33 cities would not even fill the footprint of a single L2 retrieval. Only 10 of the 33 cities would fill 4 or more L2 retrieval footprints.*

*(2) According to the manuscript, "L3 data are better suited to long timeseries analysis than L2 data owing to their smaller size". That statement is wrong. Some tools are easier and more convenient to use than others, but that does not mean that they are better suited for a given task. Analyzing long time series with L3 data may be easier, more convenient. However, easy and convenient generally comes at a cost, in this case the quality of the analysis. The manuscript continues "working with L3 data [...] requires fewer computing resources and less technical proficiency [...] L3 products thus make the MOPITT data more easily accessible, especially to less-expert users, who may lack the expertise required to scrutinize the data for potential a priori bias." Again, a tool may be easy/convenient to use but unfit for certain tasks.*

*(3) Time series are at the center of this work and are the justification provided for using L3 data in the first place. The manuscript, however, does not include a single time series. It's hard to imagine that meaningful information/trends can be identified in L3 time series covering a ~6400 days range (from 25 Aug 2001 to 28 Feb 2019) but having only a few hundreds of even a few tens of days with a L3 value at all (and that L3 value coming in all cases from a single L3 pixel). This is the case for most of the cities analyzed (Fig. 9).*

*(4) Are those few hundreds of even few tens of L3 data points representative of the 1ºx1º areas they stand for? L3 pixels (land, water, or mixed) may be produced by averaging as little as 2 L2 pixels. As an example: more than 25% of the total number of daytime pixels in a randomly selected L3 file resulted from averaging either 2 or 3 L2 measurements. These L3 pixels may not be representative of the 1ºx1º area they stand for and, thus, should be filtered out so as not to corrupt the statistical results. It is unclear if such filtering was applied.*

There are several different points made above, and we address each of these separately below (the point numbers below correspond to the numbers that we have inserted in parentheses above):

1. We have clarified in various points throughout the text (notably the Abstract, end of the Introduction, Methods, throughout Section 3.4 where "cities" are analysed, and in the Summary and Conclusions) that our focus is on *the grid boxes containing the cities*, as opposed to the cities themselves. These grid boxes are chosen for illustrative purposes, to demonstrate the impact of choosing to analyse L3 vs L2 data, given that such grid boxes are likely of interest for users of the data interested in e.g. air pollution and human health (whether using L3 or L2). Note that the focus on grid boxes containing cities only makes up one subsection of four in our analysis (Section 3.4):

we focus on *all* coastal grid boxes in Sections 3.1 – 3.3. To address the argument about L3 being inappropriate as the grid boxes are larger than city extents: CO is long-lived and therefore well-mixed horizontally around cities. Hedelius et al. (2021) studies this and shows that trends for cities do not stand out against the background on a scale of a couple of degrees. As outlined above, we have made additional clarifications throughout the text that we are studying the L3 grid boxes that contain these cities, not the cities themselves. If our analysis was restricted to L2 data that fall within city limits, as seems to be suggested, this would result in even fewer data days to base our analysis on, which is something that the reviewer raises as an issue later.

  o Screenshot showing clarification RE grid boxes containing coastal cities in Abstract:

| | |
|---|---|
| 40 | cities, we ask whether results of analyses are significantly different if using L3O compared to L3L. I is |
| 41 | shown that mean VMRs in L3O$_L$ and L3L differ significantly for 11 of the 27  grid boxes that can be |
| 42 | compared (there are no L3O$_L$ data for 6 of the grid boxes studied). The L3L – L3O$_{LM}$ mean VMR |
| 43 | difference exceeds 10 (22) ppbv for 11 (3) of the 33 grid boxes, significant in 13 cases. 9 of the 18  |
| 44 | grid boxes where WLS analysis can be performed in L3O$_L$ feature a trend that is significantly different to |
| 45 | L3L. The trends in L3O$_{LM}$ and L3L differ significantly for 5 of the 33 grid boxes. It is concluded that a |

  o Screenshot showing clarification RE grid boxes containing coastal cities in Summary and Conclusions section:

| | |
|---|---|
| 996 | • Focusing on the L3 grid boxes containing the 33 largest coastal cities in the world, mean VMRs in |
| 997 | L3O$_L$ and L3L differ significantly for 11 of the 27  grid boxes that can be compared (40 %; |
| 998 | there are no L3O$_L$ data for the remaining 6 cities). The L3L – L3O$_{LM}$ mean VMR difference across |
| 999 | all 33  grid boxes is relatively small (3.7 ppbv), but this does hide some much larger |
| 1000 | discrepancies, with the difference exceeding 10 ppbv for 11 of the 33  grid boxes and 20 ppbv |
| 1001 | for 3 of them. The difference is significant for 13 of 33  grid boxes (39 %). Of the 18  grid |
| 1002 | boxes where WLS analysis can be performed in L3O$_L$, there are 9 cases where the trend is |

54

| | |
|---|---|
| 1003 | significantly different to that in L3L. The trends in L3O$_{LM}$ and L3L differ significantly for 5 of the |
| 1004 | 33 grid boxes. |

2.  We accept the reviewer's point that our statement is wrong with respect to L3 data being "better suited" to long time series analysis. We have rephrased this to "L3 data are more convenient for long time series analysis than L2 data owing to their smaller file size" (see screenshot below). Regarding whether or not they are fit for the task: we contest that in most cases *except* for coastal grid boxes, L3 data *are* fit for the task of long time series analysis. This is because, as we outline, they are simply gridded area averages of bounded L2 data. Whether users prefer to analyse L2 or L3 data is then a trade-off between required spatial resolution, temporal coverage, ease of analysis, etc. In addition, we refute the reviewer's judgement that "easy and convenient generally comes at a cost"; one of the key objectives of data provision (in our opinion) should be to minimise barriers to using those data in order to maximise the scientific and societal use that they (and the investment made in acquiring them) can have. If the "quality of the analysis" is one of the costs then there is a problem with the way that the data are being delivered, and this is exactly what our paper seeks to solve.
* * *
112    difference that this would make on a global scale. This is necessary to understand for two reasons: firstly, L3

113    data are  more convenient for long timeseries analysis than L2 data owing to their smaller file

114    size (~25 MB vs ~450 MB respectively, for a single daily, global file). It cannot be overlooked that working
* * *
3.  The issue raised about patchy temporal coverage is unfortunately a broader problem when it comes to analysis of data acquired by satellite instruments, including MOPITT. Beyond selecting a broader averaging area (e.g. a 3 x 3 grid of L3 grid boxes), there is nothing that can be done to improve the temporal coverage in the V8 dataset. The situation is worse for L2 data (unless they averaged across a broad area, which then defeats the purpose of using L2 data). Evaluating whether the presented trends are "meaningful" requires additional verification datasets and is beyond the scope of this study, since the focus of our trend analysis is instead to demonstrate the difference in trends that can be detected using L2 vs L3 data. This was already stated in Sect. 2.5, where our trend detection method is outlined. We have now added that additional data would be needed to verify the trends if the trend values themselves were our focus, and we have reiterated this caveat at the end of Sect. 3.4.3 where trends are discussed for the case study grid boxes containing large coastal cities. Screenshots from both these sections are shown below.

- o Screenshot from Section 2.5 showing clarification that the aim of the trend analysis is for dataset comparison only, and that additional data would be needed for verification:

| | |
|---|---|
| 326 | preferred over OLS because it is less sensitive to outliers. For simplicity, no other trend detection methods – |
| 327 | e.g. the Thiel-Sen slope estimator – are applied to corroborate the trends that are detected with WLS, nor do |
| 328 | we analyse additional datasets to verify them. Such extra steps would be necessary if the actual trend values |
| 329 | were the focus of this study; however, the aim of this trend analysis is instead to identify whether the same |
| 330 | method can yield different results depending on which of L3O, L3L or L3W is analysed. |

- o Screenshot from Section 3.4.3 showing clarification that the trend analysis is illustratory only, and that additional data would be needed for trend verification:

| | |
|---|---|
| 932 | As outlined in Sect. 2.5, it is important to note that the trends presented in this section are for |
| 933 | illustrative purposes only, with the intention of demonstrating that different results can be obtained depending |
| 934 | on whether L3O or L3L (and, by extension, L2) data are analysed. More focused analysis is needed to verify |
| 935 | these trends, which is beyond the scope of this paper. |
| 936 | |

4. We do not filter our time series based on the number of L2 retrievals that are averaged together to create the L3 data which we are analysing. We do this because a) doing so would further limit the temporal sampling of the products analysed; and b) no such filtering is performed in the creation of the L3O product, available for public download, which is the focus of this study. To the reviewer's point about the representativeness of a L3 retrieval that is only created from a small number of L2 retrievals within the L3 grid box; this misses the point of our study. We are comparing the L2 retrievals performed over land and water that are averaged together to create L3O products for coastal grid boxes, and arguing that the retrievals over water shouldn't be used as they can be very different to those over land, with retrieval sensitivity differences an explanatory factor. Whether the retrievals being compared are representative of the grid box as a whole is not the focus of our study. Assuming a coastal L3 grid box has data for a given day in the as-downloaded L3O product, we compare the bounded L2 retrievals to identify whether there is evidence of a systematic difference between those made over land and water – and ultimately to demonstrate that the retrievals over water shouldn't be used when creating the L3O product.

---

## Referee Report (RR1)

**Referee report**

**Differences in MOPITT surface-level CO retrievals and trends from Level 2 and Level 1 3 products in coastal grid boxes**

The authors presents a comparison of results from analyses performed using original L3 MOPITT data products, and a new land-only L3 product ("L3L" ) and a water-only L3 product ("L3W) that have been created from L2 products, for all MOPITT L3 grid boxes that overlay coastlines. Comparing the full L3O dataset to L3L, it is shown that if L3O is filtered so that only retrievals over land (L3OL) are analyzed, there is a huge loss of days within the data. This is because L2 retrievals over land are routinely discarded during the L3O creation process. Even by retaining L3OM (mixed) retrievals, the resulting L3O "land or mixed" (L3OLM) subset still has less data days than L3L for 61 % of coastal grid boxes. The loss of data influenced the results where it is shown that, the mean VMRs in L3OL and L3L differ significantly for 11 of the 27 grid boxes that can be compared. They concluded that a L3 product based only on L2 retrievals over land – the L3L product analyzed in this paper, could be of benefit to MOPITT data users, given the significant differences in mean CO VMRs and trends that can be obtained for coastal grid boxes using L2 products.

**The paper can be published with minor changes.**

The main challenge in the paper is the length and the clarity. Paper needs to be shortened and rewritten in a clear way. It is very hard for regular user to follow up the flow of the paper.

---

## Author Response (AR2)

We sincerely thank the associate editor for their helpful feedback on this paper. We address all their comments and suggestions below.

*Associate editor comments are in black italicised text.*

Our responses are in blue, regular text.

Where practical/necessary, we provide a screenshot of the track-changes document to show the changes that we have made (in outlined boxes). In these, text that is removed is  and coloured red, while new text is underlined and coloured blue.

Ian Ashpole (on behalf of both authors).

*Dear authors,*

*Thank you for providing the revised version of your manuscript. While your response addresses most of the reviewer comments, I think some key points are currently not addressed sufficiently.*

*One major aim of your work is to demonstrate that a L3 "land-only" product (L3L) is better suited for studying CO VMRs and trends in coastal grid boxes than the official L3 product (L3O), which has issues because retrievals over land and water are often combined. You show in detail the difference between the L3L and L3O dataset. However, in order to show that a dataset is better, it is also necessary to show that the calculated VMRs and trends are more accurate. These points are brought up by reviewer #2, who questions the feasibility to compute "meaningful" mean values and trends from the dataset (Point 2.3 and 2.4). To consider your manuscript for publication, I think it is necessary that you consider the following points in more details:*

*(1) Please provide time series for the different coastal grid boxes to assess the temporal coverage for the different datasets.*

A focused case study, including more detailed time series analysis, is now included at the start of the city focus section (new Section 3.4.1; screenshot below). This clearly demonstrates the stark improvement in temporal coverage of the L3L dataset, when compared to $L3O_L$ (the subset that users of L3O are encouraged to use) and the differences in detected temporal trends and mean VMRs. This also demonstrates the trend and mean VMR differences between L3L and $L3O_{LM}$, and how they are clearly linked to the inclusion of retrievals over water in $L3O_{LM}$ (by comparison with results from L3W).

**3.4. Illustrative examples comparing L3O and L3L: analysis of the most populous coastal cities**

In this section, we analyse time series from the 33 L3 coastal grid boxes that contain cities classified amongst the 100 most populous in the world (derivation outlined in Sect. 2.5) to illustrate the differences between mean values and trends obtained from the L3O and L3L datasets.

 We focus our comparison on L3O$_L$ and L3O$_{LM}$, as these are the L3O subsets that data users would realistically choose to analyse if following the data filtering guidelines. For clarity,  from here these grid boxes are referred to by the name of the city that they contain. A detailed case study for the L3 grid box containing the city of Dubai is first presented, before considering results for all cities analysed.

**3.4.1. Detailed case study: L3 grid box containing Dubai**

Summary stats derived from the L3O subsets, L3L, and L3W (included for comparison), for the L3 grid box containing the city of Dubai, are given in Table 5. Figure 8 visualises the daily retrieved VMR time series from L3L, with L3O$_L$ overlaid for comparison purposes.
* * *
Note, we have also made one additional change to the results presentation in Section 3.4 (specifically, this change is relevant to what is now Section 3.4.2 but spanned Sect. 3.4.1 and 3.4.2 in the original submission): Instead of presenting mean VMRs and temporal trends (with standard deviations and standard errors as error bars, respectively) for every city in figure form (Fig. 8 and 9, resp., in the original submission), we have replaced the figures with a single table (new Table 6). The benefit of this is that it enables us to present all of the information in one place in as clear a fashion as possible, and makes the sample sizes of each dataset compared more obvious to readers. It also enables us to easily include information on the relative land/water surface cover for each grid box, as requested in your second point (see below). We also bore in mind that Reviewer 1 had some difficulty interpreting the meaning of different line styles/symbols in the figures in their original review, and although they seemed satisfied by how we dealt with this (noting their recommendation of "publish as is" following review), the figure complexity increased once we addressed the need to make them more legible for readers with color deficiencies. In summary, we feel that this new presentation is much simpler for readers to follow. The text discussing the results in Sect. 3.4 is largely unaffected by this change, with the exception of minor superficial changes required to account for the replacement of Figures 8 and 9 with Table 6.

Regarding concerns from original Reviewer #2 about the "meaningful(ness)" of trends presented in this paper and how this is affected by patchy temporal coverage: this is ultimately limited by the temporal coverage of the MOPITT instrument itself, and is something that we cannot affect. However, it does not prevent scientists from presenting trends based on these data in publications (e.g. Buchholz et al. 2022 https://doi.org/10.1038/s41467-022-29623-8). The central point that we demonstrate in our paper is that our L3L dataset is favorable over the original L3 data in coastal areas due to 1) demonstrably more days with land-only data due to less discarded retrievals; and 2) demonstrably greater information content in the retrievals being regressed when L3O filtering is allowed to include retrievals over water (the $L3O_{LM}$ subset that we analyse). Both of these points increase the likelihood that mean VMRs and trends in L3L are more "meaningful" than in the original L3 data in these coastal areas, although it is obviously still constrained by limitations in instrument coverage. However, this does not mean that the results have no scientific significance.

*(2) Please analysis the spatial coverage of L2 pixels in the L3 grid boxes to understand how the different datasets are affected by sparse sampling in the grid boxes.*

This was already done in our original submission, but on reflection we can understand how it was not clear. For greater clarity, we have added an explanation to the methods section (Sect. 2.4) about how the spatial coverage of the L2 pixels in L3 grid boxes is analysed (screenshot below). We introduce the metric ratio(land/water) to quantify the proportion of the grid box surface that is covered by land vs water, and refer to this in the following sections when variable surface coverage is discussed as an explanation for results: Sect. 3.2.2 (screenshot below), Sect. 3.4.1, Sect. 3.4.2.

| | |
|---|---|
| 293 | L3L and L3W retrievals. Additionally, the number of L2 retrievals that are used for calculating the area |
| 294 | averages when creating L3L and L3W ("$n\_ret_L$" and "$n\_ret_W$", respectively) is recorded. The ratio |
| 295 | $n\_ret_L/n\_ret_W$ (herein referred to as "ratio(land/water)" for simplicity) is used to indicate the proportion of |
| 296 | the L3 grid box that is covered by land vs water: a ratio of 1 indicates an even split of these surface types in |
| 297 | the grid box; a ratio < 1 indicates that a greater proportion of its surface is water covered; and a ratio > 1 |
| 298 | indicates that the grid box is land-dominated. |

•    The grid boxes of BOTH tend to have a greater proportion of their surface covered by water than land
when compared to L3L_L3W_ONLY. This is determined by analysis of ratio(land/water) values for
each grid box (derivation of this metric is outlined in Sect. 2.4). quantified by comparing the mean
number of L2 retrievals over land and water that are averaged together to make L3L and L3W each
day ("n_ret(L3L)" and "n_ret(L3W)"), for each coastal grid box compared. A mean n_ret(L3L/L3W)
ratio ratio(land/water) of 0.87 for BOTH indicates a greater water influence on L3O$_M$ than for the grid
boxes of L3L_L3W_ONLY, for which a mean ratio(land/water) n_ret(L3L/L3W) ratio of 1.00
indicates a more even land/water split. Thus, L3O$_M$ more closely resembles L3W – which is
significantly different to L3L – in BOTH than in L3L_L3W_ONLY.

*(3) Please conduct a suitable statistical analysis for computing means and trends, which can show that the L3L dataset increases accuracy of the analysis.*

The Theil-Sen slope estimator has been used to verify that the trends presented in the paper, calculated using Weighted Least Squares (WLS) regression, do not change depending on the method used (Theil-Sen is nonparametric and therefore less sensitive to outliers). These are included in the Supp. Mat. (SM7) for reference (referred to in the text).

Beyond this, we are unclear about what a "suitable statistical analysis for computing means and trend" constitutes. We asked for clarification on this from the Associate Editor in a personal communication and were directed that a different method for calculating trends might be necessary (email dated August 15 2022) – hence, our inclusion of Theil Sen analysis.

*(4) Please validate your mean values and trends with independent measurements for some cases to proof that such a product would be useful. I am aware that it might be impossible to find suitable measurements for such an analysis.*

As acknowledged in this request, we have been unable to find surface-level CO measurements for any of the coastal cities analysed to validate the mean values and trends compared. CO data *are* available for some of these cities, but these are restricted to total column measurements as part of the TCCON network. We do not consider the CO total column in this paper. Despite the lack of verification, which we feel is beyond the scope of this paper, the results are still of scientific significance in that they 1) demonstrate that the scientific value of a publicly available satellite dataset (MOPTIT L3O) is lessened in certain situations (coastal grid boxes) as a direct consequence of how it is created from finer resolution parent data (MOPITT L2), and 2) present a solution to this. Note that our L3L dataset is now published, as recommended below.

*Please also consider the following suggestion: In your manuscript, you conclude that a L3 "land-only" product would be beneficial to the research community. I noticed that your study already creates this product. Since AMT strongly encourages the publication of underlying data, I suggest that you publish your "land-only product for coastal grid boxes" in a public data repository (e.g., zenodo.org) and revise your manuscript changing the focus to describe your new dataset, analyzing the difference to the official L3 product and show that it is better suited for computing means and trends. This small change would enhance the scientific significance of your manuscript. Note that this would still require addressing the points above.*

Thank you for this suggestion. The L3L dataset (and L3W, for comparison) have been uploaded to a public data repository and are available for download from the following link: https://doi.org/10.5683/SP3/ERCG2H. We have modified the text in the Abstract, Introduction (screenshot below), Data and Methods, and Conclusion (screenshot below) sections of the paper to reflect the fact that these are now publicly available datasets, with download instructions given in the Data Availability section. Please note that this has not significantly changed the focus of the paper, which remains demonstrating L3O shortcomings over coastal grid boxes.

The full citation information for the published L3L and L3W datasets is:

"Ashpole, I., and Wiacek, A.: Land- and water-only Level 3 products from MOPITT TIR-NIR Version 8 CO retrievals, https://doi.org/10.5683/SP3/ERCG2H, Borealis, V1, 2022".

> 117    especially to less-expert users, who may lack the expertise required to scrutinize the data for potential a priori
> 118    bias. Secondly, many of the world's largest agglomerations are situated within a coastal L3 grid box (5 of
> 119    the top 10 and 33 of the top 100 largest agglomerations by population; derivation outlined in Sect. 2.5),
> 120    making these likely targets for analyses of air quality indicators, especially their changes over time.
> 121    This paper presents a comparison of results from analyses performed using original, "as downloaded"
> 122    L3 data products, and a new  land-only  L3 product ("L3L", Ashpole and Wiacek
> 123    (2022) – outlined in Sect. 2.4) that has been created  from L2 products, for all MOPITT L3 grid
> 124    boxes that overlay coastlines (a water-only L3 product "L3W" has also been created for comparison
> 125    purposes) . Section 2 describes the datasets
> 126    and methods used, including outlining the creation of the new L3L and L3W data products analysed in this
> 127    paper. Section 3.1 demonstrates the magnitude of the sensitivity difference for retrievals over land and water,

**4. Summary and Conclusions**

Motivated by the work of Ashpole and Wiacek (2020) which demonstrated, for the MOPITT L3 grid box containing the coastal city of Halifax, Canada, that mean VMR statistics and temporal trends differ depending on whether L2 or L3 data are analysed, this paper has examined what proportion of all coastal L3 grid boxes also see differences between results from analyses performed with L2 and L3 data. While it is recommended to MOPITT data users that analyses are restricted to retrievals performed over land owing to known sensitivity issues over water (MOPITT Algorithm Development Team, 2018; Deeter et al., 2015), such recommendations cannot practically be followed by users of L3 data for coastal grid boxes owing to the way the data are created from their bounded L2 retrievals. In short, this study has sought to answer the question:

"does it matter"? Analysis has focussed on comparing the original, "as-downloaded" L3 dataset ("L3O")

with new land-only and water-only L3 products ("L3L" and "L3W" respectively) that have been created from the L2 retrievals. The main results are summarised below.

*Please feel free to contact me, if you have any questions.*

Sincere thanks for your help and attention with this.

*Kind regards*

*Gerrit Kuhlmann*

*PS: Many of your figures are difficult to access for readers with color vision deficiencies. Please have a look at the guidelines on figures and tables on the AMT website:*

*https://www.atmospheric-measurement-techniques.net/submission.html#figurestables*

Apologies for overlooking this in our original submission. We have modified the following Figures: Fig. 1, Fig. 5, Fig. 6, Fig. 7. We have also replaced Figs 8 and 9 from the original submission with a Table (Table 6), which we feel is a much more straightforward presentation of the results.

---

## Author Response (AR3)

We sincerely thank both reviewers for their helpful feedback on this paper, and the associate editor for their time with this. We address all the reviewer comments and suggestions below.

*Reviewer comments are in black italicised text.*

Our responses are in blue, regular text.

Where practical/necessary, we provide a screenshot of the track-changes document to show the changes that we have made (in outlined boxes). In these, text that is removed is  and coloured red, while new text is underlined and coloured blue.

Note that the revised submission also contains a change to one of the tables, following a notification that we received from the Editorial Support Team at the previous submission stage. The change is detailed at the end of this document, along with our response to one other request for information/clarification from them regarding figure copyrights.

Ian Ashpole (on behalf of both authors).

**REVIEWER REPORT #1**

**(Anonymous referee #3)**

*Suggestions for revision or reasons for rejection*
*The study analyzes the difference, impact and vertical sensitivity of coastal pixels (land, water or mixed) from the L2 MOPITT v8 surface CO retrievals used and averaged for L3 products. The study is of interest to the scientific community using satellite retrievals of CO. Comments can be found below.*

*• The manuscript is well written but can be difficult to follow with all the annotations used. I would recommend using a table in the methodology describing all datasets in this study (L3OL, L3L, L3O, ..).*

Thank you for this suggestion. We have included this in the Data and Methods section as a new Table 1, and referenced it where appropriate when outlining the datasets used in the study.

376

**Table 1.** List of dataset short names used in the main article text, and their corresponding full descriptive name.

| Dataset short name | Full descriptive dataset name |
|---|---|
| L3O | Original, "as downloaded" Level 3 (L3) dataset |
| $L3O_L$ | Subset of L3O only containing L3 retrievals with a surface index of land |
| $L3O_M$ | Subset of L3O only containing L3 retrievals with a surface index of mixed |
| $L3O_{LM}$ | Subset of L3O only containing L3 retrievals with a surface index of land OR mixed |
| $L3O_W$ | Subset of L3O only containing L3 retrievals with a surface index of water |
| $L3O_{NF}$ | The L3O dataset with no filtering by surface index ($L3O_{NF}$ is identical to L3O) |
| L3L | A new L3 "land-only" dataset, created only from Level 2 retrievals performed over land (creation method outlined in Sect. 2.4) |
| L3W | A new L3 "water-only" dataset, created only from Level 2 retrievals performed over water (creation method outlined in Sect. 2.4) |

377

In addition, to aid readers in following the paper we have included a list of short names and abbreviations used throughout the text, along with their full descriptive name, purpose for use and section introduced as an Appendix at the end of the paper (Appendix 1).

We have also made small changes to the text throughout the paper to aid with clarity and flow where appropriate, and have thoroughly edited the abstract, introduction, and conclusion with a focus on

stating the paper aim more clearly. The focus of the writing is now more clearly on the surface level in Section 3.1.

• *Ln.195. Further details on surface index used in MOPITT should be provided. How is defined the surface index in the L2 retrievals. What are the uncertainties associated with this surface index?*

This is a great question, but one that, unfortunately, we are unable to answer. We have searched all available Product User Guides, ATBD's, Data Quality Summaries, and product validation / algorithm description papers listen on the NCAR MOPITT website (https://www2.acom.ucar.edu/mopitt/) to get an answer for this (including looking at the documentation for older product versions), but to the best of our knowledge there is no information publicly available explaining where the surface index for L2 retrievals is actually derived from. The lack of transparent information on this matter, and the fact that it is not mentioned in any publication by the MOPITT Science team, suggests to us that whatever method is used to define the surface index in L2 retrievals may be standard practise, with little uncertainty about the L2 surface indexes prescribed.

Without reliable information on this matter, we feel uncomfortable speculating, and have therefore not modified the paper in response to this comment. We hope this is understandable/acceptable, given the lack of further information.

• *Ln. 230. It is suggested that these inland false coastal grid boxes would be linked to flood and appear some of the time. Have you looked at this assumption (using cumulative precipitation or soil moisture data)? What do you mean by "some of the time"? Have you looked by season, or month? If one of the cause is due to surface ice cover or flood, this should be observed depending on seasons.*

We have not tested this assumption by analysing the seasonal distribution of "false coastal" grid boxes, or using different data sources (e.g. precipitation or soil moisture, as suggested) to verify their occurrence due to e.g. flooding or ice coverage. Our reasoning is that the aim of this step is to identify coastal grid boxes to study ("2.3. Coastal grid box classification for this study"), as opposed to analysing the distribution of grid boxes with a surface index of "mixed", and potential reasons why they get this flag. Our discussion of "false coastal" grid boxes stems from the fact that some inland areas are also classified as "mixed", which meant that we could not simply use the presence of a surface index of "mixed" as a coastal identifier (Fig.

2a in the paper). Additional analysis around this subject is beyond the scope of the paper, and would add unnecessary complication and additional length. We hope this is acceptable.

Reflecting on the text in this section (2.3), we wonder if it is the use of the term "false coastal grid boxes" that has prompted this question, since the word "false" implies that something is in error – which is not the case. These grid boxes are simply classified as mixed "some of the time" (see below for refinement of that terminology). There is no error. To help prevent such questions from arising, we have replaced the term "false coastal" with "inland_mixed", since this better reflects the actual grid box classification. We have also made minor changes to some of the wording in this section, to greater aid clarity.

Regarding what is meant by "some of the time": we have replaced this with the wording "at least once during the study period" to remove the vagueness.

| | |
|---|---|
| 286 | true for coastal grid boxes. However, analysis of the global distribution of L3 grid boxes featuring a surface |
| 287 | index of mixed revealed that, in addition to actual coastlines, a large proportion of inland grid boxes that are |
| 288 | clearly not coastal ("false coastal") are given the surface index of mixed at least once during |
| 289 | the study period ("inland_mixed"; Fig. 2a). The reason for this is unclear, but it could be for real physical |
| 290 | reasons, such as land grid boxes sporadically flooding, or due to issues in the retrieval schemes caused by |
| 291 | e.g. cloud screening problems or the presence of surface ice cover. One characteristic of these  |
| 292 | inland_mixed grid boxes is that, compared to the total number of days with L3, the relative frequency |
| 293 | with which they are flagged as land is very high (expressed as the ratio "n_days(L3O$_L$/L3O)")", plotted in |
| 294 | Fig. 2b; a list of short names and abbreviations referred to in the text can be found in Appendix A for |
| 295 | reference). This relative frequency is much lower for "true" coastal grid boxes, to be expected given prior |

• *The study works at global scale and for a 20 years period. It is difficult to learn if the sensitivity of AVK for instance in section 3.1.1 is similar for every regions across the globe. Additionally, what would have been the results if analyzed by seasons and by regions/latitudes (depending on ice cover, swamps, ..)? This is an information missing which can be of interest for the users.*

This is a fair comment, and we have now included a brief analysis of latitudinal and seasonal variability in the land-water retrieval sensitivity contrast in the Supplementary Material (SM3). The analysis shows that there is indeed some seasonal and latitudinal variance in the magnitude of the land-water sensitivity contrast: there is a tendency for greater land-water retrieval sensitivity differences in the Northern Hemisphere than Southern Hemisphere when averaged across the year, with some nuances by season,

although no clear and obvious patterns. A land-water sensitivity gradient is generally evident irrespective of latitude or season.

Given that this analysis does not alter the overall results of the paper, and that discussing the nuances would add words and figures to an already quite long and complex paper (as noted in reviewer report #2), we feel that the Supp. Mat. is the best location for it.

| 470 | An analysis of latitudinal and seasonal variability in the land-water surface level retrieval sensitivity |
|---|---|
| 471 | contrast is provided in the Supp. Mat. (SM3). Briefly, this shows a tendency for greater land-water retrieval |
| 472 | sensitivity differences in the Northern Hemisphere than Southern Hemisphere when averaged across the year. |
| 473 | The land-water AK rowsum differences tend to vary least by season in the tropical regions (between 30° |
| 474 | South and 30° North) and show the greatest contrast in the midlatitudes (30° – 60°) in the respective |
| 475 | hemisphere's spring and summer months, with smallest differences in the winter months. Overall, a land- |
| 476 | water sensitivity contrast is evident irrespective of latitude or season. |

• *Ln. 456. Which additional physical factors could play a role?*
*You have analyzed the impact of wind on the land-water difference observed in the retrieved VMRs at different locations around the world. But one area of interest you do not mention is the transport of CO concentrations from African fires to the coastal lands of South America. During Africa's fire season, CO concentrations can be found higher over ocean than on land along the African or Brazilian coasts. Have you examined and considered specific cases like that?*

Good point. We have not studied this specific case, but do already mention in this section that ocean air is not necessarily "pristine" compared to land-based air (see the quotation below). We have added a sentence more explicitly acknowledging that in some specific cases CO VMRs can be greater over water than land (see below), but err away from including additional case studies and detail at this stage. Note that whether CO VMRs are higher over land or water is not the central point in question in this paper; it is the magnitude of the difference, which we argue is too great to be explainable by physical factors alone and therefore brings the land-water contrast in retrieval sensitivity into question. This issue is navigated by our use of absolute land-water CO VMR differences (i.e. ignoring whether the CO VMR is greater over land or water), when relating them to land-water differences in retrieval sensitivity parameters.

563 relatively small distance covered by a L3 grid box. Given the relatively long-lived, well-mixed nature of
564 atmospheric CO, VMRs retrieved at a given location are a function of both local emissions *and* transport,
565 and the portion of coastal L3 grid boxes situated over water therefore do not represent pristine conditions in

22

566 comparison to the adjacent land-based portion of the grid boxes. This is verified by comparing a priori VMRs
567 (also shown in Fig.ure 4), which suggest the land-water difference in CO concentrations should be negligible
568 (mean L3L – L3W a priori VMR difference = 0.69 ppbv, compared to a mean retrieved VMR difference of
569 10.29 ppbv). Indeed, in some specific cases – e.g. uninhabited coastal areas downwind of large trans-oceanic
570 pollution sources – VMRs may be higher over the water portion of coastal gridboxes than the adjacent land
571 portion (note that Fig. 4 does show that this is the case in some grid boxes). The above reasoning can also be

• *Ln. 569. Could you detail what further information about the "true" VMR is needed?*

Apologies if this sentence caused confusion. What we mean by this is that the actual ("true") VMR amount that is being measured would need to be known if an assessment about the accuracy of a priori is to be made. Without this knowledge we can only speculate, but in the context of this section it is possible to make an informed guess when the information about retrieval sensitivity (given by averaging kernels and proxied by VMR ret-apr values) is available.

We have modified the text to make it more clear what we mean by "information about the "true" VMR":

671 corresponding L3L retrieval. Interestingly, it is also notable that retrieved and a priori VMRs are lower in
672 BOTH$_{VMRs}$ than in L3L_L3W_ONLY$_{VMRs}$, and that retrieved minus a priori VMR values are greater in
673 BOTH$_{VMRs}$ than in L3L_L3W_ONLY$_{VMRs}$. This could imply that the a priori VMRs are closer to reality (i.e.

28

674 the a priori CO amount is closer in value to the actual ("true") CO amount that is being measured) for the
675 grid boxes of L3L_L3W_ONLY$_{VMRs}$ than those of BOTH$_{VMRs}$, however further information on "true" VMRs
676 is required to properly assess this it would be necessary to know what the actual "true" VMR values are that
677 are being measured.

• *Table 5 and Ln. 736. "L3OL trend being significantly stronger than the L3L [...] given the far superior temporal coverage of L3L, this is the more reliable result". L3OL is the only dataset with less than 90% days with data and with the larger standard deviation (+/- 56 ppbv). While L3L has a superior temporal coverage, its standard deviation is important (+/- 44 ppbv) in comparison to the other datasets. How do you explain this large standard deviation with L3L, not observed with L3OLM and L3W?*

Good question. The greater standard deviation in L3L (and L3O$_L$) than in L3O$_{LM}$ and L3W is a function of the greater sensitivity in retrievals over land than over water. The more sensitive retrievals over land can deviate more from their a priori VMRs – and therefore reach higher (and lower) retrieved VMR values – than the retrievals over water which, due to their lower sensitivity, are tied more closely to their a priori VMRs, resulting in lower variance and standard deviation. The observation of reduced variability in retrievals over water than land is discussed in detail in Section 3.2.1 ("Retrieved VMR comparison between L3L and L3W") – specifically, this sentence makes the point clearly: "This may be explained as follows: when sensitivity over water is especially low…the retrieved VMR will be heavily weighted by the a priori and unable to match the variation present in the more sensitive retrieval over land. As sensitivity over water increases, this a priori weighting weakens and the retrieved VMR will more closely track the retrieval over land." (This quote is from L487 – 492 in the revised manuscript).

We have added a sentence highlighting the greater standard deviation in retrievals over land, which is an expected finding given the discussion in earlier sections of the paper:

| | |
|---|---|
| 836 | mean differences are significant (p < 0.1). Consistent with the results shown in Sect. 3.2.2 when identifying |
| 837 | factors that determine whether the averaging of L2 retrievals over land and water to create L3O$_M$ can yield |
| 838 | statistically significantly different retrievals to L3L, this L3 grid box is water-dominated, with a mean |
| 839 | ratio(land/water) of 0.60. It is also notable that the standard deviation of long-term mean retrieved VMR in |
| 840 | L3L (and L3O$_L$) is roughly twice as large as that in L3O$_{LM}$ and L3W, which is to be expected given that |
| 841 | retrievals over water are more greatly tied to their a priori than retrievals over land due to their comparatively |
| 842 | lower sensitivity (as discussed in Sect. 3.2.1). |

• *Legend Fig. 1. NASA of nasa blue Marble should be in capital letters.*

Thank you for pointing this out. The change has been made.

classification is outlined in Sect. 2.3. The coastal L3 grid box visualized here contains the city of Dubai (~centre = 55.296° E, 25.277° N), which features in the case study analysis of Sect 3.4. Faint background shading is from  NASA Blue Marble imagery.

**REVIEWER REPORT #2**

**(Anonymous referee #4)**

*Suggestions for revisions or reasons for rejection*

*Paper conclusion and abstract need to be summarized in a more clear way. The objective of the paper needs to be clarified.*

We apologise for the lack of clarity. The abstract, introduction, and conclusion have been edited with a focus on stating the paper aim more clearly. We have also made some minor additional changes to the text throughout the paper to aid with clarity and flow where appropriate.

*Notes from Report:*

*The authors presents a comparison of results from analyses performed using original L3 MOPITT data products, and a new land-only L3 product ("L3L" ) and a water-only L3 product ("L3W) that have been created from L2 products, for all MOPITT L3 grid boxes that overlay coastlines. Comparing the full L3O dataset to L3L, it is shown that if L3O is filtered so that only retrievals over land (L3OL) are analyzed, there is a huge loss of days within the data. This is because L2 retrievals over land are routinely discarded during the L3O creation process. Even by retaining L3OM (mixed) retrievals, the resulting L3O "land or mixed" (L3OLM) subset still has less data days than L3L for 61 % of coastal grid boxes. The loss of data influenced the results where it is shown that, the mean VMRs in L3OL and L3L differ significantly for 11 of the 27 grid boxes that can be compared. They concluded that a L3 product based only on L2 retrievals over land – the L3L product analyzed in this paper, could be of benefit to MOPITT data users, given the significant differences in mean CO VMRs and trends that can be obtained for coastal grid boxes using L2 products.*

*The paper can be published with minor changes.*

*The main challenge in the paper is the length and the clarity. Paper needs to be shortened and rewritten in a clear way. It is very hard for regular user to follow up the flow of the paper.*

As noted above, the abstract, introduction, and conclusion have been edited with a focus on stating the paper aim more clearly. We have also made some minor additional changes to the text throughout the paper

to aid with clarity and flow where appropriate (e.g., the focus of the writing is now more clearly on the surface level in Section 3.1). In addition, to aid readers in following the paper we have included a list of short names and abbreviations used throughout the text, along with their full descriptive name, purpose for use and section introduced as an Appendix at the end of the paper(Appendix 1).

Please note that we have been unable to make any notable reduction to the length of the paper: everything that is not essential to the paper aims has already been placed in the Supplementary Material.

**RESPONSE TO NOTIFCATION FROM EDITORIAL SUPPORT**

*Notification to the authors:*

*Checking your paper, I noticed that your table 6 contains coloured cells. Please note that this will not be possible in the final revised version of the paper due to HTML conversion of the paper. When revising the final version, you can use footnotes or italic/bold font. For now, the process will continue, but please note that the final version cannot be published by using coloured tables. Please check if your Figures S5 to S10 (maps/aerials) require a copyright statement and add it to the figure captions. If you are the originator, you can just inform us.*

Thank you for informing us of these requirements. Table 7 (was table 6 in previous submission) has been edited to remove coloured shading from cells, which has been replaced with the use of bold and/or italics where appropriate.

Please also note that we do not need a copyright statement for any figures in supp matt; these were all produced by the authors.